# Recombination between heterologous human acrocentric chromosomes

Andrea Guarracino[1,2], Silvia Buonaiuto[3], Leonardo Gomes de Lima[4], Tamara Potapova[4], Arang Rhie[5], Sergey Koren[5], Boris Rubinstein[4], Christian Fischer[1], Human Pangenome Reference Consortium*, Jennifer L. Gerton[4], Adam M. Phillippy[5], Vincenza Colonna[1,3] & Erik Garrison[1✉]

The short arms of the human acrocentric chromosomes 13, 14, 15, 21 and 22 (SAACs) share large homologous regions, including ribosomal DNA repeats and extended segmental duplications[1,2]. Although the resolution of these regions in the first complete assembly of a human genome—the Telomere-to-Telomere Consortium's CHM13 assembly (T2T-CHM13)—provided a model of their homology[3], it remained unclear whether these patterns were ancestral or maintained by ongoing recombination exchange. Here we show that acrocentric chromosomes contain pseudo-homologous regions (PHRs) indicative of recombination between non-homologous sequences. Utilizing an all-to-all comparison of the human pangenome from the Human Pangenome Reference Consortium[4] (HPRC), we find that contigs from all of the SAACs form a community. A variation graph[5] constructed from centromere-spanning acrocentric contigs indicates the presence of regions in which most contigs appear nearly identical between heterologous acrocentric chromosomes in T2T-CHM13. Except on chromosome 15, we observe faster decay of linkage disequilibrium in the pseudo-homologous regions than in the corresponding short and long arms, indicating higher rates of recombination[6,7]. The pseudo-homologous regions include sequences that have previously been shown to lie at the breakpoint of Robertsonian translocations[8], and their arrangement is compatible with crossover in inverted duplications on chromosomes 13, 14 and 21. The ubiquity of signals of recombination between heterologous acrocentric chromosomes seen in the HPRC draft pangenome suggests that these shared sequences form the basis for recurrent Robertsonian translocations, providing sequence and population-based confirmation of hypotheses first developed from cytogenetic studies 50 years ago[9].

Although the human reference genome is now 22 years old[10], fundamental limitations of the bacterial artificial chromosome (BAC) libraries on which it was built prevented its completion. Incomplete regions amount to 8% of the Genome Reference Consortium's Human Build 38, and include heterochromatic regions in the centromeres and the SAACs. Advances in long-read DNA sequencing have recently enabled the creation of a complete reference assembly—T2T-CHM13[3]—from a homozygous human cell line, providing a reference system for these regions for the first time. In parallel, our ongoing work in the HPRC has yielded 94 haplotype-resolved assemblies for human cell lines (HPRCy1) based on the same Pacific Biosciences circular consensus (HiFi) sequencing that forms the foundation of T2T-CHM13[4]. These resources enable us to characterize patterns of variation in these previously invisible regions. Here we study variation in the largest non-centromeric regions made visible in T2T-CHM13 and HPRCy1— those between the centromere and the ribosomal DNA (rDNA) on the

SAACs, where Robertsonian translocations[11] (ROBs), the most common human translocation events, frequently occur.

Eighteen of the twenty-three human chromosomes are metacentric, with the centromere found in a median position between short (p) and long (q) arms, whereas 5 are acrocentric, featuring one arm that is substantially shorter than the other. The SAACs (chromosome (chr.)13p, chr. 14p, chr. 15p, chr. 21p and chr. 22p) host the nucleolus organizer regions, the genomic segments that contain rDNA genes and that give rise to the interphase nucleoli[12,13]. Owing to their repetitive nature, rDNA repeat arrays facilitate intramolecular recombination[14]. rDNA repeats incur double-strand breaks at a high rate owing to transcription–replication conflicts[13]. Moreover, rDNA from multiple acrocentric chromosomes can be co-located in nucleoli during interphase, and multiple acrocentric chromosomes often co-localize to a single nucleolus during the pachytene stage of meiosis[15], when chromosomes synapse and recombine. As it causes them to occupy the same constrained physical

[1]Department of Genetics, Genomics and Informatics, University of Tennessee Health Science Center, Memphis, TN, USA. [2]Genomics Research Centre, Human Technopole, Milan, Italy. [3]Institute of Genetics and Biophysics, National Research Council, Naples, Italy. [4]Stowers Institute for Medical Research, Kansas City, MO, USA. [5]Genome Informatics Section, Computational and Statistical Genomics Branch, National Human Genome Research Institute, National Institutes of Health, Bethesda, MD, USA. *A list of authors and their affiliations appears at the end of the paper. ✉e-mail: egarris5@uthsc.edu

space, the positioning of rDNA-adjacent sequences in proximity to the nucleolus could be a driver of genetic exchange between heterologous chromosomes (Supplementary Note 1), given that estimates[15] of the probability of two regions adjacent to the nucleolus-organizing region being colocalized are 120,000 times higher than colocalization in a human spermatocyte nucleus. In line with this, distal and proximal sequences to rDNA repeat arrays are conserved among the acrocentric chromosomes, suggesting that recombination homogenizes them[1,2]. Experimental and sequence-based evidence indicates the presence of a common subfamily of alpha satellite DNA shared by acrocentric pairs chr. 13–chr. 21 and chr. 14–chr. 22 that provides evidence for an evolutionary process consistent with recombination between heterologous chromosomes[16]. Furthermore, ROBs—which occur in 1 out of 800 births—are most common between chr. 13 and chr. 14 (around 75% of cases), and between chr. 14 and chr. 21 (around 10% of cases), but the underlying sequences and recombination processes that drive them remain unknown[8].

The T2T-CHM13 reference fully resolves the genomic structure of the SAACs, confirming their strong similarity and providing a complete view of the homologies in this single genome[3]. However, T2T-CHM13 does not provide information on how SAACs vary among the human population and additional genomes are needed to understand whether the representation in T2T-CHM13 is typical. Notably, alignments of HPRCy1 assemblies to T2T-CHM13 reveal individual contigs with optimal alignments to multiple CHM13 acrocentric chromosomes, suggesting possible translocations[4]. This analysis was necessarily relative to only a single frame of reference used as target in alignment, leaving open questions regarding the relationships between pairs of HPRCy1 haplotypes. A complete study of this region thus requires improvements in both sequence assembly and pangenome analysis to enable an unbiased assessment of its structure and variation in the population. Here we combine T2T-CHM13 and HPRCy1 assemblies in a reference-free pangenome variation graph (PVG) model of the SAACs. Using this model and other symmetric analyses of T2T-CHM13 and the HPRCy1 assemblies, we establish a coherent model of population-scale variation in the SAACs.

## Chromosome community detection

We sought to study the chromosome groupings implied by the homologies found in all 94 assemblies of the HPRCy1 pangenome. We used homology mapping to build a reference-free model of the HPRCy1 pangenome, represented as a mapping graph with nodes as contigs and edges as mappings between them. The graph was built using chains of 50-kb seeds of 95% average nucleotide identity—features that we expect to support homologous recombination[17]—with up to 93 alternative mappings allowed per contig. After applying this process to all 38,325 HPRCy1 contigs and narrowing our focus to only mappings involving contigs that are at least 1 Mb in size, we built a reduced mapping graph by selecting the best 3 mappings per contig segment and labelling each contig with its reference-relative assignment (Fig. 1a). This simplified graph showed clusters that generally matched our expectations of higher similarity between certain chromosomes[18,19] (Fig. 1b). For a more quantitatively rigorous interpretation, we used a community detection algorithm (Methods) to divide the full mapping graph into 31 communities (Supplementary File 1). These communities were consistent with our expectations based on mapping the contigs to reference chromosomes T2T-CHM13 and GRCh38 and known patterns of similarity between chromosomes. We found that the community of the SAACs contained the most distinct chromosomes and the most contigs (Fig. 1c,d). Many contigs from the pseudoautosomal regions (PARs) and X-transposed regions (XTRs) of chromosome X[20] and all of those from chromosome Y formed one community, and others from the short arm of chromosome X—including all of those from evolutionary strata 4 and 5[21]—formed another (Extended Data Fig. 1). A few additional communities were identified that did not correspond to individual chromosomes, but typically represent single chromosome arms.

## An all-acrocentric PVG

We constructed a pangenome graph from acrocentric contigs in the HPRCy1 draft pangenome to evaluate the hypothesis that heterologous SAACs recombine. We first collected long HPRCy1 contigs that span the acrocentric centromeres and can be assigned to specific acrocentric chromosomes (Extended Data Fig. 2). We then used the PanGenome Graph Builder[22] (PGGB) to construct a single PVG from these contigs (Methods). PVG nodes represent sequences and edges indicate when concatenations of the nodes they connect occur in the contigs represented by the graph[23]. By relating pangenome sequences to the graph as paths of nodes[5], PVGs support base-level analysis of variation and homology between genomes[4,24–26]. The symmetric all-to-all alignment[27] and graph induction[28] of PGGB avoid sources of bias such as reference choice and genome inclusion order that affect progressive PVG construction methods[28]. For cross-validation of our results, we additionally include two assemblies of HG002 in the PVG: HG002-HPRCy1[4]—obtained from HiFi reads, and HG002-Verkko—a T2T diploid assembly constructed from both HiFi and Oxford Nanopore Technologies (ONT) reads as described in Methods.

The resulting acrocentric PVG (acro-PVG) presents structures that echo those observed in T2T-CHM13 and the community structure of the homology mapping graph (Fig. 2a and Supplementary Files 2 and 3). In more detail, the main connected component including all chromosomes presented a tangled region, anchored at the rDNA repeats and extending towards the centromere-proximal end of the short arms. The alpha satellite higher-order repeat arrays in the centromeres of chr. 13–chr. 21 and chr. 14–chr. 22 pairs shared high similarity within each pair[18,19], leading to collapsed motifs in the graph (Fig. 2b). The chr. 13–chr. 21 and chr. 14–chr. 22 pairs diverge in centromere-proximal regions of the q-arms. Furthermore, a region in the pangenome graph centred on the GC-rich SST1 array was present in a single copy in chr. 13, chr. 14 and chr. 21, indicating a high degree of similarity of genomes in those regions (Fig. 2c and Supplementary Fig. 1). This is compatible with the frequent involvement of these regions of chr. 13, chr. 14 and chr. 21 in ROBs[8,29]. The SST1 elements in the segmentally duplicated region are GC-rich 1.4- to 2.4-kb-long sequences arranged in tandem clusters[30], located throughout the genome including near the centromeres of the SAACs chr. 13, chr. 14 and chr. 21[31]. The SST1 array size is variable in the human population[32] and its methylation status is clinically relevant to cancer[33]. SST1 repeats on chr. 13, chr. 14 and chr. 21 in T2T-CHM13 are highly similar to each other[31], consistent with homogenization via recombination. All the graph motifs described in the acro-PVG were also confirmed by building a pangenome graph without including the T2T-CHM13 and GRCh38 references (Supplementary Figs. 2 and 3), indicating that the observed structure is independent of the reference assemblies.

## Exchange among heterologous acrocentric regions

The acro-PVG provides a representation of the multiple alignment of SAACs found in the human population. In the acro-PVG, we observe many regions in the graph where multiple T2T-CHM13 chromosomes are aligned. We expect these regions to potentially support homologous recombination, which largely depends on sequence homology and physical proximity, both of which are common among heterologous SAACs[15,34].

## HPRCy1 contigs are homology mosaics

We sought to test the hypothesis that homologous regions of the SAACs feature ongoing sequence exchange by searching for regions in the

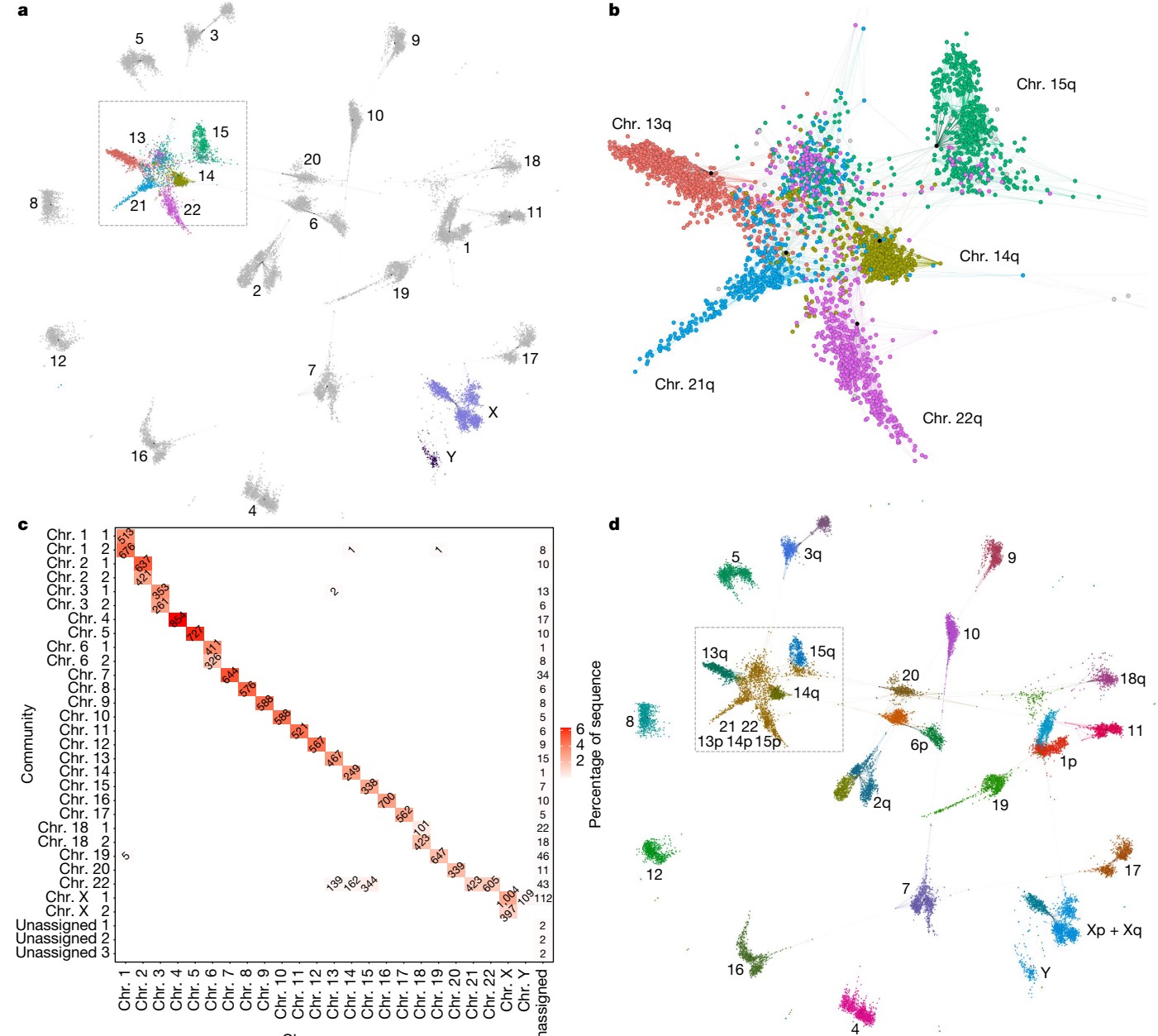

**Fig. 1 | Community detection in the HPRCy1 pangenome. a**, The reduced all-to-all mapping graph of HPRCy1 against itself, with contigs represented as nodes and mappings as edges. Colours distinguish the acrocentric or sex chromosome to which each contig was assigned by competitive mapping against T2T-CHM13 and GRCh38, with text labels indicating the chromosome for each visual cluster. **b**, A close-up view of the region indicated in **a** and **d** containing nearly all contigs that match acrocentric chromosomes. **c**, Results of community assignment on the mapping graph. The *x*-axis shows the chromosome to which contigs belong, based on competitive mapping to T2T-CHM13 and GRCh38; the *y*-axis indicates the community, which is named

according to the chromosome that contributes the largest number of contigs to it. In the squares, the numbers indicate how many contigs belong each specific chromosome and community and the shade indicates the percentage of the total assembly sequence present in the set. The sex chromosomes and the acrocentric chromosomes participate in the only clusters that mix many (more than 100) contigs belonging to different chromosomes. **d**, The reduced homology mapping graph in **a**, coloured according to community assignment (colours do not correlate with those in **a** or **b**). The p-arms of chr. 13, chr. 14 and chr. 15, and all of chr. 21 and chr. 22 form one community, and chr. Y and most of chr. X form another.

acro-PVG where individual contigs are best described as a mosaic of diverse T2T-CHM13 acrocentric chromosomes. We derived a pair-wise alignment from the acro-PVG through 'untangling'[26], a process that projects the graph into an alignment between a set of query (HPRCy1 acrocentric (HPRCy1-acro)) and reference (T2T-CHM13) sequences, jointly considering all possible alignments represented by the pangenome graph. The untangling of the acro-PVG against multiple T2T-CHM13 chromosome reference sequences simultaneously

shows the best match of segments within contigs to multiple reference chromosomes.

The hypothesis of recombination between heterologous acrocentric chromosomes implies that the HPRCy1-acro contigs untangled from the acro-PVG will be a mosaic of diverse acrocentric chromosomes in the regions undergoing homologous recombination. The same would not be true for flanking regions that should map to one specific chromosome.

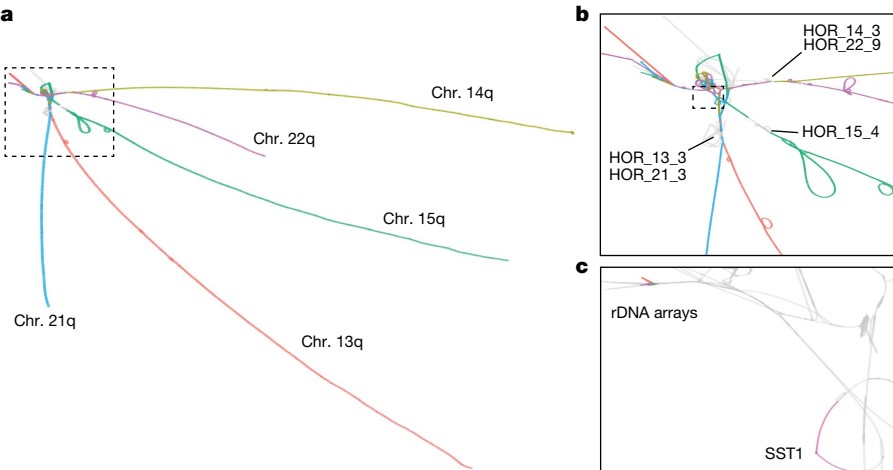

**Fig. 2 | The acro-PVG derived from the HPRCy1 assembly. a**, The major component of the acro-PVG, shown with nodes in T2T-CHM13 chromosomes labelled with the colour scheme from Fig. 1a. The acrocentric q-arms are almost completely separated, whereas the p-arms unite in a tangle adjacent to the rDNA array. **b**, A close-up view of the SAAC junction, showing the separation of centromeric high-order repeats of chr. 15 (HOR_15_4) from the other chromosomes, whereas chr. 13 and chr. 21, and chr. 14 and chr. 22 share substantial homology in their arrays, which causes them to collapse in the PVG.

A few assemblies span the rDNA array into its distal junction, which presents as a single homologous region across all chromosomes[2], and then fray into diverse sequences visible as tips in the top left. **c**, Closer view of the outlined region in **b**, focusing on the segmentally duplicated core centred in the SST1 array and the rDNA arrays, as labelled in T2T-CHM13. The highlighted region around the SST1 array is in the same orientation on T2T-CHM13 chr. 13p11.2 and chr. 21p11.2, and is inverted on chr. 14p11.2; these 3 regions have a pairwise identity[3] of more than 99%.

We queried the PVG[26] to obtain a mapping from segments of all PVG paths onto T2T-CHM13. This segments the graph, and for each HPRCy1 contig (query) subpath through each graph segment, we find the most similar reference segment (Extended Data Fig. 3). To reduce the possibility of error, we focused the alignment projection only on the confidently assembled regions of the HPRCy1-acro contigs[4] (Methods) and we filtered the mappings to retain only those at greater than 90% estimated identity, removing a total of 1.17 Gbp, or 2.52% of the total SAAC contig segments (Supplementary File 4 and Supplementary Figs. 4–8).

For a reference-relative interpretation of the results, we anchored the contigs to the single T2T-CHM13 reference chromosomes to which the q-arm maps (Methods), providing a reference-relative positioning of contigs in the PVG. We find that the q-arm of each contig maps to a single chromosome, whereas the p-arm is a mosaic of segments mapping to several acrocentric chromosomes (Fig. 3a,b). Results for all the acrocentric chromosomes are shown in Extended Data Figs. 4–8.

We cross-validated homology mosaic patterns by comparing the reference-relative anchored untangling of HG002-Verkko to HG002-HiFi, obtaining a 87.45% concordance rate in the SAACs and a 99.93% concordance rate in the acrocentric q-arms (Methods). Although HG002-HiFi contains only one contig that would meet our HPRCy1 contiguity requirements, we observe broadly concordant patterns in the two assemblies (Supplementary Figs. 9–13) and visually confirm patterns—such as those between chr. 13p and chr. 21p—that are seen in many HPRCy1 assemblies (Supplementary Fig. 9).

## Homology mosaicism grows across SAACs

We counted the number of contigs that best match each of the T2T-CHM13 acrocentric chromosomes within the PVG (Fig. 3c). On the q-arm, all contigs best match their homologous T2T-CHM13 chromosome, agreeing with the observed structure of the PVG (Fig. 2a). However, as we approach the centromere from the q-arm, we observe regions of homology between chr. 13 and chr. 21 (Fig. 3c and Extended Data Figs. 4b and 7b) and between chr. 14 and chr. 22 (Extended Data Figs. 5b and 8b). By contrast, homology with other acrocentric chromosomes begins closer to the rDNA in chr. 15 (Extended Data Fig. 6b), corroborating the pattern observed in the PVG topology (Fig. 2b).

Although the higher-order repeat arrays on chr. 13 and chr. 21 and on chr. 14 and chr. 22 are both collapsed in the PVG (Fig. 2b), we observe sparse identity mappings higher than 90% within the centromeres (Fig. 3b and Extended Data Figs. 4a, 5a, 6a, 7a and 8a). This is consistent with other reports of high divergence within centromeric satellites[35]. HPRCy1 contigs anchored on the q-arms of chr. 13, chr. 14 and chr. 21 share a segmental duplication (or homologous region) centred on the SST1 array (Fig. 3b), in line with what is seen in the pangenome graph topology (Fig. 2c and Supplementary Figs. 1 and 3). Furthermore, as in T2T-CHM13, this region is in the same orientation on chr. 13 and chr. 21, but is inverted on chr. 14 (Supplementary Figs. 14–16). All chromosomes provide similarly good matches for contigs in the regions immediately proximal and distal to the rDNA. However, this is supported by relatively few (nine) q-arm-anchored contigs that purport to cross the rDNA—loci that we do not expect to assemble correctly using current sequencing approaches[3].

To assess the homology between the acrocentric chromosomes, we developed a metric that captures the degree of disorder in the untangling of HPRCy1-acro contigs over 50-kb regions of T2T-CHM13 (Methods). This metric, 'regional homology entropy', is greater than 0 in regions where contigs match multiple T2T-CHM13 chromosomes—a pattern indicative of recombination. We find that regional homology entropy increases as we progress over each short arm and reaches a maximum immediately on the proximal flanks of the rDNA arrays (Fig. 3d). We observed an equivalent increase of regional homology entropy in the PARs on chr. X and chr. Y (Supplementary Fig. 17), which are known to actively recombine.

## Acrocentric PHRs

Our analyses suggest that regions of near-identity between multiple T2T-CHM13 chromosomes are capable of supporting large-scale homologous recombination. To study the boundaries of these regions, we derived a multiple untangling, which orders by identity multiple T2T-CHM13 matches for every contig segment (Supplementary Figs. 18–22). The order of T2T-CHM13 hits captures the leaf order of a HPRCy1 contig-rooted phylogeny[36]. Differences in chromosome-relative phylogenies across haplotypes indicate different evolutionary histories

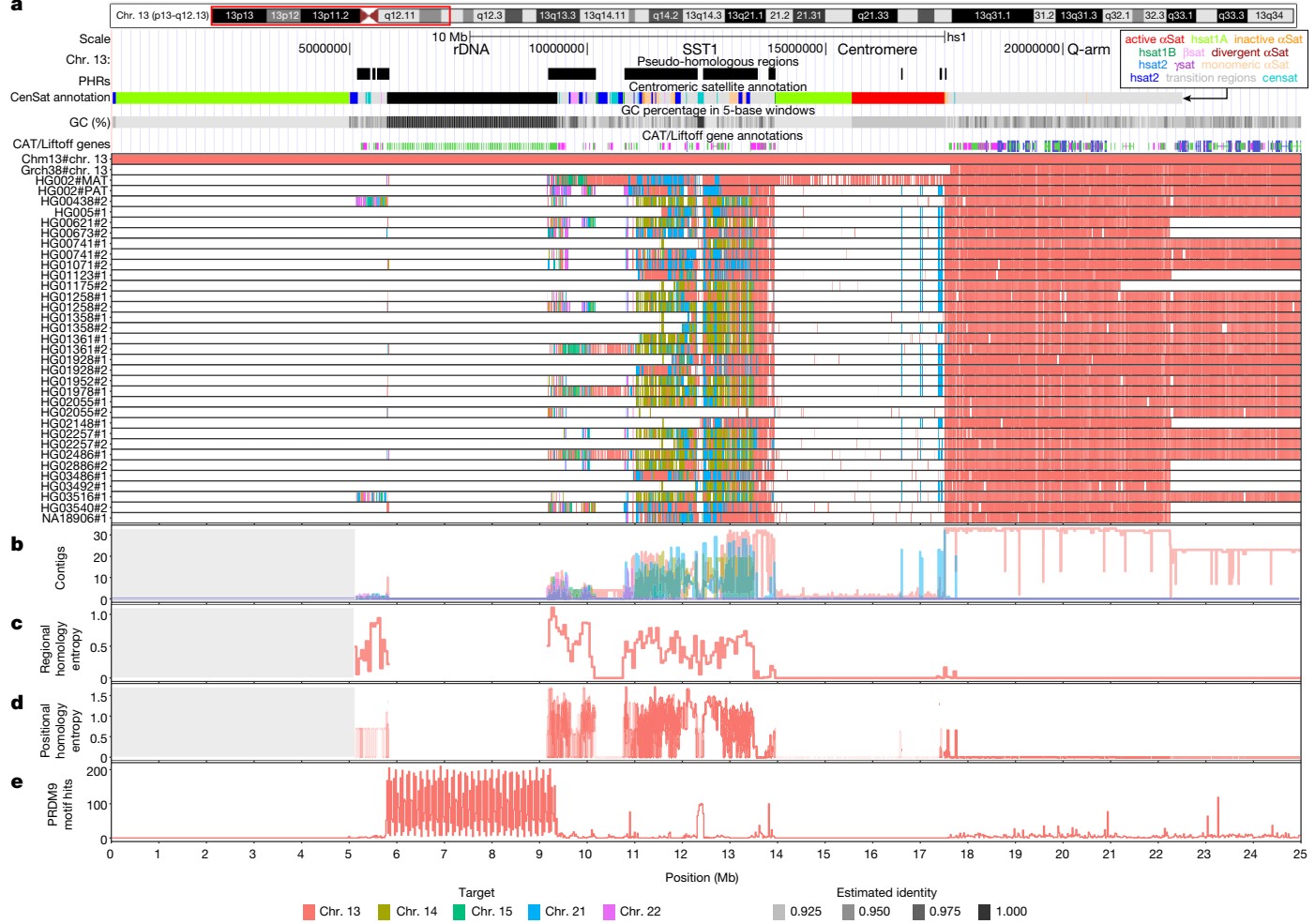

**Fig. 3 | Characteristics of the PHRs of acrocentric chromosomes. a**, We focus on the first 25 Mb of chr. 13, shown here as a red box over T2T-CHM13 cytobands. PHRs are highlighted relative to T2T-CHM13 genome annotations for centromere and satellite repeats (CenSat annotation), GC percentage and genes (CAT/Liftoff genes). Top, regions of interest described in the main text: rDNA, the SST1 array, the centromere and q-arm. Bottom, relative homology mosaics based on the T2T-CHM13 assembly for each chr. 13-matched contig from HPRCy1-acro, with colours indicating the most similar reference chromosome (target). **b,c,d**, Aggregated untangle results in the SAACs. **b**, The count of HPRCy1 q-arm-anchored contigs mapping to each acrocentric chromosome (Contigs) aggregated by target chromosome and (**c**) the regional (50 kb) untangle entropy metric (Regional homology entropy) computed over the contigs' untangling relative to T2T-CHM13. **d**, By considering the multiple untangling of each HPRCy1-acro contig, we develop a point-wise metric that captures diversity in homology patterns relative to T2T-CHM13 (Positional homology entropy), leading to our definition of the PHRs. **e**, The patterns of homology mosaicism suggest ongoing recombination exchange in the SAACs. A scan over T2T-CHM13 reveals that the rDNA and SST1 array units are enriched for PRDM9 binding motifs, and thus may host frequent double-stranded breaks during meiosis. In **b**–**d**, the grey background indicates regions with missing data due to the lack of non-T2T-CHM13 contigs.

and imply ongoing recombination[37], which leads to chequerboard patterns in the multiple untangling plots (Fig. 4c). To delineate regions where heterologous chromosomes are likely to recombine, we computed 'positional homology entropy'—a measure of the diversity of reference-relative phylogenies—for each position in T2T-CHM13 (Fig. 3e and Supplementary Fig. 23). We consider regions with positional homology entropy greater than 0 over more than 30 kb to be candidates for ongoing recombination (Methods).

These PHRs total 18.329 Mb in length (Supplementary File 5) and differ in size by chromosome: chr. 13, 4.53 Mb (Fig. 3b); chr. 14, 6.48 Mb (Extended Data Fig. 5a); chr. 15, 719.25 kb (Extended Data Fig. 6a); chr. 21, 3.79 Mb (Extended Data Fig. 7a); and chr. 22, 2.81 Mb (Extended Data Fig. 8a). We term them PHRs by analogy to the PARs of sex chromosomes, because these homology domains could enable non-homologous chromosomes to pair like homologous chromosomes. Notably, the chromosomes involved in the most common ROBs (chr. 13–chr. 14 and chr. 14–chr. 21) have larger PHRs, which could promote the recombination

events that lead to these translocations. Supporting this, BAC clones surrounding common recurrent ROB breakpoints[8] map to T2T-CHM13 PHRs (Supplementary Fig. 24). A genome-wide phylogenetic analysis of SST1 array elements indicates the expected pattern of concerted evolution by chromosome, but the repeats from chr. 13, chr. 14 and chr. 21 display a unique pattern of concerted evolution between chromosomes (Fig. 4a) and furthermore, share a deletion of around 1.0 kb relative to all other SST1 repeats (Fig. 4b and Supplementary Fig. 25), suggesting inter-array recombination similar to the surrounding non-satellite sequences of the PHRs (Fig. 4c). We confirmed that patterns observed by fluorescent in situ hybridization of these BACs are compatible with breakpoints occurring in the PHRs centred at the SST1 array (Fig. 4d).

To provide a positive control, we applied the same method to the sex chromosome PVG to identify their PHRs (chr. X, 2.75 Mb and chr. Y, 2.73 Mb; Supplementary File 6 and Supplementary Fig. 26). These regions precisely match the established boundaries for the PARs and contain sparse hits in the XTRs, which would be compatible with reports

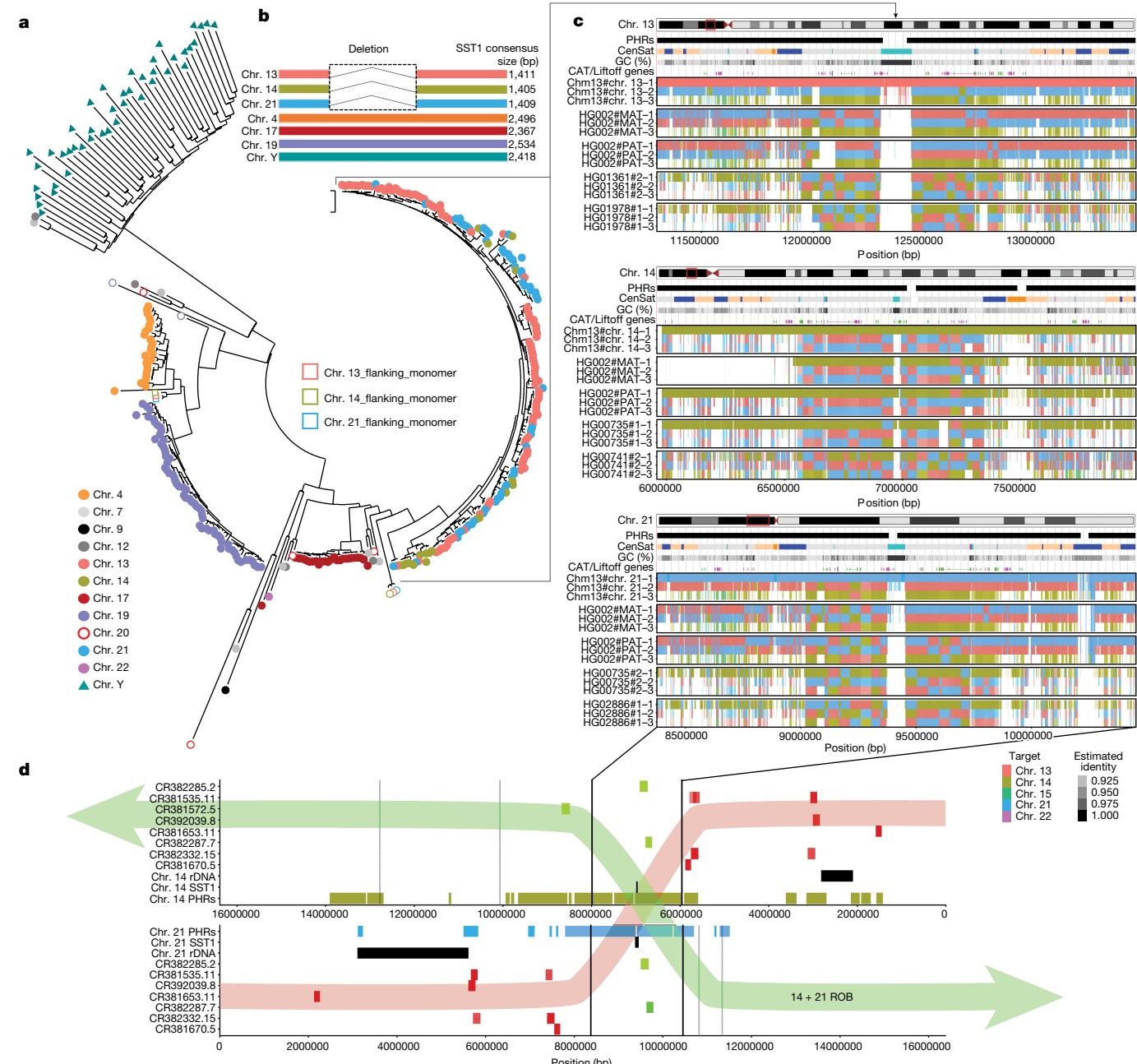

**Fig. 4 | PHRs of chr. 13, chr. 14 and chr. 21, centred on the SST1 array.**
**a**, Maximum likelihood phylogenetic analysis of SST1 full-length elements indicates a recent homogenization process of acrocentric arrays. Coloured circles next to chromosome labels indicate individual monomers retrieved from the T2T-CHM13 assembly. Coloured triangles indicate SST1 full-length monomers retrieved from the HG002 chr. Y assembly. Partial or chimeric monomers flanking chr. 13, chr. 14 and chr. 21 arrays (located around 250 kb from the main array) are labelled as open circles or squares, respectively, coloured according to the corresponding chromosome. **b**, Schematic representation of SST1 consensus alignments, indicating a deletion that is present only in the SST1 unit from arrays on chr. 13, chr. 14 and chr. 21. **c**, Multiple untangling of T2T-CHM13, HG002-Verkko haplotypes and HPRCy1-acro contigs versus T2T-CHM13. Three of the five acrocentric chromosomes are represented. The degree of transparency indicates the estimated identity

of the mappings. All mappings above 90% estimated pairwise identity are shown. To enable the display of simultaneous hits to all acrocentric regions, each grouping shows the first three best alternative mappings. SST1 arrays described in **a** are at the centre of a PHR that displays chequerboard patterns indicative of recombination between heterologous acrocentric chromosomes (black arrows link the SST1 arrays in all panels). These patterns are less common on chr. 15 and chr. 22 (Supplementary Figs. 20 and 22). **d**, The PHRs on T2T-CHM13 (yellow and light blue) in relation to BACs localized cytogenetically[8] to recurrent chr. 14–chr. 21 ROB breakpoints. BACs shown in green are found in dicentric Robertsonian chromosomes, whereas those in red are not. Chr. 14 is shown in an inverted orientation aligned to chr. 21 at the breakpoint region suggested experimentally[8]. In a transparent overlay, we propose a retained dicentric chromosome (14+21 ROB, green) and lost (red) products of the studied recurrent translocations.

of X–Y interchange in the XTR[38]. Biallelic SNP calls from a whole-genome HPRCy1 graph released in the accompanying Article[4] show that variant density in the PARs and the acrocentric p-arms is markedly higher than elsewhere in these chromosomes (Supplementary Fig. 27),

which is consistent with increased rates of recombination in these regions[20].

In humans and many other mammals, the sequence specificities of the DNA-binding zinc finger protein PRDM9 regulate the formation of

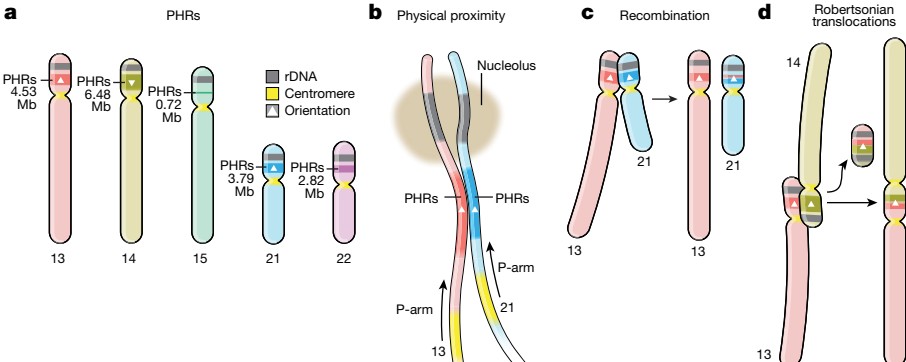

**Fig. 5 | The PHRs of human acrocentric chromosomes. a**, PHRs are found on the rDNA-proximal regions of the SAACs chr. 13, chr. 14, chr. 15, chr. 21 and chr. 22. **b**, PHRs physically co-locate owing to their proximity to the nucleolar organizing regions and rDNA, encouraging sequence exchange. **b**, Patterns of sequence similarity observed in the PHRs indicate ongoing recombination exchange between heterologous chromosomes, in particular chr. 13, chr. 14 and chr. 21, which may be mediated by both non-crossover recombination or crossover of the telomeric ends of heterologous chromosomes. **d**, The PHR surrounding the SST1 arrays on chr. 13, chr. 14 and chr. 21 is nearly identical on all three chromosomes, but is typically inverted on chr. 14 relative to chr. 13 and chr. 21 (triangles). Owing to the inversion, crossover type recombination between PHRs in chr. 14 and chr. 13 or chr. 21 produce an ROB.

double-stranded breaks that drive meiotic homologue synapsis and recombination[39,40]. We scanned T2T-CHM13 acrocentric chromosomes for PRDM9 motifs detected by chromatin immunoprecipitation with sequencing[41] (Extended Data Fig. 9 and Supplementary File 7), finding that both rDNA and SST1 arrays are enriched for PRDM9 motifs relative to the surrounding sequence (Fig. 3f and Extended Data Figs. 4e, 5e, 6e, 7e and 8e). By contrast, we find almost no PRDM9 motifs in the centromeres, where meiotic recombination is harmful and suppressed by diverse mechanisms[42].

## Linkage disequilibrium decay in PHRs

To quantify the magnitude of putative recombination in the PHRs, we calculated the rate of the linkage disequilibrium decay between SNPs detected in the acro-PVG[43] (Supplementary Fig. 28). For each acrocentric chromosome, we plot the $R^2$ allele correlation versus distance for three sets of pairs of variants separated by up to 4 kb: variants on the q-arm, on the p-arm, and within the PHRs. The overall trend of linkage disequilibrium decay on the q-arms is similar to trends seen in other datasets that evaluate linkage disequilibrium in humans[44,45]. On chr. 13 the decay of linkage disequilibrium in PHRs and the p-arm is similar and faster than on the q-arm, and on chr. 14, chr. 15 and chr. 22, linkage disequilibrium decay in PHR is even faster on PHRs compared with the p-arm. The same trend does not apply to chr. 21 (Extended Data Fig. 10).

The fast linkage disequilibrium decay in the PHRs compared to q-arms supports the hypothesis of ongoing recombination exchange. In general, there is a higher level of linkage disequilibrium on the p-arms than on the q-arms, perhaps owing to lower recombination in heterochromatic regions[46]. However, for the majority of acrocentric chromosomes, we observe the opposite in the PHRs. This effect is clearest within the chromosomes that other analyses suggest share the most homologous sequence: chr. 13, chr. 14, chr. 21 and chr. 22, whereas chr. 15—which appears to be an outlier[7] (as also observed in Figs. 1 and 2)—contains shorter PHRs and we have less confidence in the linkage disequilibrium decay trends (error bars on Extended Data Fig. 10, chr. 15 PHRs). This pattern is consistent with an increased recombination rate in the PHRs[6,7].

## Discussion

Here we develop multiple lines of evidence indicating active recombination between heterologous human acrocentric chromosomes. First, we find that a symmetric comparison of the sequences of a draft human pangenome contains multi-chromosome communities corresponding to both the sex chromosomes and acrocentric chromosomes (Fig. 1). An acrocentric pangenome graph reveals base-level homologies that outline patterns of exchange between the heterologous chromosomes (Fig. 2). The graph highlights regions featuring a diverse patchwork of best-match patterns involving non-homologous T2T-CHM13 chromosomes (Fig. 3), and we cross-validate these findings with a T2T diploid assembly of a target sample. We develop an entropy metric sensitive to recombination between heterologous chromosomes—such as that seen between chr. X and chr. Y—and apply it to delineate PHRs where heterologous SAACs recombine (Fig. 4 and Supplementary File 5). Finally, we show that on chr. 13, chr. 14, chr. 21 and chr. 22, the resulting 18 Mb of sequence in the PHRs presents a higher rate of linkage disequilibrium decay than seen in sequences from non-PHR regions of the same chromosomes (Extended Data Fig. 10). These lines of evidence all suggest that heterologous SAACs recombine.

BACs used in a previous cytogenetic study of Robertsonian chromosomes map to the PHRs of chr. 14 and chr. 21, with the recurrent breakpoint region found in a highly homologous region on chr. 13p, chr. 14p and chr. 21p centred on the PRDM9 motif-enriched SST1 array. This leads us to propose that PHRs are maintained by recombination between heterologous chromosomes, which occasionally results in an ROB (Fig. 5). We posit that these homologous regions (Fig. 5a) might share a biological function as sequences proximal to the nucleolar organizing regions. Their proximity (Fig. 5b) can facilitate inter-chromosomal recombination (Fig. 5c)—which may be of both crossover or non-crossover types and may occur during meiosis or mitosis. Owing to an inversion of this region on chr. 14p relative to chr. 13p and chr. 21p, crossover type recombination between pairs chr. 13–chr. 14 and chr. 14–chr. 21 leads to ROBs (Fig. 5d), which our study suggests are a pathological outcome of otherwise benign recombination between heterologous chromosomes.

The HPRC pangenome provides base-level resolution of homology patterns across many SAAC haplotypes, enabling us to examine in detail the regions in which ROBs occur. We observe that the GC-rich SST1 array lies at the centre of a segmentally duplicated region on chr. 13p, chr. 14p and chr. 21p, which shows a clear pattern of haplotype mixing between these chromosomes. This may also implicate this array as a nucleation point for recombination, as suggested by the observation (in 1 out of 220 oocytes) of heterologous chr. 14p–chr. 21p synapse formation in pachytene oocytes[29]. We speculate that these segmentally duplicated regions are where common ROBs occur, a hypothesis supported by our reanalysis of previous cytogenetic mapping of the common ROB breakpoint for chr. 14 and chr. 21 (Fig. 4d).

Although we find that SAACs present challenges for assembly methods[4], our validation based on ONT and HiFi data integration shows that the patterns that we observe in HiFi-only assemblies are consistent with ONT reads from the same sample. HG002-Verkko recapitulates key T2T-CHM13-relative untangling patterns also seen in HPRCy1 HiFi assemblies, such as the SST1-linked PHR at chr. 13p11.2 and chr. 21p11.2, rDNA-proximal mixing of all SAACs, and mixing of chr. 22q11.21 and chr. 14q11.2. Our analyses rest on the extensive assembly validation carried out by the HPRC, and observations used to establish signals for recombination are based only on assembly regions deemed to be reliable by mapping analyses[4]. Our study confirms previous hypotheses based on decades of diverse inquiry, which provide additional assurance that patterns observed bioinformatically are biologically grounded. This body of evidence suggests that our definition of the PHRs is likely to evolve with improved resolution of rDNA arrays and distal regions of the SAACs, which remain among the most challenging regions of the human genome to assemble and lie beyond the scope of our presented work.

Our study is fundamentally population-based. We cannot directly observe recombination of SAACs in this context, leaving open questions about recombination mechanisms that may be difficult to resolve from sequence information alone[47]. However, similar to mutation, recombination is a rare event, which makes it easier to measure its distribution over chromosomes in a population genetic context as we have done here. This addresses key issues with many previous studies of recombination in the SAACs, which often feature small numbers of individuals[29,48] selected on the basis of medically relevant genomic states such as trisomy and ROB[49]. Our resolution of the acrocentric PHRs confirms reported homologies between the SAACs[2,3], providing a reference for their structure that will be useful for future genomic and cytogenetic studies. In principle, recombination in the PHRs may be of either crossover or non-crossover type. Our data support both, but outside of recurrent ROBs and our expectation that non-crossover recombination is substantially more common (by a ratio of around 10:1) than crossover recombination[50,51], we lack distinguishing evidence for either. To estimate the relative rates of each type of event, we can use linkage disequilibrium patterns[50] to study the PHRs in large genomic cohorts[52,53], which will require realigning cohort short read data to T2T-CHM13 or the HPRC pangenome. Future improvements to assembly of the SAACs and the planned increase in the number of individuals included in the HPRC should allow for confident estimates of the relative rates of recombination types.

The co-location of rDNA repeats from different acrocentric chromosomes in a nucleolus provides physical proximity that can facilitate recombination events, both between rDNA repeats and between the adjacent PHRs. Our analyses suggest that the rate of recombination between heterologous pairs of acrocentric chromosomes varies, leading to characteristic patterns in the homology spaces that we have explored. Human cells generally have fewer nucleoli (between one and five) than acrocentric chromosomes[54] (ten). One possibility is that groups of acrocentric chromosomes between which we observe stronger homology and recombination—such as chr. 13, chr. 14 and chr. 21—may be more likely to co-localize to the same nucleolus, as observed in pachytene spermatocytes[15,55]. Proximity, homology, recombination initiation sites and sequence orientation are likely to be factors in the high rate of ROBs between these chromosomes. The HPRC draft human pangenome has enabled us to approach genome evolution from a chromosome scale. By stepping away from a reference-centric model and directly comparing whole-chromosome assemblies of the acrocentric regions, we have obtained sequence-resolved responses to long-standing questions first posed in early cytogenetic studies of human genomes.

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

**Human Pangenome Reference Consortium**

Haley J. Abel[6], Lucinda L. Antonacci-Fulton[7], Mobin Asri[8], Gunjan Baid[9], Carl A. Baker[10], Anastasiya Belyaeva[9], Konstantinos Billis[11], Guillaume Bourque[12,13,14], Silvia Buonaiuto[3], Andrew Carroll[9], Mark J. P. Chaisson[15], Pi-Chuan Chang[9], Xian H. Chang[8], Haoyu Cheng[16,17], Justin Chu[16], Sarah Cody[7], Vincenza Colonna[1,3], Daniel E. Cook[9], Robert M. Cook-Deegan[18], Omar E. Cornejo[19], Mark Diekhans[8], Daniel Doerr[20], Peter Ebert[20], Jana Ebler[20], Evan E. Eichler[10,21], Jordan M. Eizenga[8], Susan Fairley[11], Olivier Fedrigo[22], Adam L. Felsenfeld[23], Xiaowen Feng[16,17], Christian Fischer[1], Paul Flicek[11], Giulio Formenti[22], Adam Frankish[11], Robert S. Fulton[7], Yan Gao[24], Shilpa Garg[25], Erik Garrison[1], Nanibaa' A. Garrison[26,27,28], Carlos Garcia Giron[11], Richard E. Green[29,30], Cristian Groza[31], Andrea Guarracino[1,2], Leanne Haggerty[11], Ira Hall[32,33], William T. Harvey[10], Marina Haukness[8], David Haussler[8,21], Simon Heumos[34,35], Glenn Hickey[8], Kendra Hoekzema[10], Thibaut Hourlier[11], Kerstin Howe[36], Miten Jain[37], Erich D. Jarvis[21,38], Hanlee P. Ji[39], Eimear E. Kenny[40], Barbara A. Koenig[41], Alexey Kolesnikov[9], Jan O. Korbel[42], Jennifer Kordosky[10], Sergey Koren[5], HoJoon Lee[39], Alexandra P. Lewis[10], Heng Li[16,17], Wen-Wei Liao[7,32,43], Shuangjia Lu[32], Tsung-Yu Lu[15], Julian K. Lucas[8], Hugo Magalhães[20], Santiago Marco-Sola[44,45], Pierre Marijon[20], Charles Markello[8], Tobias Marschall[20,46], Fergal J. Martin[11], Ann McCartney[5], Jennifer McDaniel[47], Karen H. Miga[8], Matthew W. Mitchell[48], Jean Monlong[8], Jacquelyn Mountcastle[22], Katherine M. Munson[10], Moses Njagi Mwaniki[49], Maria Nattestad[9], Adam M. Novak[8], Sergey Nurk[5], Hugh E. Olsen[8], Nathan D. Olson[47], Benedict Paten[8], Trevor Pesout[8], Adam M. Phillippy[5], Alice B. Popejoy[50], David Porubsky[10], Pjotr Prins[1], Daniela Puiu[51], Mikko Rautiainen[5], Allison A. Regier[7], Arang Rhie[5], Samuel Sacco[52], Ashley D. Sanders[53], Valerie A. Schneider[54], Baergen I. Schultz[23], Kishwar Shafin[9], Jonas A. Sibbesen[55], Jouni Sirén[5], Michael W. Smith[23], Heidi J. Sofia[23], Ahmad N. Abou Tayoun[56,57], Françoise Thibaud-Nissen[54], Chad Tomlinson[7], Francesca Floriana Tricomi[11], Flavia Villani[1], Mitchell R. Vollger[10,58], Justin Wagner[47], Brian Walenz[5], Ting Wang[59], Jonathan M. D. Wood[36], Aleksey V. Zimin[51,60] & Justin M. Zook[47]

[6]Division of Oncology, Department of Internal Medicine, Washington University School of Medicine, St Louis, MO, USA. [7]McDonnell Genome Institute, Washington University School of Medicine, St Louis, MO, USA. [8]UC Santa Cruz Genomics Institute, University of California, Santa Cruz, Santa Cruz, CA, USA. [9]Google, Mountain View, CA, USA. [10]Department of Genome Sciences, University of Washington School of Medicine, Seattle, WA, USA. [11]European Molecular Biology Laboratory, European Bioinformatics Institute, Cambridge, UK. [12]Department of Human Genetics, McGill University, Montreal, Quebec, Canada. [13]Canadian Center for Computational Genomics, McGill University, Montreal, Quebec, Canada. [14]Institute for the Advanced Study of Human Biology (WPI-ASHBi), Kyoto University, Kyoto, Japan. [15]Quantitative and Computational Biology, University of Southern California, Los Angeles, CA, USA. [16]Department of Data Sciences, Dana-Farber Cancer Institute, Boston, MA, USA. [17]Department of Biomedical Informatics, Harvard Medical School, Boston, MA, USA. [18]Arizona State University, Barrett and O'Connor Washington Center, Washington, DC, USA. [19]School of Biological Sciences, Washington State University, Pullman, WA, USA. [20]Institute for Medical Biometry and Bioinformatics, Medical Faculty, Heinrich Heine University Düsseldorf, Düsseldorf, Germany. [21]Howard Hughes Medical Institute, Chevy Chase, MD, USA. [22]The Vertebrate Genome Laboratory, The Rockefeller University, New York, NY, USA. [23]National Human Genome Research Institute, National Institutes of Health, Bethesda, MD, USA. [24]Center for Computational and Genomic Medicine, The Children's Hospital of Philadelphia, Philadelphia, PA, USA. [25]NNF Center for Biosustainability, Technical University of Denmark, Copenhagen, Denmark. [26]Institute for Society and Genetics, College of Letters and Science, University of California, Los Angeles, Los Angeles, CA, USA. [27]Institute for Precision Health, David Geffen School of Medicine, University of California, Los Angeles, Los Angeles, CA, USA. [28]Division of General Internal Medicine and Health Services Research, David Geffen School of Medicine, University of California, Los Angeles, Los Angeles, CA, USA. [29]Department of Biomolecular Engineering, University of California, Santa Cruz, Santa Cruz, CA, USA. [30]Dovetail Genomics, Scotts Valley, CA, USA. [31]Quantitative Life Sciences, McGill University, Montreal, Québec, Canada. [32]Department of Genetics, Yale University School of Medicine, New Haven, CT, USA. [33]Center for Genomic Health, Yale University School of Medicine, New Haven, CT, USA. [34]Quantitative Biology Center (QBiC), University of Tübingen, Tübingen, Germany. [35]Biomedical Data Science, Department of Computer Science, University of Tübingen, Tübingen, Germany. [36]Tree of Life, Wellcome Sanger Institute, Cambridge, UK. [37]Northeastern University, Boston, MA, USA. [38]The Rockefeller University, New York, NY, USA. [39]Division of Oncology, Department of Medicine, Stanford University School of Medicine, Stanford, CA, USA. [40]Institute for Genomic Health, Icahn School of Medicine at Mount Sinai, New York, NY, USA. [41]Program in Bioethics and Institute for Human Genetics, University of California, San Francisco, San Francisco, CA, USA. [42]Genome Biology Unit, European Molecular Biology LaboratoryGenome Biology Unit, Heidelberg, Germany. [43]Department of Medicine, Washington University School of Medicine, St Louis, MO, USA. [44]Computer Sciences Department, Barcelona Supercomputing Center, Barcelona, Spain. [45]Departament d'Arquitectura de Computadors i Sistemes Operatius, Universitat Autònoma de Barcelona, Barcelona, Spain. [46]Center for Digital Medicine, Heinrich Heine University Düsseldorf, Düsseldorf, Germany. [47]Material Measurement Laboratory, National Institute of Standards and Technology, Gaithersburg, MD, USA. [48]Coriell Institute for Medical Research, Camden, NJ, USA. [49]Department of Computer Science, University of Pisa, Pisa, Italy. [50]Department of Public Health Sciences, University of California, Davis, Davis, CA, USA. [51]Department of Biomedical Engineering, Johns Hopkins University, Baltimore, MD, USA. [52]Department of Ecology and Evolutionary Biology, University of California, Santa Cruz, Santa Cruz, CA, USA. [53]Berlin Institute for Medical Systems Biology, Max Delbrück Center for Molecular Medicine in the Helmholtz Association, Berlin, Germany. [54]National Center for Biotechnology Information, National Library of Medicine, National Institutes of Health, Bethesda, MD, USA. [55]Center for Health Data Science, University of Copenhagen, Copenhagen, Denmark. [56]Al Jalila Genomics Center of Excellence, Al Jalila Children's Specialty Hospital, Dubai, United Arab Emirates. [57]Center for Genomic Discovery, Mohammed Bin Rashid University of Medicine and Health Sciences, Dubai, United Arab Emirates. [58]Division of Medical Genetics, University of Washington School of Medicine, Seattle, WA, USA. [59]Department of Genetics, Washington University School of Medicine, St Louis, MO, USA. [60]Center for Computational Biology, Johns Hopkins University, Baltimore, MD, USA.

## Methods

### Genome assemblies

We analysed the 47 T2T phased diploid de novo assemblies (94 haplotypes in total) produced by the HPRC[4]. We included both T2T-CHM13 version 2[3] and GRCh38.

### Chromosome communities overview

**The homology graph.** We first used all-to-all mapping to build a reference-free model of homology relationships in the HPRCy1 pangenome. This models the full HPRCy1 as a mapping graph in which nodes are contigs and edges represent mappings between them. To build the HPRCy1 mapping graph, we generated homology mappings based on chains of 50-kb seeds of 95% average nucleotide identity—which we expect to support homologous recombination[17]—allowing up to $(n-1) = 93$ alternative mappings over any part of each contig. We first applied this process to map all 38,325 HPRCy1 contigs against all others, obtaining mappings for 38,036 of them covering the 99.9% of the total assembly sequence. This indicated that 38,036 out of 38,325 (99.2%) of the HPRCy1 assembly contigs are homologous to at least 1 other contig. Complex tangles in the assembly graphs used to build the HPRCy1 generate short contigs and tend to result in higher rates of error[4]. Thus, to simplify later analysis and focus on well-resolved regions of the assemblies, we narrowed our focus to consider only mappings involving the 16,118 contigs at least 1 Mb long, covering the 98.72% of the total assembly sequence.

We then built a graph where nodes are contigs and edges represent the mappings between them—the 'mapping graph'. Edges in this mapping graph have weights equal to the estimated sequence identity multiplied by the length of the mapping. To infer the chromosome represented by each contig, we mapped all contigs against both T2T-CHM13 and GRCh38 references and assigned them a chromosome identity based on this mapping. This mapping graph is very dense, with up to 93 mappings per contig, making it difficult to directly visualize with existing methods. To develop intuition about patterns in this graph, we instead viewed a reduced mapping graph built from the best three mappings per contig segment, labelling each contig with its reference-relative assignment (Fig. 1a). The acrocentric cluster (Fig. 1b) generally matches our prior expectations of higher similarity between chr. 13 and chr. 21, and between chr. 14 and chr. 22[18,19].

**Community detection.** To quantify the significance of these patterns, we then applied a community detection algorithm[56] to the full mapping graph. The algorithm assigns each contig to a community such that the total assignment maximizes modularity, which can be understood as the density of (weighted) links inside communities compared to links between communities. This process yielded 31 communities (Supplementary File 1). We hypothesized that each cluster represented one chromosome or chromosome arm. Around half of the chromosomes ($n = 11$) were each represented by a single community. Chromosomes 1, 2, 3, 6 and 18 were each represented in two communities corresponding to their short and long arms, likely due to frequent assembly breaks across their centromeres (Fig. 1c,d). Contigs from chromosomes X and Y fell in the same community, although the short arm of chromosome X was represented in two communities (Fig. 1d). The SAACs formed the community with the most distinct chromosomes and most contigs (1,706 contigs containing 3.91% of HPRCy1 sequence), composed of contigs belonging to the short arms of all the five acrocentric chromosomes plus chr. 21q and chr. 22q (Fig. 1c,d). chr. 13q, chr. 14q and chr. 15q each had their own community. The inclusion of the q-arms of chromosomes 21 and 22 in the community composed of p-arms contigs is likely related to their short lengths compared to chromosomes 13, 14 and 15. We obtained similar results when we increased the sensitivity of the mappings (Supplementary Fig. 29).

In the homology mapping graph of the HPRCy1, only the acrocentric and sex chromosomes form combined communities containing multiple chromosomes. The sex chromosome community reflects the PARs on X and Y[57], which are telomeric regions where these otherwise non-homologous chromosomes recombine as if they were homologues. We hypothesized that the acrocentric community might also reflect ongoing pseudo-homologous recombination

### Community detection workflow

We performed pairwise mapping for all contigs from the 47 T2T phased diploid de novo assemblies with the WFMASH sequence aligner[58] (commit ad8aeba). We set the following parameters:

```
wfmash HPRCy1.fa -s 50k -l 250k -p 95 -n 93 -Y '#' -H 0.001 -m
```

We used segment seed length of 50 kb (-s), requiring homologous regions at least ~250 kb long (-l) and estimated nucleotide identity of at least ~95% (-p). Having 94 haplotypes in total, we kept up to 93 mappings for each contig (-n). Moreover, we skipped mappings when the query and target had the same prefix before the '#' character (-Y), that is when involving the same haplotype. To properly map through repetitive regions, only the 0.001% of the most frequent kmers were ignored (-H). We skipped the base-level alignment (-m). We also generated pairwise mapping with the same parameters, but using a segment seed length of 10 kb and requiring homologous regions at least ~50 kb long.

From the resulting mappings, we excluded those involving contigs shorter than 1 Mb to reduce the possibility of spurious matches. We then used the paf2net.py Python script (delivered in the PGGB repository) to build a graph representation of the result (a mapping graph), with nodes and edges representing contigs and mappings between them, respectively.

```
python3 ~/pggb/scripts/paf2net.py -p HPRCy1.1Mbps.paf
```

The script produces a file representing the edges, a file representing the edge weights, and a file to map graph nodes to sequence names. The weight of an edge is given by the product of the length and the nucleotide identity of the corresponding mapping (higher weights were associated with longer mappings at higher identities). Finally, we used the net2communityes.py Python script (delivered in the PGGB repository) to apply the Leiden algorithm[56], implemented in the igraph tools[59], to detect the underlying communities in the mapping graph.

```
python3 ~/pggb/scripts/net2communities.py \
-e HPRCy1.1Mbps.edges.list.txt \
-w HPRCy1.1Mbps.edges.weights.txt \
-n HPRCy1.1Mbps.vertices.id2name.chr.txt --accurate-detection
```

To identify which chromosomes were represented in each community, we partitioned all contigs by mapping them against both T2T-CHM13v1.1 and GRCh38 human reference genomes with WFMASH, this time requiring homologous regions at least 150 kb long and nucleotide identity of at least 90%.

```
wfmash chm13+grch38.fa HPRCy1.fa -s 50k -l 150k -p 90 -n 1 -H 0.001 -m -N
```

We disabled the contig splitting (-N) during mapping to obtain homologous regions covering the whole contigs. For the unmapped contigs, we repeated the mapping with the same parameters, but allowing the contig splitting (without specifying -N). We labelled contigs 'p' or 'q' depending on whether they cover the short arm or the long arm of the chromosome they belonged to. Contigs fully spanning the centromeres were labelled 'pq'. We used such labels to identify the chromosome composition of the communities detected in the mapping

graph obtained without reference sequences, and to annotate the nodes in the mapping graph.

To obtain a clean visualization of the homology relationships between the HPRC assemblies, we generated a simpler mapping graph by using the same parameters used for the main graph, but keeping up to 3 mappings for each contig and adding the T2T-CHM13 reference genome version 2, which includes also the complete HG002 chromosome Y (https://www.ncbi.nlm.nih.gov/assembly/GCF_009914755.1):

```
wfmash HPRCy1+chm13v2.fa -s 50k -l 250k -p 95 -n 3 -Y '#' -H
0.001 -m -w 5000
```

We set window size for sketching equal to 5000 (-w) to reduce the runtime by sampling fewer kmers. We used the paf2net.py Python script to build the mapping graph and then used Gephi[60] (version 0.9.4) to visualize it. We computed the mapping graph layout by running 'Random Layout' and then the 'Yifan Hu' algorithm.

### Pangenome graph building
For each of the 47 T2T phased diploid de novo assemblies, we mapped all contigs against the T2T-CHM13 human reference genome with the WFMASH sequence aligner (commit ad8aeba). For the HG002 sample, we included two assemblies: the HG002-HPRCy1 phased diploid de novo assembly (built with HiFi reads) and a phased diploid de novo assembly based on both HiFi and ONT reads, built with the Verkko assembler. We set the following parameters:

```
wfmash chm13.fa assembly.fa -s 50k -l 150k -p 90 -n 1 -H 0.001 -m
```

We used segment seed length of 50 kb (-s), requiring homologous regions at least ~150 kb long (-l) and estimated nucleotide identity of at least ~90% (-p). We kept only one mapping (the best one) for each contig (-n). To properly map through repetitive regions, only the 0.001% of the most frequent kmers were ignored (-H). We skipped the base-level alignment (-m). For the HG002-HPRCy1 contigs, we disabled the contig splitting (-N).

Then, we identified contigs originating from acrocentric chromosomes and covering both the short and long arms of the chromosome they belonged to. We considered only contigs with mappings at least 1 kb long on both arms and at least 1 Mb away from the centromere. We call such contigs 'p–q acrocentric contigs'. For HG002-HPRCy1, only contigs longer than or equal to 300 kb were considered, regardless of covering both arms of the belonging chromosomes.

Finally, we built a pangenome graph with all the p–q acrocentric contigs and both T2T-CHM13 and GRCh38 human reference genomes by applying PGGB[22] (commit a4a6668). We set the following parameters:

```
pggb -i contigs.fa.gz -s 50k -l 250k -p 98 -n 162 -F 0.001 -k 311 -G
13117,13219 -O 0.03
```

We used segment seed length of 50 kb (-s), requiring homologous regions at least ~250 kb long (-l) and estimated nucleotide identity of at least ~98% (-p). Having 142 p–q acrocentric contigs in input (132 from HG002-HPRCy1 and 10 from HG002-Verkko) plus 10 acrocentric chromosomes from the T2T-CHM13 and GRCh38 reference genomes plus 49 HG002-HPRCy1 contigs representing other 10 acrocentric haplotypes (5 maternal and 5 paternal), we kept up to 162 mappings (142 + 10 + 10) for each contig (-n). To properly map through repetitive regions, only the 0.001% of the most frequent kmers were ignored (-F). We filtered out alignment matches shorter than 311 bp to remove possible spurious relationships caused by short repeated homologies (-k). We set big target sequence lengths and a small sequence padding for two rounds of graph normalization (-G and -O). To visualize the acrocentric pangenome graph, we built the graph layout with ODGI LAYOUT[26] (commit e2de6cb) and visualized with GFAESTUS[61] (commit 50fe37a). This renders sequences and chains of small variants as linear structures, while repeats caused by segmental duplications, inversions and other structural variants tend to form loops.

### Pangenome graph untangling
We untangled the pangenome graph by applying ODGI UNTANGLE (commit e2de6cb). Practically, we projected the graph into an alignment between a set of query (HPRCy1 contigs) and reference (T2T-CHM13) sequences. We set the following parameter:

```
odgi untangle -i graph.og -e 50000 -m 1000 -n 100 -j 0 -R targets.
txt -d cuts.txt
```

We segmented the graph into regular-sized regions of ~50 kb (-e), merging regions shorter than 1 kb (-m). We reported up to the 100th best target mapping for each query segment (-n), not applying any threshold for the Jaccard similarity (-j). We used all paths in the graphs as queries and projected them against the five acrocentric chromosomes of the T2T-CHM13 genome (-R). Moreover, we emit the cut points used to segment the graph (-d).

For each query segment, if there were multiple best hits against different targets (that is, hits with the same, highest Jaccard similarity), we put as the first one the hit having as target the chromosome of origin of the query (obtained from the chromosome partitioning of the contigs).

We repeated the graph untangling another five times, but constrained the algorithm to use only one of the acrocentric chromosomes of T2T-CHM13 as a target at a time (-r) and return the best-matching hit (-n).

```
odgi untangle -i graph.og -e 50000 -m 1000 -n 1 -j 0 -r chr13 -c
cuts.txt
odgi untangle -i graph.og -e 50000 -m 1000 -n 1 -j 0 -r chr14 -c
cuts.txt
odgi untangle -i graph.og -e 50000 -m 1000 -n 1 -j 0 -r chr15 -c
cuts.txt
odgi untangle -i graph.og -e 50000 -m 1000 -n 1 -j 0 -r chr21 -c
cuts.txt
odgi untangle -i graph.og -e 50000 -m 1000 -n 1 -j 0 -r chr22 -c
cuts.txt
```

We used the cut points generated when using all of the acrocentric chromosomes of T2T-CHM13 as targets (-c). In this way, all untangling runs (six in total) used the same cut points for the segment boundaries.

Finally, we 'grounded' the untangled output generated with all acrocentric chromosomes as targets: in more detail, each untangled query segment was placed against a particular acrocentric chromosome (not only the best-matching one) by using the untangled outputs constrained to a single target. We split the result by acrocentric chromosome and kept only queries untangling both p- and q-arms of the targets. Furthermore, we removed query segments overlapping regions flagged in the assemblies as unreliable (that is, having coverage issues) by FLAGGER[4]. FLAGGER is a HiFi read-based pipeline that detects different types of mis-assemblies within a phased diploid assembly by identifying read-mapping coverage inconsistencies across the maternal and paternal haplotypes. To focus on the more similar query-target hits, we used the Jaccard metric to estimate the sequence identity by applying the corrected formula reported in ref. 62, and retained only results at greater than 90% estimated identity. To analyse the orientation status of HPRCy1 contigs in the segmental duplication centred on the SST1 array, we generated a new pangenome graph with ODGI FLIP (commit 0b21b35). In more detail, we first flipped paths around if they tend to be in the reverse complement orientation relative to the pangenome graph. This leads to having a uniform orientation for the HPRCy1 contigs, all in forward orientation with respect to the graph. Then, we untangled the flipped graph in the same

way as described above. We displayed the untangling results for each acrocentric chromosome with the R development environment (version 3.6.3), equipped with the following packages: tidyverse (version 1.3.0), RColorBrewer (version 1.1.2), ggplot2 (version 3.3.3) and ggrepel (version 0.9.1).

## Recombination pattern analysis

**Aggregating best-hit untangle results.** For each group of HPRCy1-acro contigs anchored to a T2T-CHM13 acrocentric q-arm, we counted the number of contigs having as best-hit each one of the acrocentrics. In particular, for each base position of the T2T-CHM13 acrocentric chromosome of each group, we quantified how many times each of the acrocentrics appeared as best-hit in the pangenome graph untangling. We considered only best hits with an estimated identity of at least 90%.

**Regional homology entropy.** To quantify the degree of disorder in the untangling result, we calculated the diversity entropy across the different acrocentrics that were present as best-hit. In more detail, we projected each HPRCy1 acrocentric p–q contig against the T2T-CHM13 acrocentric to which it is anchored via the q-arm and associated each reference base position to the corresponding acrocentric best-hit. We considered only best hits with an estimated identity of at least 90%. Then, we computed the Shannon diversity index (SDI) in windows 50 kb long over the contigs. We used −1 as missing SDI value in the regions where the contigs do not match any targets. For each group of contigs, we aggregated the SDI results by computing their average (ignoring the missing SDI values) for each reference base position. We call this metric the positional homology entropy, and it serves to show regions where contigs can be described as mosaics of different reference chromosomes. However, it cannot distinguish regions where there are different orders of reference chromosome similarity—that might be indicative of recombination exchange—from those where there is regional diversity in each contig's relationship to T2T-CHM13. The latter case could occur if T2T-CHM13 itself contains rare recombinations between acrocentrics, or where ancient homology might result in 'noise' in untangling alignments as contigs pick from two equally good alternative mappings. To avoid these pitfalls and establish a more stringent graph-space recombination metric, we then extended the untangling diversity metric to operate on multiple mappings.

**Positional homology entropy.** To take into account the other hits in addition to the first one, including their order, we generalized the diversity entropy metric to work over orders of the top 5 untangling hits and consider all contigs jointly. For each reference segment, we collected the corresponding best 5 untangling hits for each of the HPRCy1 acrocentric p–q contig; this is possible because the reference segments are stable across all contigs. We considered only best hits with an estimated identity of at least 90%. To avoid driving the untangle entropy by intra-chromosomal similarity caused by segmental duplications modelled in the structure of the PVG (as seen in loops on chr. 13q, chr. 15q and chr. 22q; Fig. 2), we ignored consecutive duplicate target hits—in other words, we took the ordered set of unique reference targets. When multiple contig segments were grounded against the same reference segment, we considered the first contig segment having the best grounding, that is having the highest estimated identity when placed against the current reference segment. Then, we ranked the five best hits by estimated similarity. Finally, for each reference segment, we computed the SDI across all available five best hits orders. We used −1 as missing SDI value in the reference regions without any contig matches. We kept in the output also the information about how many HPRCy1 acrocentric p–q contigs contributed to the entropy computation in each reference segment. This yielded the positional homology entropy.

**PHR derivation.** To obtain the PHRs, we aggregated the final results by considering regions with positional homology entropy greater than 0 and supported by at least 1 contig, merging with BEDtools[63] those that were less than 30 kb away, and removing merged regions shorter than 30 kb.

**Display of untangle mosaics.** We displayed the aggregated results for each acrocentric chromosome. We used genome annotations for the first 25 Mb of each acrocentric chromosome, using T2T-CHM13v2.0 UCSC trackhub (https://genome.ucsc.edu/cgi-bin/hgTracks?db=hub_3671779_hs1). We made the figures with the scripts available at https://github.com/pangenome/chromosome_communities/tree/main/scripts. To plot the figures, we used R (version 3.6.3), equipped with the following packages: tidyverse (version 1.3.0), RColorBrewer (version 1.1.2), ggplot2 (version 3.3.3) and ggrepel (version 0.9.1). Finally, we used Inkscape (https://inkscape.org/) to compose main text figures based on the results, and provide supplementary figures directly produced by these methods.

**HPRCy1 SNP density plots.** We displayed biallelic SNP density in the full HPRCy1 draft pangenome built with PGGB[4] versus both GRCh38 and T2T-CHM13. To do so, we extracted biallelic SNPs from the released VCF files versus each chromosome, for both references (get_bisnp.sh). Because T2T-CHM13 version 1.1, which was used in HPRCy1, does not have a Y chromosome, we used that of GRCh38, which includes masked PAR1 and PAR2 regions. We displayed biallelic SNP density in bins of 100 kb, using R (version 4.1.1) and tidyverse (version 1.3.1) package (plot_bisnp_dens.R).

**ROB breakpoints.** We mapped BAC clones from ref. 8 against the T2T-CHM13 human reference genome with the WFMASH sequence aligner (commit ad8aeba). We kept only mappings covering acrocentric chromosomes and with an estimated identity of at least 90%. To plot the figures, we used R (version 3.6.3), equipped with the following packages: tidyverse (version 1.3.0), RColorBrewer (version 1.1.2), ggplot2 (version 3.3.3) and ggrepel (version 0.9.1). We coloured BAC clones' mappings according to ref. 8.

**Maximum likelihood phylogenetic analysis.** We conducted the phylogenetic analysis by using the maximum likelihood method based on the best-fit substitution model (Kimura 2-parameter +G, parameter = 5.5047) inferred by Jmodeltest2[64] with 1,000 bootstrap replicates. Bootstrap values higher than 75 are indicated at the base of each node.

**Recombination hotspots analysis.** We obtained the human PRDM9 binding motifs (17 in total) from ref. 41 and used FIMO[65] to scan their occurrences in T2T-CHM13v2.0 human reference genome:

```
fimo --thresh 1.0E-4 PRDM9_motifs.human.txt chm13v2.fa
```

FIMO computes a log-likelihood ratio score for each motif with respect to each sequence position and converts these scores to *P* values using dynamic programming (assuming a zero-order null model in which sequences are generated at random with user-specified per-letter background frequencies) and then estimate false discovery rates[65]. Each motif is associated with a measure of how likely it represents a true binding target for PRDM9. We retained for downstream analyses only motifs for which such a measure is at least 70% (14 of 17). For each motif, we counted the number of occurrences present in windows 20 kb long across each T2T-CHM13v2.0 chromosome by using BEDtools[63].

```
bedtools intersect -a chm13v2.windows_20kbp.bed -b fimo_output.motif$i.bed
```

To plot the figures, we used R (version 3.6.3), equipped with the following packages: tidyverse (version 1.3.0), RColorBrewer (version 1.1.2), ggplot2 (version 3.3.3) and ggrepel (version 0.9.1).

## Linkage disequilibrium analysis

We identified variants embedded in the pangenome graph by using VG DECONSTRUCT[5]:

```
vg deconstruct -P chm13 -H '?' --ploidy 1 -e -a graph.gfa > variants.vcf
```

We called variants with respect to the T2T-CHM13 reference genome (-P), reporting variants for each HPRCy1 acrocentric p–q contig (-H and --ploidy). We considered only traversals that correspond to paths (that is, contigs) in the graph (-e) and also reported nested variation (-a). From the variant set, we considered only single nucleotide variants. We estimated linkage disequilibrium between pairs of markers within 70 kb by using PLINK v1.9[66] upon specification of haploid sets and retaining all values of $r^2 > 0$ (plot_ld_1.R). Finally, we generated binned linkage disequilibrium decay plots with confidence intervals using R (version 3.6.3), focusing on pairs less than 4 kb apart.

## Validating homology mosaicism

**HG002-Verkko assembly.** We applied an earlier version of Verkko (beta 1, commit vd3f0b941b5facf5807c303b0c0171202d83b7c74) to build a diploid assembly graph for the HG002 cell line using the HiFi (105x) and ONT (85x) reads as described[67]. The resulting assembly graph resolves the proximal junction in single contigs for each haplotype up to multi-mega bases, while the distal junctions remain to be resolved. We used homopolymer compressed markers from the parental Illumina reads to assign unitigs to maternal, paternal haplotype or ambiguous when not enough markers supported either haplotype. For estimating the number of times a unitig has to be visited, we aligned HiFi and ONT reads to the assembly graph using GraphAligner with the following parameters: –seeds-mxm-length 30 –seeds-mem-count 10000 -b 15 –multimap-score-fraction 0.99 –precise-clipping 0.85 –min-alignment-score 5000 –clip-ambiguous-ends 100 –overlap-incompatible-cutoff 0.15 –max-trace-count 5 –hpc-collapse-reads –discard-cigar[68]. Four distal junctions were connected to the rDNA arrays with ambiguous nodes connecting the maternal and paternal nodes, supporting they belong to the same chromosome. Two distal junction unitigs, one maternal and paternal, were disconnected from each other but connected to the rDNA arrays, which were assigned to the same chromosome. Using the marker and ONT alignments, we identified paths in the graph and assigned them according to the most supported haplotype. If only ambiguous nodes were present between the haplotype assigned unitigs, with no ONT reads to resolve the path, nodes were randomly assigned to one haplotype to build the contig. After all paths were identified, we produced the consensus using verkko --assembly <path-to-original-assembly> --paths <path-to-paths>. The entire procedure to produce parental markers, tagging unitigs according to its haplotype on the assembly graph and finding the path using ONT reads is now available in the latest Verkko (v1) in a more automated way.

**Untangling validation.** To provide cross-validation of the HiFi contig assemblies and our analysis of them, we compared the untangling of two assemblies of the same sample (HG002). One was made with the HPRCy1 pipeline, while the other used the Verkko diploid T2T assembler. Verkko employs ONT to untangle ambiguous regions in a HiFi-based assembly graph, automating techniques first developed in production of T2T-CHM13. Verkko's assembly aggregates information from ONT, thus providing a single integrated target for cross-validation of our analysis using an alternative sequencing and assembly approach.

We validated the results of the pangenome untangling by comparing the best hits of the two HG002 assemblies, the one built with HiFi reads and the other based on both HiFi and ONT reads, built with the Verkko assembler[67]. For each base position of each T2T-CHM13 acrocentric chromosome, we compared the untangling best-hit of the HG002-HPRCy1 contigs with the best-hit supported by HG002-Verkko contigs. We considered only best hits with an estimated identity of at least 90%. We defined reference regions as concordant when both HG002 assemblies supported the same T2T-CHM13 acrocentric as best-hit. We treated the two haplotypes (maternal and parental) separately.

We observe a high degree of concordance between the two methods at a level of the chromosome homology mosaicism plots. The best-hit untangling shows similar patterns in the HG002-Verkko assembly as those seen in HG002-HPRCy1 (Supplementary Figs. 9–13). However, some SAAC haplotypes appear to be poorly assembled in HG002-HPRCy1. On the q-arms we measured 99.93% concordance between HG002-HPRCy1 and HG002-Verkko untangling results, but only 87.45% concordance on the p-arms (Supplementary File 8). This lower level is consistent with greater difficulty in assembling the SAACs due to their multiple duplicated sequences (including the PHRs), satellite arrays and the rDNA. We found the discordance was driven by a single chromosome haplotype: while most p-arms achieve around 90% concordance, HG002-HPRCy1 14p-maternal exhibits a high degree of discordance in the assemblies (66.19% concordance) (Supplementary Fig. 19). Although this $n = 1$ validation focuses on only 10 haplotypes, it incorporates many independent reads provided by deep HiFi (105x) and ONT (85x) data used in HG002-Verkko. We thus have compared the concordance between structures observed in single molecule reads across the SAACs and the HG002-HPRCy1 assembly that represents the HiFi-only assembly process that produced our pangenome.

However, this analysis should be seen as presenting a lower bound on our process accuracy. We are considering *all* HG002-HPRCy1 contigs that map to the acrocentrics—not only those that would meet our centromere-crossing requirement, and HG002-HPRCy1 is itself more fragmented than the other assemblies which we have selected for the acro-PVG[4]. The fragmented nature of its contigs (only one from chr. 22 meets our p–q mapping requirement) may introduce additional disagreements with HG002-Verkko. The overall result indicates that most patterns observed in HiFi-only assemblies are likely to be supported by an automated near-T2T assembly of the same sample.

## Reporting summary

Further information on research design is available in the Nature Portfolio Reporting Summary linked to this article.

## Data availability

Assemblies produced by the HPRC are available at AnVIL (https://anvil-project.org/), in the AnVIL_HPRC workspace. Data are also available as part of the AWS Open Data Program (https://registry.opendata.aws/) in the human-pangenomics S3 bucket (https://s3-us-west-2.amazonaws.com/human-pangenomics/index.html). In addition, the data have been uploaded to the International Nucleotide Sequence Database Collaboration (INSDC) for long-term storage and availability. Supporting information about the data (including index files with S3 and GCP file locations) can be found at the following GitHub repository: https://github.com/human-pangenomics/HPP_Year1_Assemblies. All supplementary files, including the PVG and its layout, are available on Zenodo at https://doi.org/10.5281/zenodo.7692554.

## Code availability

Code and links to methods and tools used to perform all the analyses and produce all the figures can be found on Zenodo at https://doi.org/10.5281/zenodo.7697614.

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

**Acknowledgements** Our work depends on the HPRC draft human pangenome resource established in the accompanying Article[4], and we thank the production and assembly groups for their efforts in establishing this resource. This work used the computational resources of the UTHSC Octopus cluster and NIH HPC Biowulf cluster. We acknowledge support in maintaining these systems that was critical to our analyses. The authors thank M. Miller for the development of a graphical synopsis of our study (Fig. 5); and R. Williams and N. Soranzo for support and guidance in the design and discussion of our work. This work was supported, in part, by National Institutes of Health/NIDA U01DA047638 (E.G.), National Institutes of Health/NIGMS R01GM123489 (E.G.), NSF PPoSS Award no. 2118709 (E.G. and C.F.), the Tennessee Governor's Chairs programme (C.F. and E.G.), National Institutes of Health/NCI R01CA266339 (T.P., L.G.d.L. and J.L.G.), and the Intramural Research Program of the National Human Genome Research Institute, National Institutes of Health (A.R., S.K. and A.M.P.). We acknowledge support from Human Technopole (A.G.), Consiglio Nazionale delle Ricerche, Italy (S.B. and V.C.), and Stowers Institute for Medical Research (T.P., L.G.d.L., B.R. and J.L.G.).

**Author contributions** Paper writing: A.G. and E.G. Paper editing: A.G., S.B., L.G.d.L., T.P., A.R., S.K., B.R., C.F., J.L.G., A.M.P., V.C. and E.G. Development of algorithms and software: A.G. and E.G. Chromosome community detection: A.G. and E.G. Pangenome graph building and analyses: A.G. and E.G. Pangenome visualization: A.G., C.F. and E.G. ROB breakpoints: A.G., T.P., J.L.G. Maximum likelihood phylogenetic analysis: L.G.d.L. Recombination hotspots analysis: A.G. Linkage disequilibrium analysis: A.G., S.B. and V.C. Homology mosaicism analysis: A.G. and E.G. Physical distance modelling: B.R. and J.L.G. HG002-Verkko assembly: A.R., S.K. and A.M.P. Untangling validation: A.G. and E.G.

**Competing interests** The authors declare no competing interests.

**Additional information**
**Correspondence and requests for materials** should be addressed to Erik Garrison.

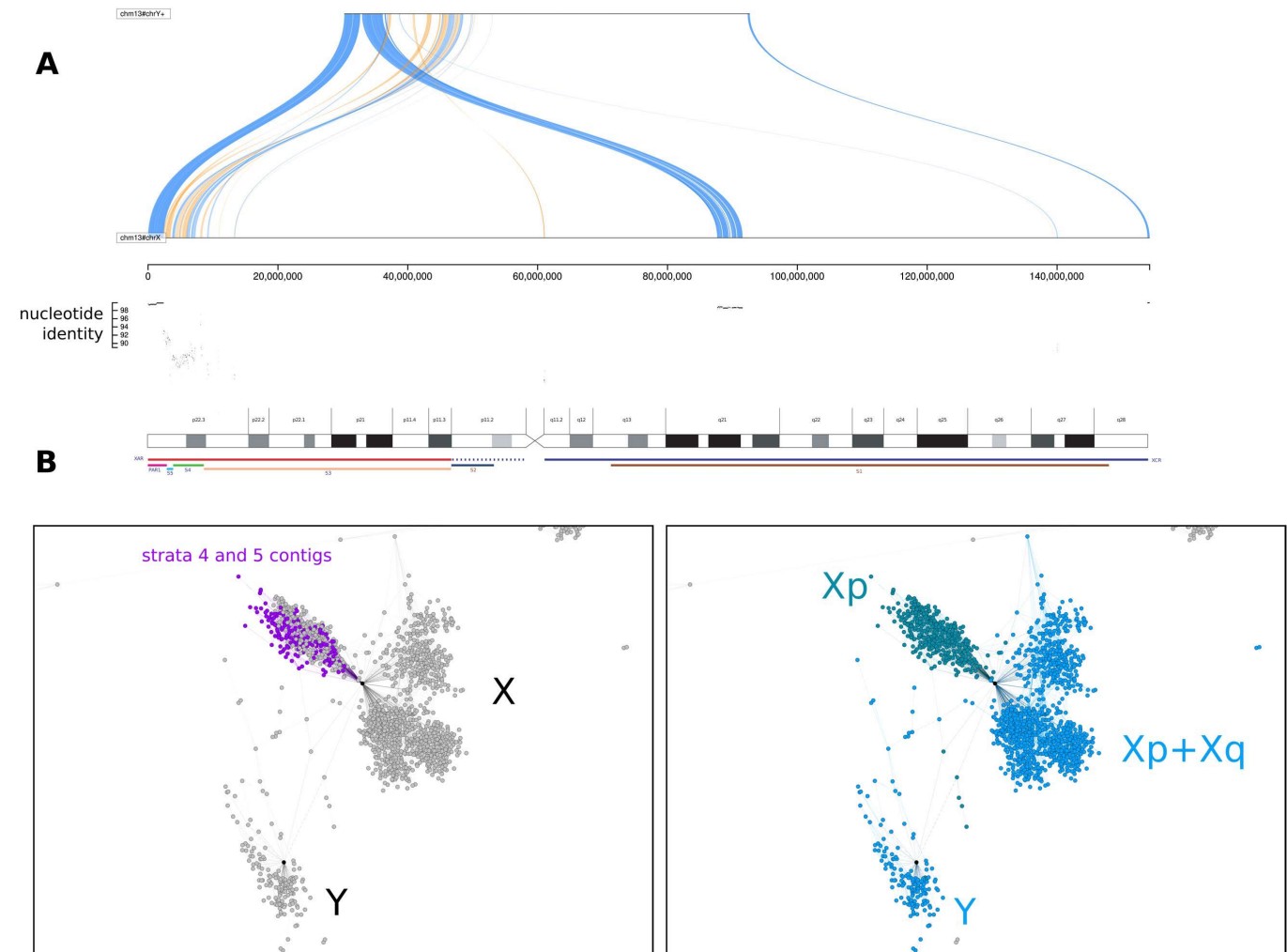

**Extended Data Fig. 1 | (A) Evolutionary strata 5 and 4.** Visualization with Saffire (https://mrvollger.github.io/SafFire/) of the alignment between T2T-CHM13 X and Y reveals that strata 5 and 4 feature low identity (~90%), numerous inversions, and some rearrangements; (**B**) X chromosome ideogram according to[21]. On the bottom, its evolutionary domains: the X-added region (XAR), the X-conserved region (XCR; dotted region in proximal Xp does not appear to be part of the XCR), the pseudoautosomal region PAR1, and evolutionary strata S5–S1. (**C**) The reduced all-to-all mapping graph of HPRCy1 versus itself, with contigs represented as nodes and mappings as edges. In red contigs covering the evolutionary strata 5 and 4 on chromosome X; (**D**) Coloring the reduced homology mapping graph in C with community assignments. Panels **C** and **D** use the same layout as Fig. 1 but focus only on the X and Y region of the visualization.

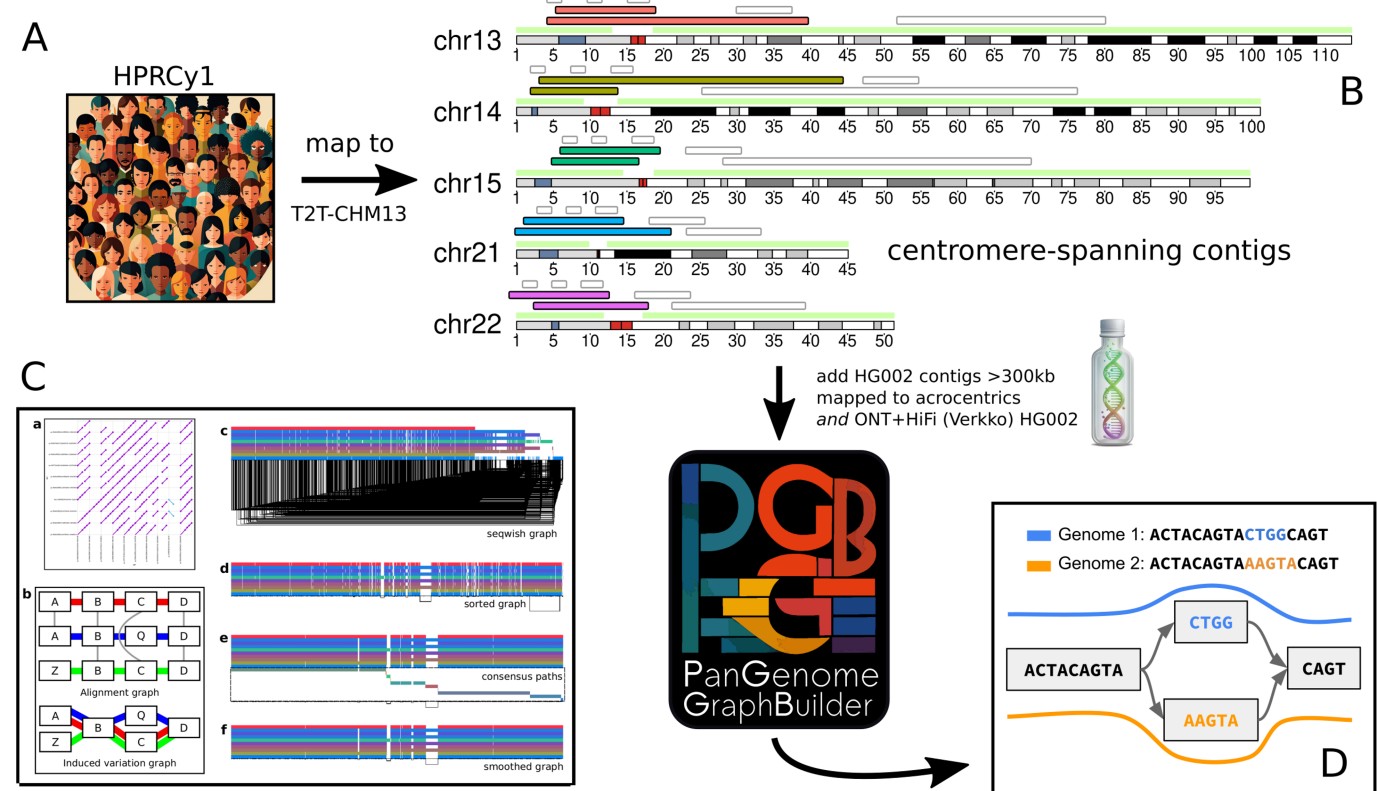

**Extended Data Fig. 2 | An overview of our approach to build a PVG for HPRCy1 contigs that can be anchored to a specific acrocentric q-arm.**
(**A**) As input, we take the entire HPRCy1 and map it to T2T-CHM13. (**B**) This yields mappings to acrocentric chromosomes, which we filter to select contigs that map across the centromeres (red cytobands) between non-centromeric regions (over-labeled green). We include two HG002 assemblies based on standard HiFi (from HPRCy1) and on both HiFi and ONT data (from Verkko). (**C**) We then apply PGGB to build a PVG from the HPRCy1-acro collection. PGGB first obtains an all-to-all alignment of the input (C.a.), which is converted to a variation graph with SEQWISH[28] (C.b.), then normalized with sorting and multiple sequence alignment steps in SMOOTHXG (C.c-f). (**D**) The resulting PVG expresses genomes as paths, or walks, through a common sequence graph. This model thus contains all input sequences and their relative alignments to all others—in the example we see a CTGG/AAGTA block substitution between genomes 1 and 2.

# *Untangling* extracts pairwise alignments from variation graphs.

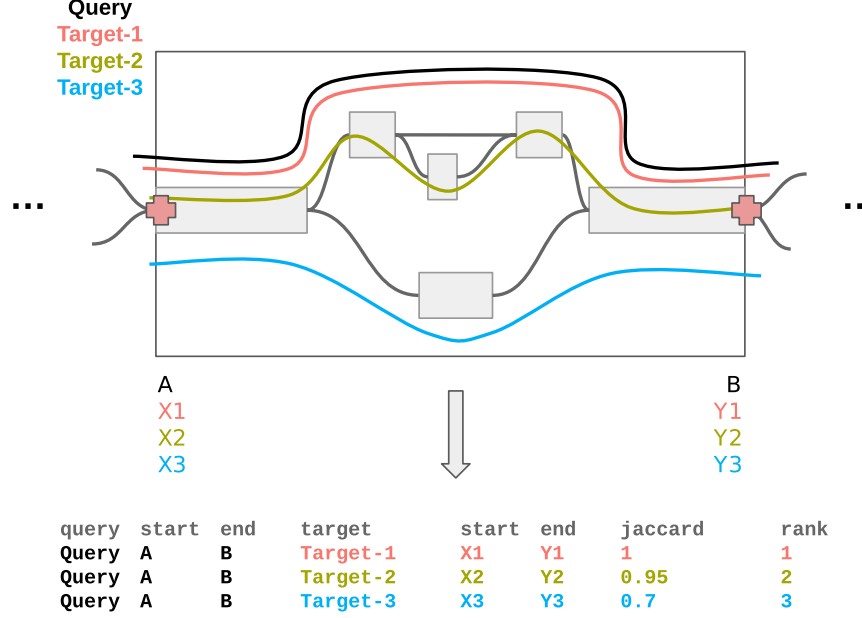

| query | start | end | target | start | end | jaccard | rank |
|-------|-------|-----|--------|-------|-----|---------|------|
| Query | A | B | Target-1 | X1 | Y1 | 1 | 1 |
| Query | A | B | Target-2 | X2 | Y2 | 0.95 | 2 |
| Query | A | B | Target-3 | X3 | Y3 | 0.7 | 3 |

Identify cut points (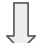) in the graph

⬇

Define segment boundaries:

[A,B), [X1, Y1), [X2, Y2), [X3, Y3)

⬇

Compare segments with jaccard in sequence space

**Extended Data Fig. 3 | Scheme of the graph untangling.** We applied ODGI UNTANGLE to obtain a mapping from segments of all PVG paths onto T2T-CHM13. The segmentation cuts the graph into regular-sized regions whose boundaries occur at structural variant breakpoints. For each query subpath through a graph segment, we use a Jaccard metric over the sequence space of the subpaths to find the best-matching reference segment.

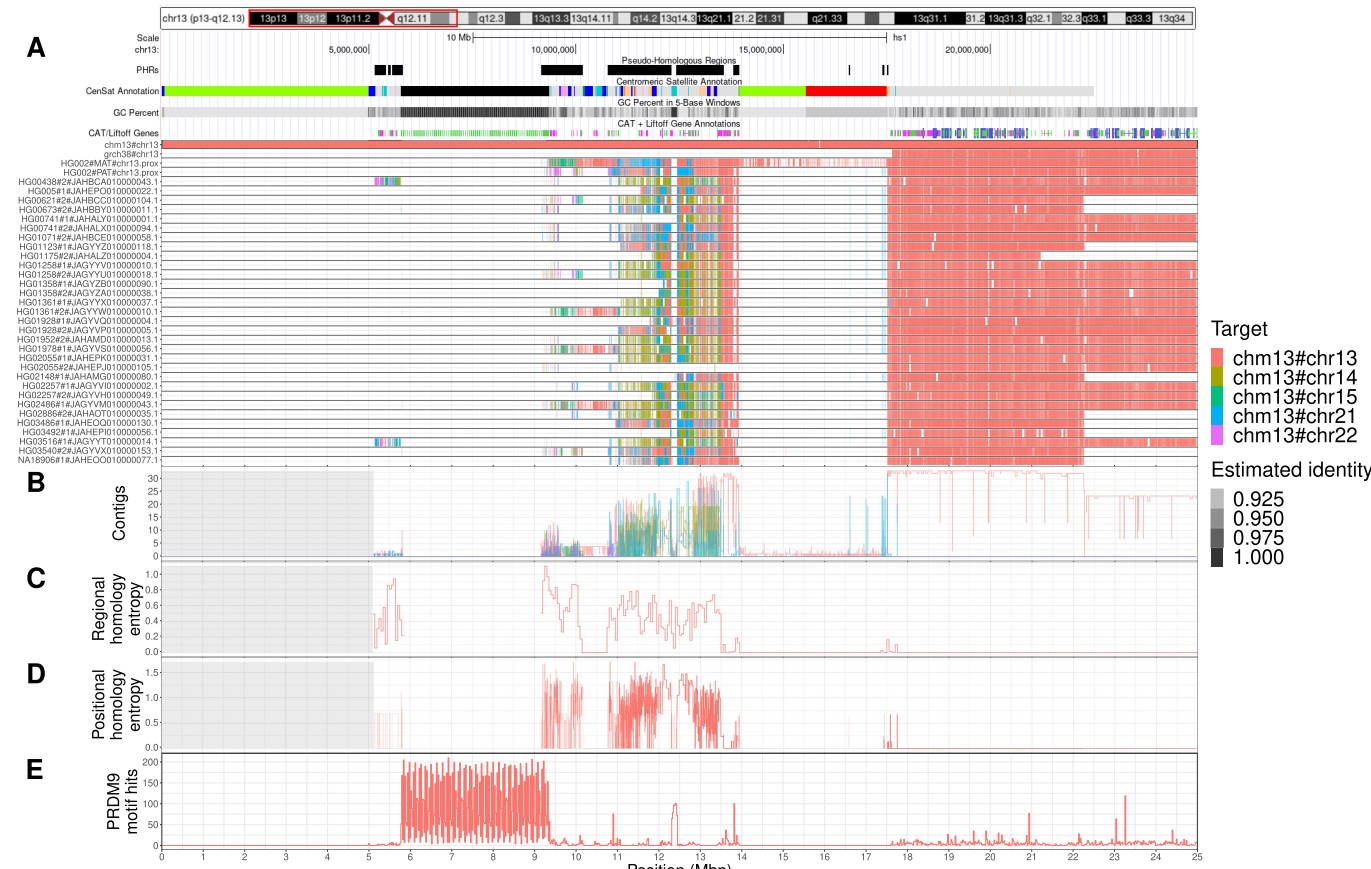

**Extended Data Fig. 4 | Characteristics of the pseudo-homologous regions of acrocentric chromosomes on chromosome 13. (A)** We focus on the first 25 Mbp of chromosome 13, shown here as a red box over T2T-CHM13 cytobands. Pseudo-homologous regions (**PHRs**), where diverse sets of acrocentric chromosomes recombine, are highlighted relative to T2T-CHM13 genome annotations for repeats, GC percentage, and genes. Above, we indicate regions of interest described in the main text: rDNA, SST1 array, centromere, and q-arm. Below, we show T2T-CHM13-relative homology mosaics for each chromosome 13 matched contig from HPRCy1-acro, with the most-similar reference chromosome at each region shown using the given colors (**Target**). **(B)** Aggregated untangle results in the SAACs. For each acrocentric chromosome, we show the count of its HPRCy1 q-arm-anchored contigs mapping itself and all other acrocentrics (**Contigs**), **(C)** as well as the regional (50kbp) untangle entropy metric (**Regional homology entropy**) computed over the contigs' T2T-CHM13-relative untanglings. **(D)** By considering the multiple untangling of each HPRCy1-acro contig, we develop a point-wise metric that captures diversity in T2T-CHM13-relative homology patterns (**Positional homology entropy**), leading to our definition of the PHRs. **(E)** The patterns of homology mosaicism suggest ongoing recombination exchange in the SAACs. A scan over T2T-CHM13 reveals that the rDNA units are enriched for PRDM9 binding motifs, and thus may host frequent double stranded breaks during meiosis. In (B-D) a gray background indicates regions with missing data due to the lack of non-T2T-CHM13 contigs. We provide the Centromeric Satellite Annotation (CenSat Annotation) track legend in Extended Data Table 1.

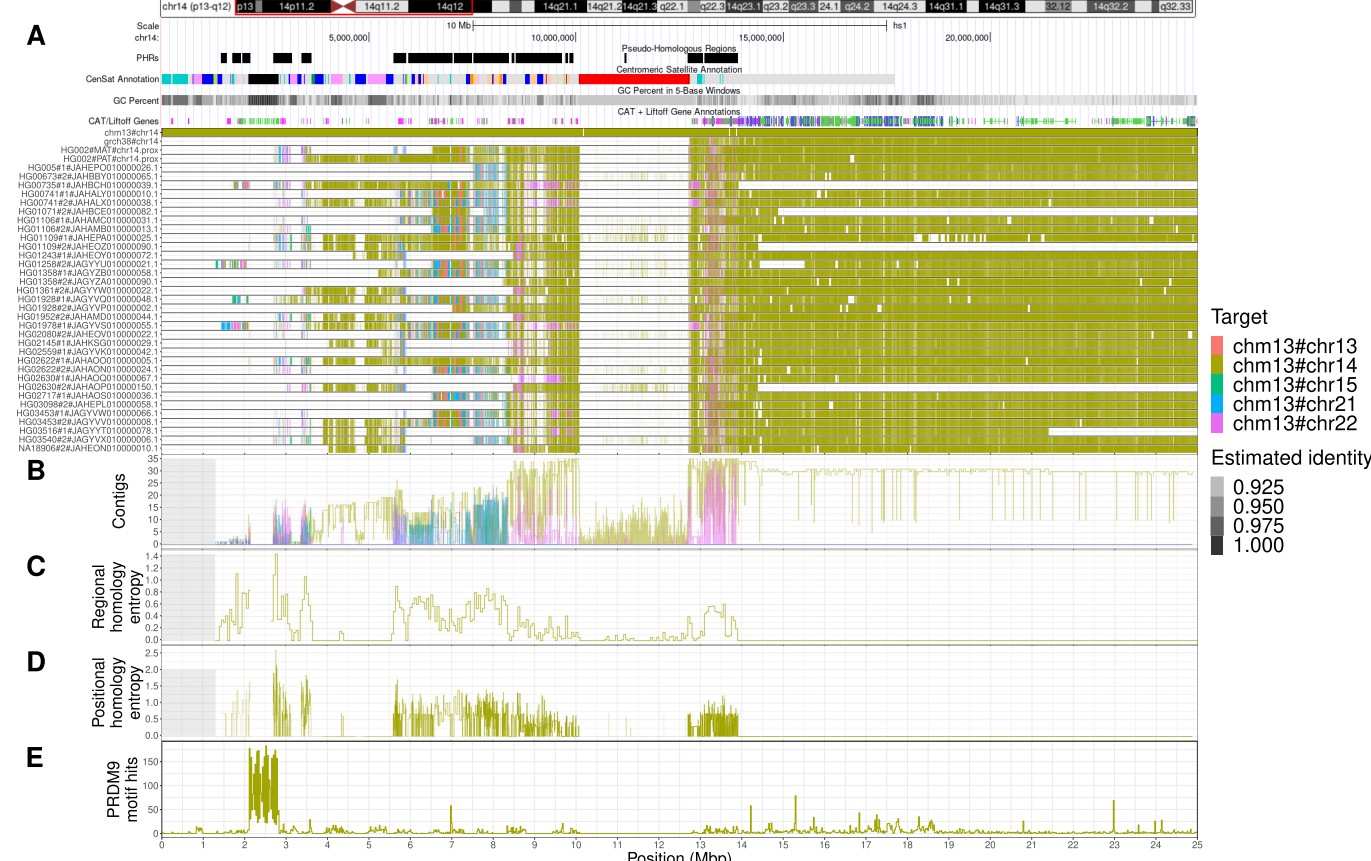

**Extended Data Fig. 5 | Characteristics of the pseudo-homologous regions of acrocentric chromosomes on chromosome 14. (A)** We focus on the first 25 Mbp of chromosome 14, shown here as a red box over T2T-CHM13 cytobands. Pseudo-homologous regions (**PHRs**), where diverse sets of acrocentric chromosomes recombine, are highlighted relative to T2T-CHM13 genome annotations for repeats, GC percentage, and genes. Above, we indicate regions of interest described in the main text: rDNA, SST1 array, centromere, and q-arm. Below, we show T2T-CHM13-relative homology mosaics for each chromosome 13 matched contig from HPRCy1-acro, with the most-similar reference chromosome at each region shown using the given colors (**Target**). **(B)** Aggregated untangle results in the SAACs. For each acrocentric chromosome, we show the count of its HPRCy1 q-arm-anchored contigs mapping itself and all other acrocentrics (**Contigs**), **(C)** as well as the regional (50kbp) untangle entropy metric (**Regional homology entropy**) computed over the contigs' T2T-CHM13-relative untanglings. **(D)** By considering the multiple untangling of each HPRCy1-acro contig, we develop a point-wise metric that captures diversity in T2T-CHM13-relative homology patterns (**Positional homology entropy**), leading to our definition of the PHRs. **(E)** The patterns of homology mosaicism suggest ongoing recombination exchange in the SAACs. A scan over T2T-CHM13 reveals that the rDNA units are enriched for PRDM9 binding motifs, and thus may host frequent double stranded breaks during meiosis. In (B-D) a gray background indicates regions with missing data due to the lack of non-T2T-CHM13 contigs. We provide the Centromeric Satellite Annotation (CenSat Annotation) track legend in Extended Data Table 1.

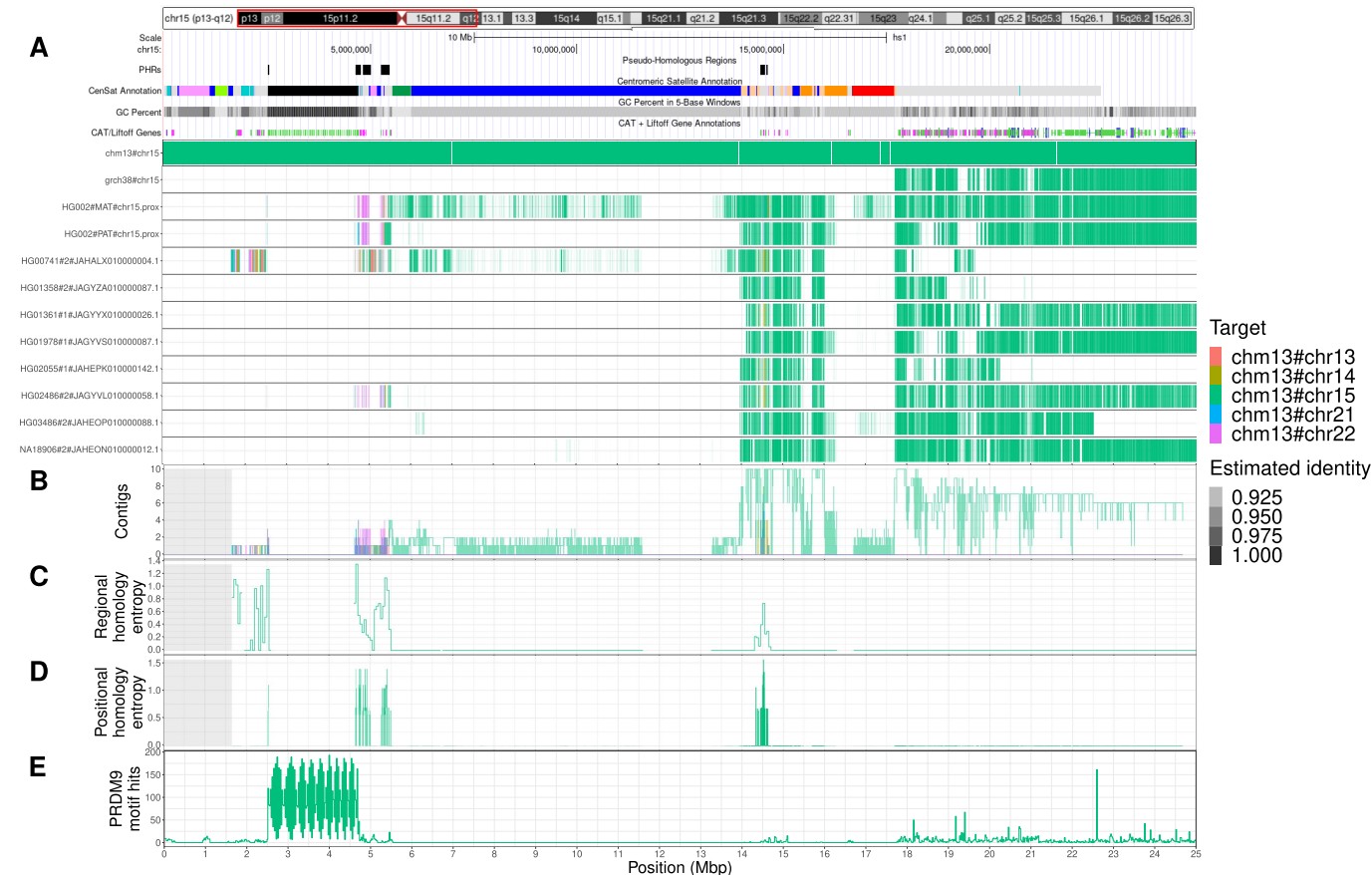

**Extended Data Fig. 6 | Characteristics of the pseudo-homologous regions of acrocentric chromosomes on chromosome 15. (A)** We focus on the first 25 Mbp of chromosome 15, shown here as a red box over T2T-CHM13 cytobands. Pseudo-homologous regions (**PHRs**), where diverse sets of acrocentric chromosomes recombine, are highlighted relative to T2T-CHM13 genome annotations for repeats, GC percentage, and genes. Above, we indicate regions of interest described in the main text: rDNA, SST1 array, centromere, and q-arm. Below, we show T2T-CHM13-relative homology mosaics for each chromosome 13 matched contig from HPRCy1-acro, with the most-similar reference chromosome at each region shown using the given colors (**Target**). **(B)** Aggregated untangle results in the SAACs. For each acrocentric chromosome, we show the count of its HPRCy1 q-arm-anchored contigs mapping itself and all other acrocentrics (**Contigs**), **(C)** as well as the regional (50kbp) untangle entropy metric (**Regional homology entropy**) computed over the contigs' T2T-CHM13-relative untanglings. **(D)** By considering the multiple untangling of each HPRCy1-acro contig, we develop a point-wise metric that captures diversity in T2T-CHM13-relative homology patterns (**Positional homology entropy**), leading to our definition of the PHRs. **(E)** The patterns of homology mosaicism suggest ongoing recombination exchange in the SAACs. A scan over T2T-CHM13 reveals that the rDNA units are enriched for PRDM9 binding motifs, and thus may host frequent double stranded breaks during meiosis. In (B-D) a gray background indicates regions with missing data due to the lack of non-T2T-CHM13 contigs. We provide the Centromeric Satellite Annotation (CenSat Annotation) track legend in Extended Data Table 1.

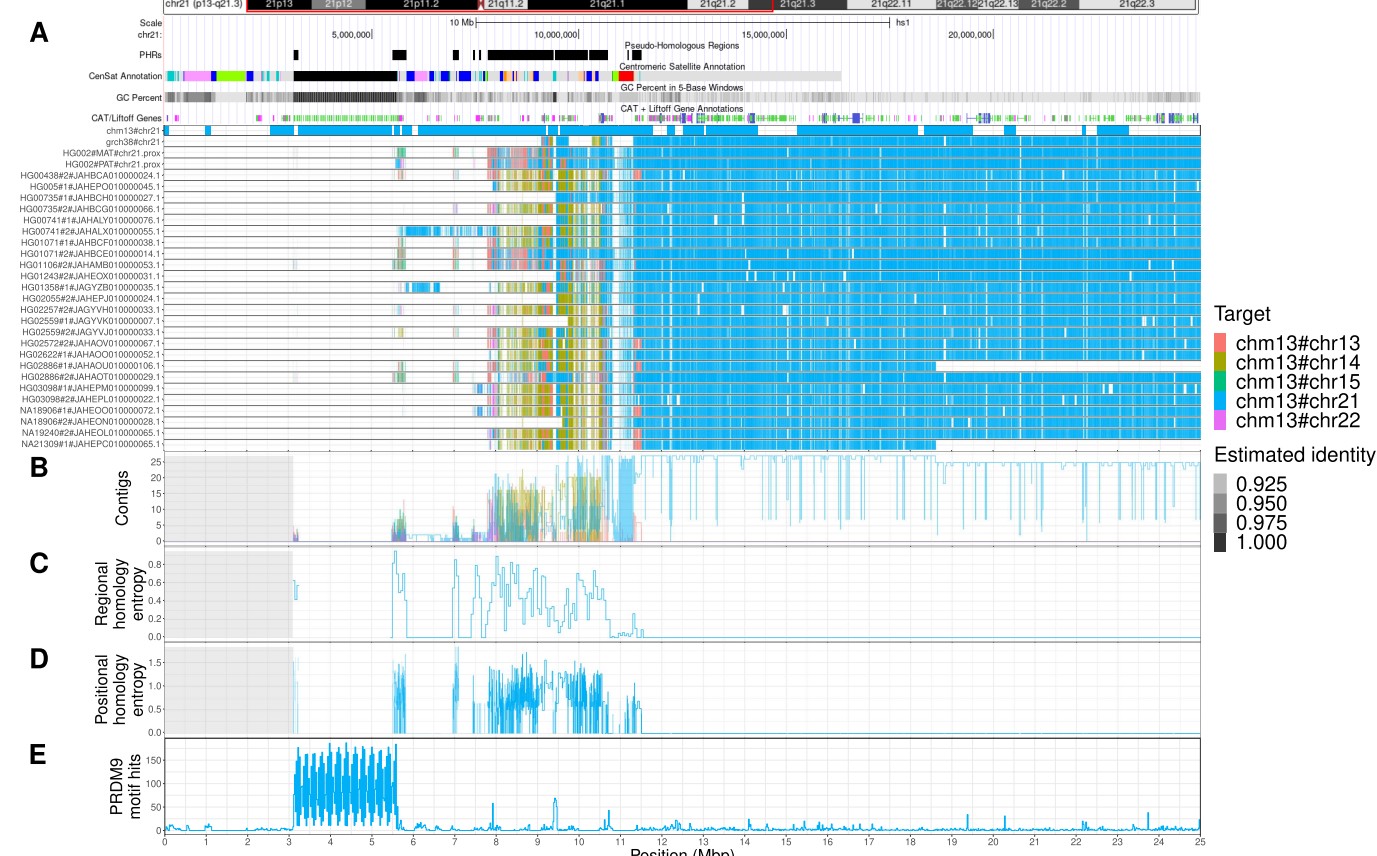

**Extended Data Fig. 7 | Characteristics of the pseudo-homologous regions of acrocentric chromosomes on chromosome 21. (A)** We focus on the first 25 Mbp of chromosome 21, shown here as a red box over T2T-CHM13 cytobands. Pseudo-homologous regions (**PHRs**), where diverse sets of acrocentric chromosomes recombine, are highlighted relative to T2T-CHM13 genome annotations for repeats, GC percentage, and genes. Above, we indicate regions of interest described in the main text: rDNA, SST1 array, centromere, and q-arm. Below, we show T2T-CHM13-relative homology mosaics for each chromosome 13 matched contig from HPRCy1-acro, with the most-similar reference chromosome at each region shown using the given colors (**Target**). **(B)** Aggregated untangle results in the SAACs. For each acrocentric chromosome, we show the count of its HPRCy1 q-arm-anchored contigs mapping itself and all other acrocentrics (**Contigs**), **(C)** as well as the regional (50kbp) untangle entropy metric (**Regional homology entropy**) computed over the contigs' T2T-CHM13-relative untanglings. **(D)** By considering the multiple untangling of each HPRCy1-acro contig, we develop a point-wise metric that captures diversity in T2T-CHM13-relative homology patterns (**Positional homology entropy**), leading to our definition of the PHRs. **(E)** The patterns of homology mosaicism suggest ongoing recombination exchange in the SAACs. A scan over T2T-CHM13 reveals that the rDNA units are enriched for PRDM9 binding motifs, and thus may host frequent double stranded breaks during meiosis. In (B-D) a gray background indicates regions with missing data due to the lack of non-T2T-CHM13 contigs. We provide the Centromeric Satellite Annotation (CenSat Annotation) track legend in Extended Data Table 1.

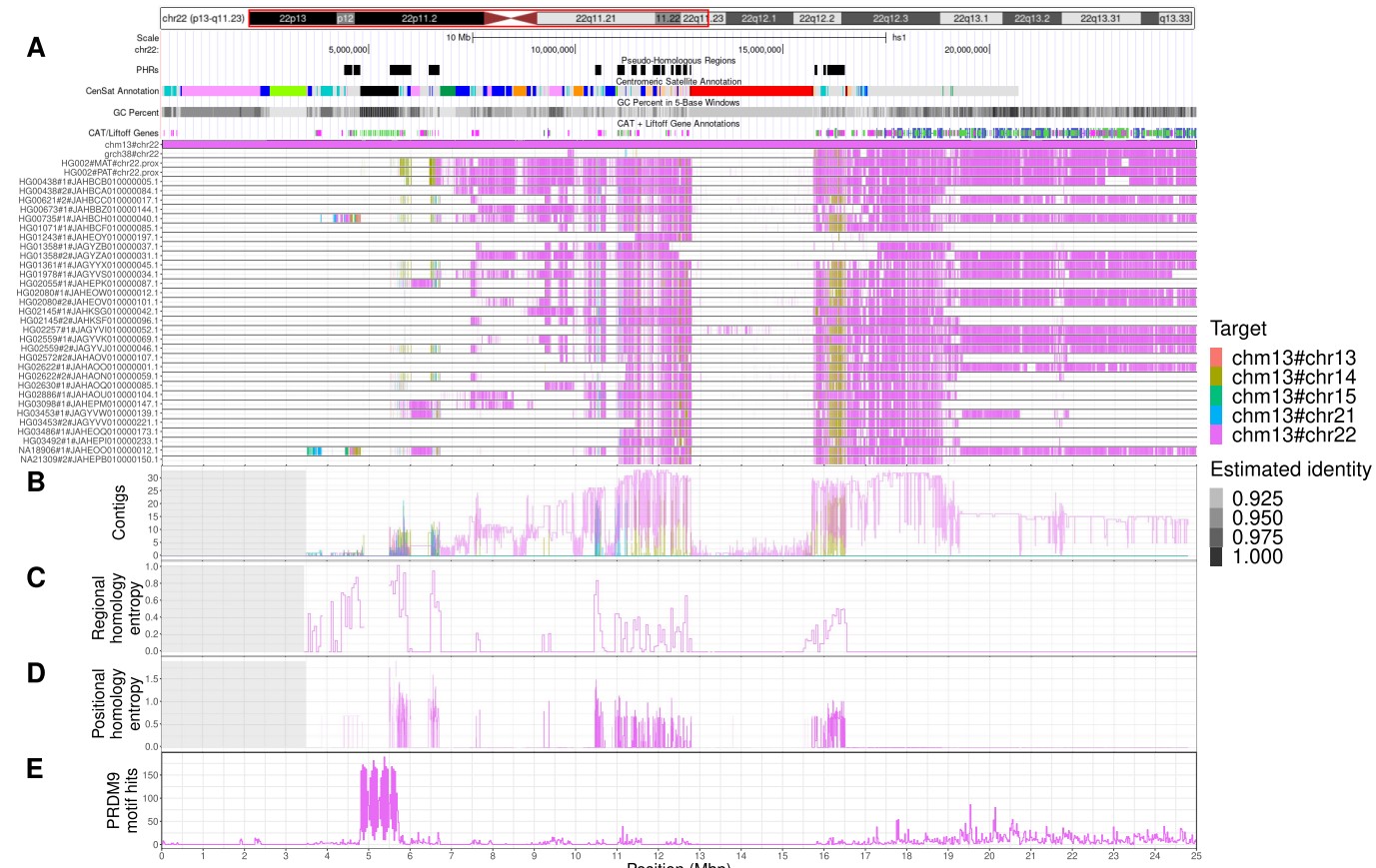

**Extended Data Fig. 8 | Characteristics of the pseudo-homologous regions of acrocentric chromosomes on chromosome 22. (A)** We focus on the first 25 Mbp of chromosome 22, shown here as a red box over T2T-CHM13 cytobands. Pseudo-homologous regions (**PHRs**), where diverse sets of acrocentric chromosomes recombine, are highlighted relative to T2T-CHM13 genome annotations for repeats, GC percentage, and genes. Above, we indicate regions of interest described in the main text: rDNA, SST1 array, centromere, and q-arm. Below, we show T2T-CHM13-relative homology mosaics for each chromosome 13 matched contig from HPRCy1-acro, with the most-similar reference chromosome at each region shown using the given colors (**Target**). **(B)** Aggregated untangle results in the SAACs. For each acrocentric chromosome, we show the count of its HPRCy1 q-arm-anchored contigs mapping itself and all other acrocentrics (**Contigs**), **(C)** as well as the regional (50kbp) untangle entropy metric (**Regional homology entropy**) computed over the contigs' T2T-CHM13-relative untanglings. **(D)** By considering the multiple untangling of each HPRCy1-acro contig, we develop a point-wise metric that captures diversity in T2T-CHM13-relative homology patterns (**Positional homology entropy**), leading to our definition of the PHRs. **(E)** The patterns of homology mosaicism suggest ongoing recombination exchange in the SAACs. A scan over T2T-CHM13 reveals that the rDNA units are enriched for PRDM9 binding motifs, and thus may host frequent double stranded breaks during meiosis. In (B-D) a gray background indicates regions with missing data due to the lack of non-T2T-CHM13 contigs. We provide the Centromeric Satellite Annotation (CenSat Annotation) track legend in Extended Data Table 1.

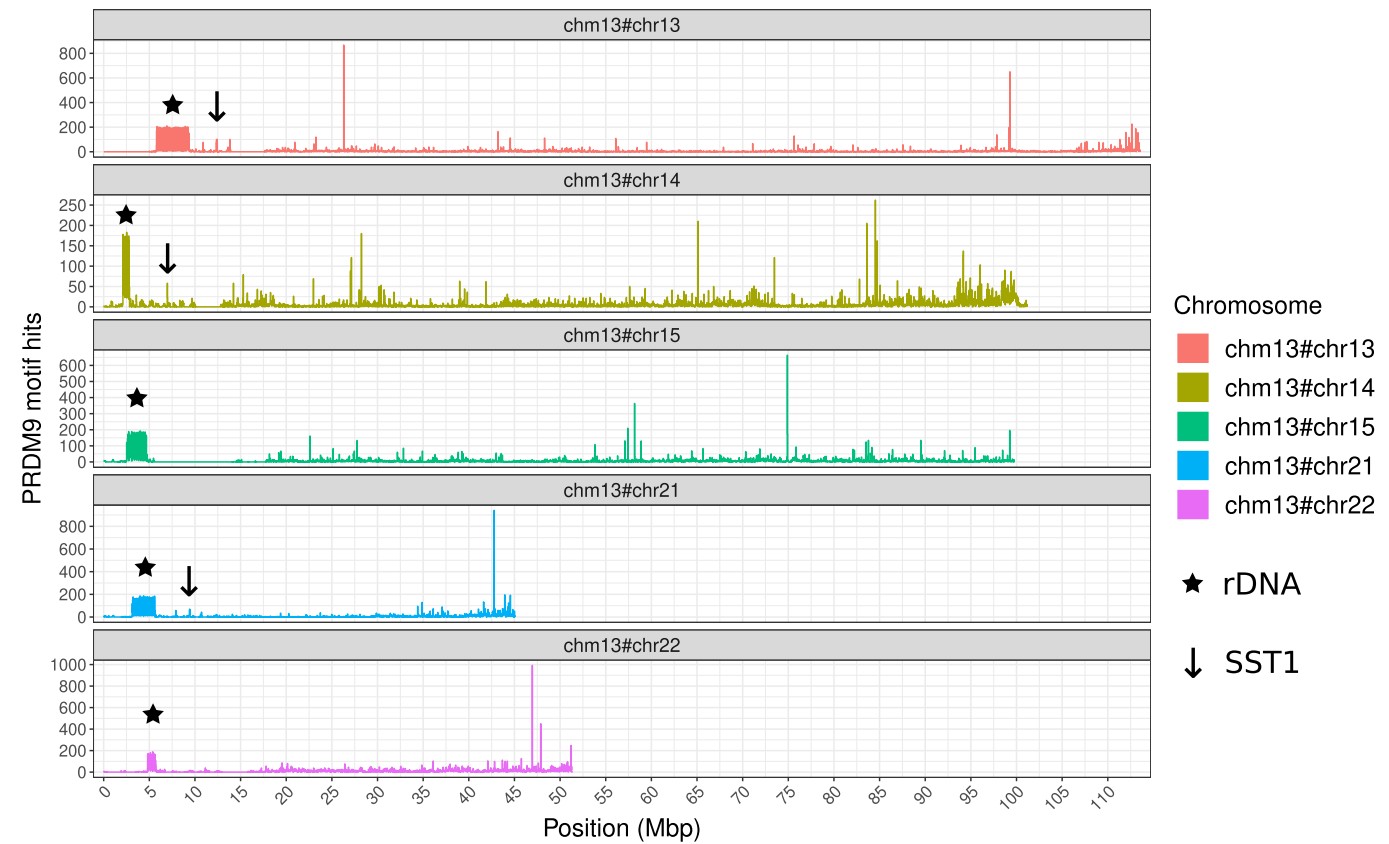

**Extended Data Fig. 9 | PRDM9 binding motif in the acrocentric chromosomes.** For each T2T-CHM13 acrocentric chromosome, we show the number of human PRDM9 binding motif hits present in windows 20 kbps long.

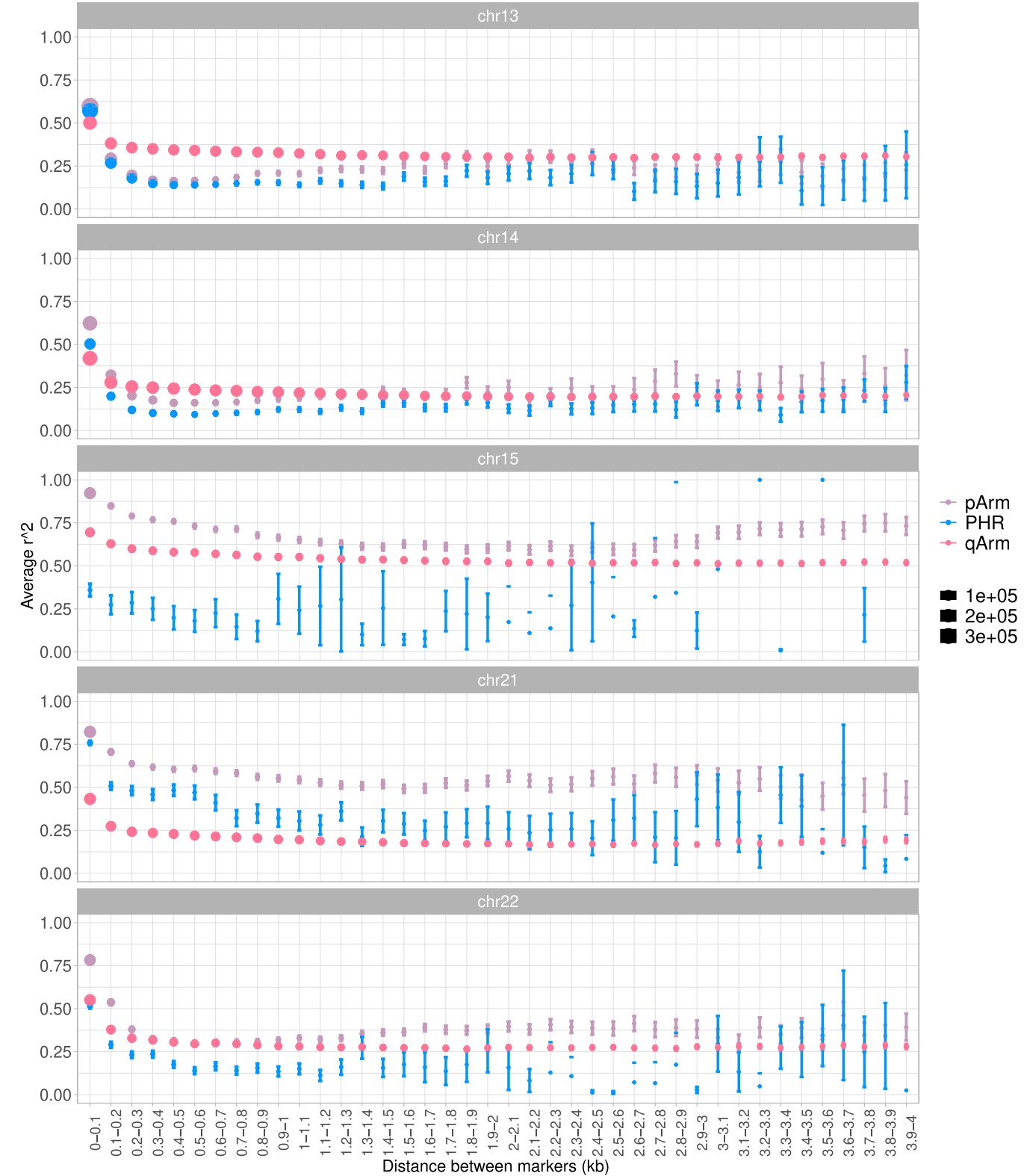

**Extended Data Fig. 10 | Linkage disequilibrium decay with distance between markers per acrocentric chromosome.** Each LD decay plot shows the p-arm (purple), q-arm (pink), and PHR (blue) mean $r^2$ (points) and 95% confidence intervals (error bars) for marker pairs binned by the given inter-marker distance range (x-axis). Dot size is proportional to the number of pairwise comparisons within a bin. LD decay is faster in PHRs for chromosomes 13, 14, and 22. No notable LD decay is observed in PHRs for chromosome 15.

**Extended Data Table 1 | Centromeric Satellite Annotation (CenSat Annotation) track legend**

| Annotation | Color |
|---|---|
| Active αSat HOR (hor … L) | red |
| Inactive αSat HOR (hor) | orange |
| Divergent αSat HOR (dhor) | dark red |
| Monomeric αSat (mon) | peach/yellow |
| Classical Human Satellite 1A (hsat1A) | light green |
| Classical Human Satellite 1B (hsat1B) | dark green |
| Classical Human Satellite 2 (hsat2) | light blue |
| Classical Human Satellite 3 (hsat3) | blue |
| Beta Satellite (bsat) | pink |
| Gamma Satellite (gsat) | purple |
| Other centromeric satellites (censat) | teal |
| Centromeric transition regions (ct) | grey |

All annotations are set to the forward strand. HOR = Higher Order Repeat.

# Reporting Summary

## Statistics

For all statistical analyses, confirm that the following items are present in the figure legend, table legend, main text, or Methods section.

| n/a | Confirmed | |
|---|---|---|
| ☒ | ☐ | The exact sample size (*n*) for each experimental group/condition, given as a discrete number and unit of measurement |
| ☒ | ☐ | A statement on whether measurements were taken from distinct samples or whether the same sample was measured repeatedly |
| ☒ | ☐ | The statistical test(s) used AND whether they are one- or two-sided<br>*Only common tests should be described solely by name; describe more complex techniques in the Methods section.* |
| ☒ | ☐ | A description of all covariates tested |
| ☒ | ☐ | A description of any assumptions or corrections, such as tests of normality and adjustment for multiple comparisons |
| ☐ | ☒ | A full description of the statistical parameters including central tendency (e.g. means) or other basic estimates (e.g. regression coefficient) AND variation (e.g. standard deviation) or associated estimates of uncertainty (e.g. confidence intervals) |
| ☒ | ☐ | For null hypothesis testing, the test statistic (e.g. *F*, *t*, *r*) with confidence intervals, effect sizes, degrees of freedom and *P* value noted<br>*Give P values as exact values whenever suitable.* |
| ☒ | ☐ | For Bayesian analysis, information on the choice of priors and Markov chain Monte Carlo settings |
| ☒ | ☐ | For hierarchical and complex designs, identification of the appropriate level for tests and full reporting of outcomes |
| ☒ | ☐ | Estimates of effect sizes (e.g. Cohen's *d*, Pearson's *r*), indicating how they were calculated |

*Our web collection on statistics for biologists contains articles on many of the points above.*

## Software and code

Policy information about availability of computer code

| | |
|---|---|
| Data collection | No software was used |
| Data analysis | wfmash, ad8aeba, https://github.com/waveygang/wfmash<br>seqwish, 706ef7e, https://github.com/ekg/seqwish<br>smoothxg, b3f4578, https://github.com/pangenome/smoothxg<br>vg, 1.40.0, https://github.com/vgteam/vg<br>GFAffix, 0.1.3          https://github.com/marschall-lab/GFAffix<br>odgi, 454197f, ttps://github.com/pangenome/odgi<br>pggb, a4a6668, https://github.com/pangenome/pggb<br>fastix, 331c115, https://github.com/ekg/fastix.git<br>igraph, 0.10.1, https://igraph.org/<br>gephi, 0.9.4, https://gephi.org/<br>PLINK, 1.90p, https://www.cog-genomics.org/plink/<br>verkko, vd3f0b, https://github.com/marbl/verkko<br>R, 4.2.1, https://www.r-project.org/<br>InkScape, 1.2.2, https://inkscape.org/<br>MEME Suite, 5.5.0, https://meme-suite.org/<br>Jmodeltest2, 2.1.10, https://github.com/ddarriba/jmodeltest2 |

For manuscripts utilizing custom algorithms or software that are central to the research but not yet described in published literature, software must be made available to editors and reviewers. We strongly encourage code deposition in a community repository (e.g. GitHub). See the Nature Portfolio guidelines for submitting code & software for further information.

# Data

Assemblies produced by the HPRC are available in the AnVIL (https://anvilproject.org/), in the AnVIL_HPRC workspace. Data is also available as part of the AWS Open Data Program (https://registry.opendata.aws/) in the human-pangenomics S3 bucket (https://s3-us-west-2.amazonaws.com/human-pangenomics/index.html). In addition, data is uploaded to INSDC for long term storage and availability. Supporting information about the data (including index files with S3 and GCP file locations) can be found at the following GitHub repository: https://github.com/human-pangenomics/HPP_Year1_Assemblies.

All supplementary files, including the pangenome variation graph and its layout, are available on Zenodo at https://doi.org/10.5281/zenodo.7692555.

# Human research participants

Policy information about studies involving human research participants and Sex and Gender in Research.

| | |
|---|---|
| Reporting on sex and gender | *Use the terms sex (biological attribute) and gender (shaped by social and cultural circumstances) carefully in order to avoid confusing both terms. Indicate if findings apply to only one sex or gender; describe whether sex and gender were considered in study design whether sex and/or gender was determined based on self-reporting or assigned and methods used. Provide in the source data disaggregated sex and gender data where this information has been collected, and consent has been obtained for sharing of individual-level data; provide overall numbers in this Reporting Summary. Please state if this information has not been collected. Report sex- and gender-based analyses where performed, justify reasons for lack of sex- and gender-based analysis.* |
| Population characteristics | *Describe the covariate-relevant population characteristics of the human research participants (e.g. age, genotypic information, past and current diagnosis and treatment categories). If you filled out the behavioural & social sciences study design questions and have nothing to add here, write "See above."* |
| Recruitment | *Describe how participants were recruited. Outline any potential self-selection bias or other biases that may be present and how these are likely to impact results.* |
| Ethics oversight | *Identify the organization(s) that approved the study protocol.* |

Note that full information on the approval of the study protocol must also be provided in the manuscript.

# Field-specific reporting

Please select the one below that is the best fit for your research. If you are not sure, read the appropriate sections before making your selection.

☒ Life sciences          ☐ Behavioural & social sciences          ☐ Ecological, evolutionary & environmental sciences

For a reference copy of the document with all sections, see nature.com/documents/nr-reporting-summary-flat.pdf

# Life sciences study design

All studies must disclose on these points even when the disclosure is negative.

| | |
|---|---|
| Sample size | As determined by HPRC funding, described in our companion paper. |
| Data exclusions | For initial community detection analysis, we considered all contigs >1Mbp. For acrocentric analysis, we retained assembly contigs that crossed acrocentric centromeres. |
| Replication | Experiments were computational so replication is not applicable. All methods are publicly posted in linked repositories and data has been generated from publicly available code and scripts. |
| Randomization | We did not allocate samples into different groups. |
| Blinding | Blinding is not relevant to our study as it is a survey of genetic variation in human acrocentric chromosomes. |

# Behavioural & social sciences study design

All studies must disclose on these points even when the disclosure is negative.

| | |
|---|---|
| Study description | *Briefly describe the study type including whether data are quantitative, qualitative, or mixed-methods (e.g. qualitative cross-sectional, quantitative experimental, mixed-methods case study).* |
| Research sample | *State the research sample (e.g. Harvard university undergraduates, villagers in rural India) and provide relevant demographic information (e.g. age, sex) and indicate whether the sample is representative. Provide a rationale for the study sample chosen. For studies involving existing datasets, please describe the dataset and source.* |
| Sampling strategy | *Describe the sampling procedure (e.g. random, snowball, stratified, convenience). Describe the statistical methods that were used to predetermine sample size OR if no sample-size calculation was performed, describe how sample sizes were chosen and provide a rationale for why these sample sizes are sufficient. For qualitative data, please indicate whether data saturation was considered, and what criteria were used to decide that no further sampling was needed.* |
| Data collection | *Provide details about the data collection procedure, including the instruments or devices used to record the data (e.g. pen and paper, computer, eye tracker, video or audio equipment) whether anyone was present besides the participant(s) and the researcher, and whether the researcher was blind to experimental condition and/or the study hypothesis during data collection.* |
| Timing | *Indicate the start and stop dates of data collection. If there is a gap between collection periods, state the dates for each sample cohort.* |
| Data exclusions | *If no data were excluded from the analyses, state so OR if data were excluded, provide the exact number of exclusions and the rationale behind them, indicating whether exclusion criteria were pre-established.* |
| Non-participation | *State how many participants dropped out/declined participation and the reason(s) given OR provide response rate OR state that no participants dropped out/declined participation.* |
| Randomization | *If participants were not allocated into experimental groups, state so OR describe how participants were allocated to groups, and if allocation was not random, describe how covariates were controlled.* |

# Ecological, evolutionary & environmental sciences study design

All studies must disclose on these points even when the disclosure is negative.

| | |
|---|---|
| Study description | *Briefly describe the study. For quantitative data include treatment factors and interactions, design structure (e.g. factorial, nested, hierarchical), nature and number of experimental units and replicates.* |
| Research sample | *Describe the research sample (e.g. a group of tagged Passer domesticus, all Stenocereus thurberi within Organ Pipe Cactus National Monument), and provide a rationale for the sample choice. When relevant, describe the organism taxa, source, sex, age range and any manipulations. State what population the sample is meant to represent when applicable. For studies involving existing datasets, describe the data and its source.* |
| Sampling strategy | *Note the sampling procedure. Describe the statistical methods that were used to predetermine sample size OR if no sample-size calculation was performed, describe how sample sizes were chosen and provide a rationale for why these sample sizes are sufficient.* |
| Data collection | *Describe the data collection procedure, including who recorded the data and how.* |
| Timing and spatial scale | *Indicate the start and stop dates of data collection, noting the frequency and periodicity of sampling and providing a rationale for these choices. If there is a gap between collection periods, state the dates for each sample cohort. Specify the spatial scale from which the data are taken* |
| Data exclusions | *If no data were excluded from the analyses, state so OR if data were excluded, describe the exclusions and the rationale behind them, indicating whether exclusion criteria were pre-established.* |
| Reproducibility | *Describe the measures taken to verify the reproducibility of experimental findings. For each experiment, note whether any attempts to repeat the experiment failed OR state that all attempts to repeat the experiment were successful.* |
| Randomization | *Describe how samples/organisms/participants were allocated into groups. If allocation was not random, describe how covariates were controlled. If this is not relevant to your study, explain why.* |
| Blinding | *Describe the extent of blinding used during data acquisition and analysis. If blinding was not possible, describe why OR explain why blinding was not relevant to your study.* |

Did the study involve field work?  ☐ Yes  ☐ No

## Field work, collection and transport

| | |
|---|---|
| Field conditions | *Describe the study conditions for field work, providing relevant parameters (e.g. temperature, rainfall).* |
| Location | *State the location of the sampling or experiment, providing relevant parameters (e.g. latitude and longitude, elevation, water depth).* |
| Access & import/export | *Describe the efforts you have made to access habitats and to collect and import/export your samples in a responsible manner and in compliance with local, national and international laws, noting any permits that were obtained (give the name of the issuing authority, the date of issue, and any identifying information).* |
| Disturbance | *Describe any disturbance caused by the study and how it was minimized.* |

# Reporting for specific materials, systems and methods

We require information from authors about some types of materials, experimental systems and methods used in many studies. Here, indicate whether each material, system or method listed is relevant to your study. If you are not sure if a list item applies to your research, read the appropriate section before selecting a response.

### Materials & experimental systems

| n/a | Involved in the study |
|---|---|
| ☒ | ☐ Antibodies |
| ☐ | ☒ Eukaryotic cell lines |
| ☒ | ☐ Palaeontology and archaeology |
| ☒ | ☐ Animals and other organisms |
| ☒ | ☐ Clinical data |
| ☒ | ☐ Dual use research of concern |

### Methods

| n/a | Involved in the study |
|---|---|
| ☒ | ☐ ChIP-seq |
| ☒ | ☐ Flow cytometry |
| ☒ | ☐ MRI-based neuroimaging |

## Antibodies

| | |
|---|---|
| Antibodies used | *Describe all antibodies used in the study; as applicable, provide supplier name, catalog number, clone name, and lot number.* |
| Validation | *Describe the validation of each primary antibody for the species and application, noting any validation statements on the manufacturer's website, relevant citations, antibody profiles in online databases, or data provided in the manuscript.* |

## Eukaryotic cell lines

Policy information about cell lines and Sex and Gender in Research

| | |
|---|---|
| Cell line source(s) | We used cell lines from the 1KG collection that is banked at Coriell, with processing conducted by the HPRC. |
| Authentication | Cell lines were authenticated by NHGRI and Coriell, as well as in our companion HPRC publication. |
| Mycoplasma contamination | Cell lines tested negative for mycoplasma contamination. |
| Commonly misidentified lines (See ICLAC register) | None used. |

## Palaeontology and Archaeology

| | |
|---|---|
| Specimen provenance | *Provide provenance information for specimens and describe permits that were obtained for the work (including the name of the issuing authority, the date of issue, and any identifying information). Permits should encompass collection and, where applicable, export.* |
| Specimen deposition | *Indicate where the specimens have been deposited to permit free access by other researchers.* |
| Dating methods | *If new dates are provided, describe how they were obtained (e.g. collection, storage, sample pretreatment and measurement), where they were obtained (i.e. lab name), the calibration program and the protocol for quality assurance OR state that no new dates are provided.* |

☐ Tick this box to confirm that the raw and calibrated dates are available in the paper or in Supplementary Information.

| Ethics oversight | *Identify the organization(s) that approved or provided guidance on the study protocol, OR state that no ethical approval or guidance was required and explain why not.* |

Note that full information on the approval of the study protocol must also be provided in the manuscript.

# Animals and other research organisms

Policy information about studies involving animals; ARRIVE guidelines recommended for reporting animal research, and Sex and Gender in Research

| Laboratory animals | *For laboratory animals, report species, strain and age OR state that the study did not involve laboratory animals.* |
| Wild animals | *Provide details on animals observed in or captured in the field; report species and age where possible. Describe how animals were caught and transported and what happened to captive animals after the study (if killed, explain why and describe method; if released, say where and when) OR state that the study did not involve wild animals.* |
| Reporting on sex | *Indicate if findings apply to only one sex; describe whether sex was considered in study design, methods used for assigning sex. Provide data disaggregated for sex where this information has been collected in the source data as appropriate; provide overall numbers in this Reporting Summary. Please state if this information has not been collected. Report sex-based analyses where performed, justify reasons for lack of sex-based analysis.* |
| Field-collected samples | *For laboratory work with field-collected samples, describe all relevant parameters such as housing, maintenance, temperature, photoperiod and end-of-experiment protocol OR state that the study did not involve samples collected from the field.* |
| Ethics oversight | *Identify the organization(s) that approved or provided guidance on the study protocol, OR state that no ethical approval or guidance was required and explain why not.* |

Note that full information on the approval of the study protocol must also be provided in the manuscript.

# Clinical data

Policy information about clinical studies

All manuscripts should comply with the ICMJE guidelines for publication of clinical research and a completed CONSORT checklist must be included with all submissions.

| Clinical trial registration | *Provide the trial registration number from ClinicalTrials.gov or an equivalent agency.* |
| Study protocol | *Note where the full trial protocol can be accessed OR if not available, explain why.* |
| Data collection | *Describe the settings and locales of data collection, noting the time periods of recruitment and data collection.* |
| Outcomes | *Describe how you pre-defined primary and secondary outcome measures and how you assessed these measures.* |

# Dual use research of concern

Policy information about dual use research of concern

## Hazards

Could the accidental, deliberate or reckless misuse of agents or technologies generated in the work, or the application of information presented in the manuscript, pose a threat to:

No | Yes
- ☒ ☐ Public health
- ☒ ☐ National security
- ☒ ☐ Crops and/or livestock
- ☒ ☐ Ecosystems
- ☒ ☐ Any other significant area

## Experiments of concern

Does the work involve any of these experiments of concern:

| No | Yes | |
|----|-----|---|
| ☒ | ☐ | Demonstrate how to render a vaccine ineffective |
| ☒ | ☐ | Confer resistance to therapeutically useful antibiotics or antiviral agents |
| ☒ | ☐ | Enhance the virulence of a pathogen or render a nonpathogen virulent |
| ☒ | ☐ | Increase transmissibility of a pathogen |
| ☒ | ☐ | Alter the host range of a pathogen |
| ☒ | ☐ | Enable evasion of diagnostic/detection modalities |
| ☒ | ☐ | Enable the weaponization of a biological agent or toxin |
| ☒ | ☐ | Any other potentially harmful combination of experiments and agents |

# ChIP-seq

## Data deposition

☐ Confirm that both raw and final processed data have been deposited in a public database such as GEO.

☐ Confirm that you have deposited or provided access to graph files (e.g. BED files) for the called peaks.

| | |
|---|---|
| Data access links *May remain private before publication.* | *For "Initial submission" or "Revised version" documents, provide reviewer access links. For your "Final submission" document, provide a link to the deposited data.* |
| Files in database submission | *Provide a list of all files available in the database submission.* |
| Genome browser session (e.g. UCSC) | *Provide a link to an anonymized genome browser session for "Initial submission" and "Revised version" documents only, to enable peer review. Write "no longer applicable" for "Final submission" documents.* |

## Methodology

| | |
|---|---|
| Replicates | *Describe the experimental replicates, specifying number, type and replicate agreement.* |
| Sequencing depth | *Describe the sequencing depth for each experiment, providing the total number of reads, uniquely mapped reads, length of reads and whether they were paired- or single-end.* |
| Antibodies | *Describe the antibodies used for the ChIP-seq experiments; as applicable, provide supplier name, catalog number, clone name, and lot number.* |
| Peak calling parameters | *Specify the command line program and parameters used for read mapping and peak calling, including the ChIP, control and index files used.* |
| Data quality | *Describe the methods used to ensure data quality in full detail, including how many peaks are at FDR 5% and above 5-fold enrichment.* |
| Software | *Describe the software used to collect and analyze the ChIP-seq data. For custom code that has been deposited into a community repository, provide accession details.* |

# Flow Cytometry

## Plots

Confirm that:

☐ The axis labels state the marker and fluorochrome used (e.g. CD4-FITC).

☐ The axis scales are clearly visible. Include numbers along axes only for bottom left plot of group (a 'group' is an analysis of identical markers).

☐ All plots are contour plots with outliers or pseudocolor plots.

☐ A numerical value for number of cells or percentage (with statistics) is provided.

## Methodology

| | |
|---|---|
| Sample preparation | *Describe the sample preparation, detailing the biological source of the cells and any tissue processing steps used.* |
| Instrument | *Identify the instrument used for data collection, specifying make and model number.* |

| Software | *Describe the software used to collect and analyze the flow cytometry data. For custom code that has been deposited into a community repository, provide accession details.* |
|---|---|
| Cell population abundance | *Describe the abundance of the relevant cell populations within post-sort fractions, providing details on the purity of the samples and how it was determined.* |
| Gating strategy | *Describe the gating strategy used for all relevant experiments, specifying the preliminary FSC/SSC gates of the starting cell population, indicating where boundaries between "positive" and "negative" staining cell populations are defined.* |

☐ Tick this box to confirm that a figure exemplifying the gating strategy is provided in the Supplementary Information.

# Magnetic resonance imaging

## Experimental design

| Design type | *Indicate task or resting state; event-related or block design.* |
|---|---|
| Design specifications | *Specify the number of blocks, trials or experimental units per session and/or subject, and specify the length of each trial or block (if trials are blocked) and interval between trials.* |
| Behavioral performance measures | *State number and/or type of variables recorded (e.g. correct button press, response time) and what statistics were used to establish that the subjects were performing the task as expected (e.g. mean, range, and/or standard deviation across subjects).* |

## Acquisition

| Imaging type(s) | *Specify: functional, structural, diffusion, perfusion.* |
|---|---|
| Field strength | *Specify in Tesla* |
| Sequence & imaging parameters | *Specify the pulse sequence type (gradient echo, spin echo, etc.), imaging type (EPI, spiral, etc.), field of view, matrix size, slice thickness, orientation and TE/TR/flip angle.* |
| Area of acquisition | *State whether a whole brain scan was used OR define the area of acquisition, describing how the region was determined.* |

Diffusion MRI      ☐ Used      ☐ Not used

## Preprocessing

| Preprocessing software | *Provide detail on software version and revision number and on specific parameters (model/functions, brain extraction, segmentation, smoothing kernel size, etc.).* |
|---|---|
| Normalization | *If data were normalized/standardized, describe the approach(es): specify linear or non-linear and define image types used for transformation OR indicate that data were not normalized and explain rationale for lack of normalization.* |
| Normalization template | *Describe the template used for normalization/transformation, specifying subject space or group standardized space (e.g. original Talairach, MNI305, ICBM152) OR indicate that the data were not normalized.* |
| Noise and artifact removal | *Describe your procedure(s) for artifact and structured noise removal, specifying motion parameters, tissue signals and physiological signals (heart rate, respiration).* |
| Volume censoring | *Define your software and/or method and criteria for volume censoring, and state the extent of such censoring.* |

## Statistical modeling & inference

| Model type and settings | *Specify type (mass univariate, multivariate, RSA, predictive, etc.) and describe essential details of the model at the first and second levels (e.g. fixed, random or mixed effects; drift or auto-correlation).* |
|---|---|
| Effect(s) tested | *Define precise effect in terms of the task or stimulus conditions instead of psychological concepts and indicate whether ANOVA or factorial designs were used.* |

Specify type of analysis:      ☐ Whole brain      ☐ ROI-based      ☐ Both

| Statistic type for inference (See Eklund et al. 2016) | *Specify voxel-wise or cluster-wise and report all relevant parameters for cluster-wise methods.* |
|---|---|
| Correction | *Describe the type of correction and how it is obtained for multiple comparisons (e.g. FWE, FDR, permutation or Monte Carlo).* |

## Models & analysis

| n/a | Involved in the study |
|-----|----------------------|
| ☐ | ☐ Functional and/or effective connectivity |
| ☐ | ☐ Graph analysis |
| ☐ | ☐ Multivariate modeling or predictive analysis |

**Functional and/or effective connectivity**  
*Report the measures of dependence used and the model details (e.g. Pearson correlation, partial correlation, mutual information).*

**Graph analysis**  
*Report the dependent variable and connectivity measure, specifying weighted graph or binarized graph, subject- or group-level, and the global and/or node summaries used (e.g. clustering coefficient, efficiency, etc.).*

**Multivariate modeling and predictive analysis**  
*Specify independent variables, features extraction and dimension reduction, model, training and evaluation metrics.*

