## [Peer Review File · Nature]

Manuscript Title: Recombination between heterologous human acrocentric chromosomes

Reviewer Comments & Author Rebuttals

Reviewer Reports on the Initial Version:

Referees' comments:

Referee #1 (Remarks to the Author):

Guarracino and colleagues provide sequence and population-based confirmation for the hypotheses of heterologous recombination between the short arms of the acrocentric contigs (SAACs). Firstly, the authors performed a chromosome community detection using HPRCy1 pangenome's assemblies and found that the acrocentric form combined communities containing multiple chromosomes, suggesting the ongoing pseudo-homologous recombination. Second, the authors presented the exchange between the heterologous chromosomes by constructing an all-acrocentric pangenome graph using PGGB. Third, the authors cross-validate these findings using the HG002 verkko assembly. Last, the authors provided LD decay results of the pseudo-homologous regions to further support the findings.

The methodology for contrasting the PVG of complex regions will be of particular interest in the era of graph pangenome. In addition, I believe the study will be likely of broad interests if the authors provide an example to show a biological impact of the heterologous recombination.

Below are some of my concerns:

1. Is there any biological implication for the recombination?
2. The manuscript is more about the confirmation of the phenomenon from graph pangenome, but how these findings will help the biological interpreting of the recombination?
3. Page 6: The quality of centromere-spanning contigs. Since these contigs were derived from HiFi-based assembler. Spanning centromere doesn't mean it was not a misjoin. Although, Liao et.al presented dozens of validations for HPRCy1, but it will be helpful if authors can show some evidence about these contigs quality validation with the correspond sequencing data. It was the basis for acro-PVG.
4. Page 6: The authors only used the 132 contigs covering both sides of the centromere of the acrocentric chromosomes for contrasting the PVG. Theoretically, the total number should be 470. Why the left 338 contigs cannot be assembled? Is there any other biological explanation? Will the recombination cause the translocation of different short arms?
5. Page 8: The term SST1 array is a specific region in human genome. It will be better to explain this term before the conclusion as it is likely unfamiliar with these non-human researchers.
6. Page 15: As the accuracy of SNPs is much important to calculate the LD decay. The authors claimed they used the marker within the variant space. But this concept is not clear in the manuscript. What's the features of the variant space? Did it mean the SNP directly called with pangenome graph, e.g., vg deconstruct?
7. If the ongoing exchange valid, the larger cohorts may can help for validation. Maybe we could see

these region found by acro-PVG have overlap with genomeAD or TOPMED translocation diversity although it was short-reads based. And maybe we can see more rapid recombination on cancer samples, since it have more aneuploid in naturally and maybe more non-homologous recombination.

8. Considering that this manuscript will be of broad interest, it would be better to have a figure illustrate the phenomenon for general readers.

Referee #2 (Remarks to the Author):

The rules governing the behaviour of the short arms of human acrocentric chromosomes are one of the major remaining unanswered question in human chromosome biology. These chromosome arms bear nucleolar organiser regions (NORs) that are composed of ribosomal gene (rDNA) arrays and seed formation of the nucleolus. How sequence uniformity of rDNA arrays across these chromosome arms is maintained is a key question. It has been hypothesised for quite some time that exchanges/recombination between the short p-arms of heterologous acrocentric chromosomes is involved. This is further supported by the shared satellite composition of these chromosome arms, as demonstrated by fluorescent in situ hybridisation (FISH) experiments. One noted example being sequence similarities of higher order (HOR) alpha satellite arrays on chromosomes 13/21 and 14/22.

Through the efforts of large sequencing consortia, the Telomere to Telomere (T2T) and latterly the Human Pangenome Reference consortium (HPRC), evidence for exchanges between heterologous chromosomes can now be described at the highest level of resolution, DNA sequence. Primarily using PacBio HiFi reads supplemented with Nanopore ultralong reads, the T2T determined the close to complete genome sequence, T2T-CHM13, for a highly homozygous genome of a complete hydatidiform mole cell line (CHM13). The only significant gaps being the longer rDNA arrays on some of the acrocentrics, which could none the less be modelled using other approaches. Ongoing work in the HPRC has yielded 94 haplotype-resolved assemblies for a series of normal diploid human cell lines (HPRCy1), based on HiFi sequencing For some cell lines, including HG002, more complete genome drafts have been obtained by combing HiFi with Nanopore ultralong reads, HiC and BioNano (optical mapping) data.

In this manuscript, HPRCy1 is used to explore patterns of variation among these chromosome arms and to reveal evidence for “ongoing recombination exchange” between heterologous chromosomes. With reference to the pseudo-autosomal regions (PAR1 and PAR2) on the X and Y sex chromosomes, the elements on acrocentric p-arms supporting these exchanges are described as “pseudo-homologous regions”. This is a significant advance on our understanding of the chromosome biology of these critical regions of our genome and will undoubtedly seed further investigation into when, where and how these exchanges occur. It will likely also lead to a molecular explanation of how Robertsonian translocations, the most common form of chromosomal translocation in the human population, arise.

Comments

The initial set of results describes visualisation of similarities between acrocentric short-arms by

grouping sequences according to their homologies, thus identifying “chromosome communities”. Fig 1A and B shows a homology map with >16k contigs of ≥ 1 Mb from HPRCy1 covering 98.72% of the genome. Chromosome identity was determined by mapping to T2T-CHM13 and GRCh38 and the number of hits was limited to 3 for ease of visualisation. This is further refined by use of a community detection algorithm that identified a community involving the p-arms of all five acrocentrics as well as the 21 and 22 q-arms. Although not stated, I assume that inclusion of these q-arms is related to their short length compared to 13-15, rather than direct sequence homologies. While really just confirmatory in nature, these results set the scene for a more refined comparison of acrocentric p-arms at the sequence level.

As HPRCy1 assemblies are derived solely from HiFi reads derived from normal diploid cell lines, they are less complete than T2T-CHM13. Nevertheless, by building assemblies from contigs anchored on the long q-arm of each acrocentric and spanning the centromere, sequences of rDNA proximal regions for each p-arm and haplotype could be determined. As before rDNA spanning contigs were rare. To bolster this list two versions of a more complete genome from cell line HG002, T2T-CHM13 and relevant contigs from GRCh38 were included, resulting in a collection, referred to as HPRCy1-acro. Pan genome graph builder (PGGB) was used to build pangenome variation graphs (PVGs) representing all pairwise alignments between contigs. The visualisation of these alignments is shown in a 2d graph in Fig 3. Not unsurprisingly, it reveals that all the across are anchored in rDNA, and that centromeres on alpha satellite HORs, 13 and 21 show close similarity, as do 14 and 22 and 15 is more isolated. More intriguing is identification of “pseudo-homologous regions” around tandem arrays of SST1 elements close to centromeres on T2T-CHM13’s 13p, 14p and 21p (identity >99%). However, I am confused by the statement in the legend to Fig 3 that they are in the “direct orientation” on these chromosome arms. As I understood from Fig 4 in the original Nurk et al T2T Science paper, this region was inverted on 14 relative to 13 and 21. Is this an error? If so, could it be that this is due to many of the contigs in HPRCy1-acro being based only on HiFi and not supported by Nanopore reads? This should be clarified as it may have relevance to the origin of the most common Robertsonian translocations, involving these regions on the same chromosomes.

The core findings of this manuscript are displayed in Fig 4 where the acrocentric PVG is “untangled” by deriving pairwise alignments, mapping each part of each HPRCy1-acro contig against T2T-CHM13, retaining only those that have >90% estimated identity. The amount of data presented in this figure is very large and the annotation is incomplete, particularly in relation to the colour coded repeat masking data. Nevertheless, evidence for significant levels of recombination/exchanges within “pseudo-homologous regions” (PHRs), particularly involving 13, 14 and 21, is very striking. Diversity entropy is another metric describing the degree of disorder in the untangling of HPRCy1-acro contigs. This begins at zero in the acrocentric q-arms where there is no evidence of exchange to approaching 1 in PHRs.

The data presented in Fig 5 focuses on a subset of the HPRCy1-acro contigs for 13, 14, and 21. It shows the best 5 mappings to T2T-CHM13 for each contig. This data further refines PHRs as linked to rDNA arrays and surrounding the STS arrays on 13, 14 and 21. Moreover, better estimates of their lengths can be determined. Chromosomes 13, 14 and 21 have PHRs of comparable sizes to the PARs on sex chromosomes. In line with previous data in this manuscript, chromosome 15 has the shortest PHR at 0.5Mb. Does this higher resolution data provide any information on the lengths of typical exchanges. Are their lengths compatible with gene conversion or do they arise from double

crossover events?

The next data concerns the decay of linkage disequilibrium by plotting the r^2 allele correlation versus distance for variants on the q-arm, the p-arm, and within the PHRs. They observe that for chromosomes 13, 14, and 22, the LD decay is faster in the PHRs compared to p- and q-arms and point out that "This effect is strongest within the chromosomes that other analyses suggest to share the most homologous sequence: 13, 14, and 22". I find this somewhat strange as much of the previous data concentrates on homologies between 13,14 and 21 (not 22) involving PHRs that include SST1 elements. Can the authors comment.

Finally, as most HPRCy1-acro contigs are assembled using only HiFi contigs there is an attempt to validate this data by comparing the two HG002 genome builds HG002-HPRCy1 and HG002-Verkko. The results suggest that HiFi-only assemblies are "likely" accurate representation of haplotypes. The authors rightly declare that their population-based study that does not speak to mechanism or timing of exchanges/recombination between heterologous acrocentrics. However, it would be interesting to hear their views on whether examination of family pedigrees would be expected to shed light on mechanism and frequency of exchanges. For example like Robertsonian translocations do they arise primarily during female meiosis.

Additional comment:

In a number of instances the text describes association of acrocentric chromosomes with the "nucleolar membrane" (eg last paragraph of discussion). It is now firmly established that nucleoli are membraneless organelles.

Referee #3 (Remarks to the Author):

That acrocentric chromosomes share sequence identities, as discovered in the T2T human genome sequencing project, suggestive of ongoing recombination is exciting. Indeed, rDNA clusters from different clusters localizing to the nucleolus within cells makes for an opportunity for non-homologous chromosomes to potentially share sequence. This study seeks to detect evidence of recombination between non-homologous chromosomes rDNA regions, resulting in shared sequences. The authors analyze high-quality, haplotype-resolved, sequences of a diversity of humans to address this question. They provide convincing evidence that recombination has shaped the evolution of the human acrocentric chromosomal regions around the rDNA clusters. This is an important finding and step to understanding these complex genomic regions. However, outside of simply finding there are shared sequences across the non-homologous chromosomes, the manuscript does not address some fundamental questions on these shared sequences. Moreover, the manuscript text is heavy on jargon, making it difficult to follow for the general genetics/genomics/bioinformatics reader.

Major concerns:

1. What is the significance of the shared sequence homologies? For example, do they provide an anchor for pairing rDNA clusters in the nucleus? Outside of stating just that these shared homologies exist, it would be important to provide an understanding of what is driving the sharing of these sequences.

1. Can the authors distinguish between gene conversion or crossing over to explain the homologous

sequences in the acrocentric chromosomes? It is only inferred that there are homologies, but how these homologies are generated, outside of speculating that recombination, is important. Examining homogenization tract lengths be assessed across the diversity panels to assess if there are any patterns that illuminate the recombination dynamics?

3. The authors argue that there is ongoing sequence exchange between the acrocentric chromosome autosomes. What is the frequency of this exchange? Does it happen every generation? For example, to assess the frequency of recombination, could the authors use trio data or comparing diverged haplotypes? Are some regions more “hotspots” for exchange versus other regions?

4. The authors argue “In the homology mapping graph of the HPRCy1, only the acrocentric and sex chromosomes form combined communities containing multiple chromosomes. The sex chromosome community reflects the pseudoautosomal regions (PARs) on X and Y (23), which are telomeric regions where these otherwise non-homologous chromosomes recombine as if they were homologs.”

While the four acrocentric chromosomes are near each other in the community plot (Fig. 1), relating it to the X and Y doesn't seem justified. The X and Y are separated on the plot, similar to other chromosomes. Other autosomes, for example 11 and 1 or 18 and 11 look closer, than the sex chromosomes. Do the other autosomes have examples of non-rDNA repeats, satellite DNA, or other repetitive sequence that promote recombination between the autosomes, like the rDNA regions?

5. To build on using the sex chromosomes as a control, the authors could map or highlight strata 4 + 5 of the X and Y, which have similar levels of nucleotide identity and see if they connect to each other on the community plots? These last two strata are the primary regions known to still undergo homologous recombination between the X and Y, albeit not that frequently.

Author Rebuttals to Initial Comments:

Referee #1 (Remarks to the Author):

Guarracino and colleagues provide sequence and population-based confirmation for the hypotheses of heterologous recombination between the short arms of the acrocentric contigs (SAACs). Firstly, the authors performed a chromosome community detection using HPRCy1 pangenome's assemblies and found that the acrocentric form combined communities containing multiple chromosomes, suggesting the ongoing pseudo-homologous recombination. Second, the authors presented the exchange between the heterologous chromosomes by constructing an all-acrocentric pangenome graph using PGGB. Third, the authors cross-validate these findings using the HG002 verkko assembly. Last, the authors provided LD decay results of the pseudo-homologous regions to further support the findings.

The methodology for contrasting the PVG of complex regions will be of particular interest in the era of graph pangenome. In addition, I believe the study will be likely of broad interests if the authors provide an example to show a biological impact of the heterologous recombination.

Below are some of my concerns:

1. Is there any biological implication for the recombination?

We agree with the reviewer that we did not address the possible implications of recombination in the first version of the manuscript. In this revised version we add evidence that links the pseudo-homologous regions (PHRs) to Robertsonian translocation (ROB), the main relevant implication for the recombination between acrocentric chromosomes. ROB is the most common form of chromosomal translocation in humans (1 out of every 800 births). This aberrant event involves only the acrocentric chromosomes and results in one chromosome being joined end to end with another, reducing the total number of chromosomes in the cell. This type of translocation can involve non-homologous chromosomes, and this is in line with our results. Interestingly, Robertsonian translocations form in the region between the centromere and ribosomal DNA, the region we address in our study. Furthermore, bacterial artificial chromosome clones surrounding common recurrent ROB breakpoints map to the PHRs we identified on T2T-CHM13. We updated the manuscript by including this new analysis on ROB breakpoints (**Figure 4D**) and discussing more in-depth the possible implications of the recombination. We summarized the findings and speculation on ROB in an entirely new figure in the discussion (**Figure 6**, see also response to comment #8 of Reviewer1 and our response to Reviewer2).

2. The manuscript is more about the confirmation of the phenomenon from graph pangenome, but how these findings will help the biological interpreting of the recombination?

This study paves the way to sequence-based interpretation of biological phenomena in the SAACs. Our analysis is the first regarding genome variation in this region, whose sequence in multiple individuals has only recently become available for study (HPRCy1), and provides an a-posteriori explanation for recombination-related phenomena which we observe. We stress the fact that this is just the beginning of sequence-based study into the biological features of the PHRs and acrocentric p-arms. We were previously unable to precisely localize these regions of recombination hotspots on the acrocentric arms, and this study is

the first to explore these with the very recently produced complete sequences of the chromosomes (T2T-CHM13). Comparing T2T-CHM13 to the HPRCy1 assemblies allows us to make the first “population” study of sequence variation in this region.

In hopes of addressing biological implications of recombination, we improved our work’s connection with existing studies of Robertsonian translocations, finding that regions suggested to be recombinant by our population based analyses include putative recurrent Robertsonian breakpoints. We include discussion of biological features relevant to the recombination process, including meiotic PRDM9 sites, physical proximity and potential shared function of duplicated sequences found on the SAACs. This leads to an overview of recombination processes in the SAACs that should provide the foundation for future biological studies into the region.

3. Page 6: The quality of centromere-spanning contigs. Since these contigs were derived from HiFi-based assembler. Spanning centromere doesn’t meaning it was not a misjoin. Although, Liao et.al presented dozens of validations for HPRCy1, but it will be helpful if authors can show some evidence about these contigs quality validation with the correspond sequencing data. It was the basis for acro-PVG.

In the submitted manuscript we had already reported some quality information on the contigs covering the centromere, but we acknowledge that they have not been adequately presented. In the revised manuscript, we explained that we used FLAGGER (Liao et a., 2022) to determine which portions of the assemblies are reliable. FLAGGER is a HiFi read-based pipeline that detects different types of misassemblies within a phased diploid assembly by identifying read-mapping coverage inconsistencies across the maternal and paternal haplotypes. With this approach, misjoins are often detected as regions that are unreliable. With FLAGGER, we then validated the centromere-spanning contigs by using their corresponding sequencing data, focusing our analysis only on the portions of the contigs not labeled as unreliable. In **Supplementary Table 2**, we reported, for each contig in the acro-PVG, the amounts of sequence labeled as reliable and unreliable by FLAGGER. Furthermore, during this review, we also produced new supplementary figures (**Supplementary Figures 7-11**) to highlight the location and size of the unreliable regions across the contigs. These analyses show that the contigs we used only have a limited number of regions classified as unreliable (with chromosome 22’s contigs having the highest amount of them), regions that we excluded from downstream analyses. We refer to the (Liao et a., 2022) manuscript for further details about FLAGGER and all the thorough validations performed for the HPRCy1 dataset.

4. Page 6: The authors only used the 132 contigs covering both sides of the centromere of the acrocentric chromosomes for contruting the PVG. Theotically, the total number should be 470. Why the left 338 contigs cannot be assembled? Is there any other biological explanation? Will the recombination cause the translocation of different short arms?

To answer the first two questions: We assume that the reviewer refers to 470 as 94 haplotypes (haploid genomes) times 5 acrocentric chromosomes. This is an ideal situation in which every haplotype has one contig spanning the centromere for each acrocentric

chromosome, and one would use all those 470 contigs to build the pangenome. In reality, despite the high quality of the sequence data, many haploid genomes lack centromere-spanning contigs and therefore the analysis was restricted to the 132 contigs that cover both sides of the centromere of at least 1 Mbp, discarding contigs from 338 chromosomes that do not have a contig which meets this requirement. There is no other biological explanation for leaving these potential 338 contigs out of the analysis. Centromeres are difficult to assemble due to their repetitive nature, which is itself one of their key biological features. Although difficult, it is possible to assemble them in humans using HiFi-only data. This is why for many individuals we do have contigs that cover the entire centromere, but the rate at which this happens is still low. In our work, the centromere-spanning requirement is essential to determine the identity of each contig, that is, the chromosome to which it belongs. Assemblies produced by a process that combines both HiFi and ONT ultra-long reads (e.g. HG002-Verkko) tend to yield complete centromere assemblies, but at the time of our work these have been produced for only a handful of genomes in the HPRC and the data generation and assembly process to produce these for all samples is still in development.

As for the third question: Regarding the effect of the recombination, the reviewer is right, it can cause the translocation of different short arms non-homologous acrocentric chromosomes. Crossover is not something we can directly observe in HPRCy1—instead we see patterns of homology indicative of past recombination. Furthermore we do not believe HPRCy1 is sufficient to support confident estimation of the relative rate of crossover of different short arms. That noted, we do know that some crossover is occurring at some rate, as Robertsonian translocation directly implies crossover type recombination. See our response to the first comment for more details about this chromosome rearrangement.

5. Page 8: The term SST1 array is a specific region in human genome. It will be better to explain this term before the conclusion as it is likely unfamiliar with these non-human researchers.

We agree with the reviewer. In the revised manuscript, we added the following explanation when we first introduce the SST1 array in our exposition of the acrocentric pangenome graph:

The SST1 elements in the segmentally duplicated region, also known as NBL2 (Nishiyama et al. 2005), are GC-rich 1.4-2.4 kb long sequences arranged in tandem clusters (Epstein et al. 1987), located throughout the genome including near the centromeres of the short arm of the acrocentric chromosomes 13, 14, 21 (Hoyt et al. 2022). The SST1 array size is variable in the human population (Tremblay et al. 2010) and its methylation status is clinically relevant to cancer (Samuelsson et al. 2017; González et al. 2021). SST1 repeats on chromosomes 13, 14, and 21 in T2T-CHM13 are highly similar to each other (Hoyt et al. 2022), consistent with homogenization via recombination.

6. Page 15: As the accuracy of SNPs is much important to calculate the LD decay. The authors claimed they used the marekerd within the variant space. But this concept is not

clear in the manuscript. What's the features of the variant space? Did it mean the SNP directly called with pangenome graph, e.g., vg deconstruct?

With variant space, we meant the subset of variants present in the HPRCy1 dataset and then represented in the all-acrocentric pangenome graph that we specifically built for the analysis presented here, using the centromere-spanning contigs. We agree that this terminology is jargon that is unlikely to be understood by a wider audience. Therefore, we have removed this wording and extensively rewritten the text to make it more broadly accessible and made reference to the specific algorithm used to extract SNPs from the PVG (<https://doi.org/10.1089/cmb.2017.0251>) in the introduction to the section “Linkage disequilibrium decay in pseudo-homologous regions”.

7. If the ongoing exchange valid, the larger cohorts may can help for validation. Maybe we could see these region found by acro-PVG have overlap with genomeAD or TOPMED translocation diversity although it was short-reads based. And maybe we can see more rampant recombination on cancer samples, since it have more aneuploid in naturally and maybe more non-homologous recombination.

We agree with the reviewer that we must study these regions in the context of diverse large scale cohorts. However, to do so will require a research interface with the groups who manage their associated genomic data. For instance, we will need to realign the entirety of these cohorts against at least T2T-CHM13 in order to utilize them in research into the PHRs. This alone is a large project that might require significant research funding or support from established groups researching these cohorts. We will additionally need to develop methods to infer phased structural haplotypes from short read data and an HPRC-like pangenome reference.

Analyzing cancer samples is a great idea, but it will require improvements to assembly methods and benchmarking to develop confidence that we can build assemblies of sufficient quality to support the kind of analysis that we've presented here. One study direction would be to develop assemblies from cells representative of the clonal expansion of a tumor, and then to combine them in a PVG to look for signals of (somatic, mitotic) recombination and homogenization between the SAACs. This would be a fascinating topic for future study and we thank the reviewer for the suggestion.

The reviewer's ideas highlight important directions for future research into the acrocentric PHRs, and we have mentioned them in the discussion in hopes that they spur efforts in this direction. Given the association of this region with the most metabolically intense class of cell activity (protein synthesis), it seems very plausible that it is involved in phenotypes like cancer and chronic disease. It has not been accessible to study until the present, and our work is the first to provide a complete perspective on sequence evolution in this locus—a necessary foundation for future work here.

8. Considering that this manuscript will be of broad interest, it would be better to have an figure illustrate the phenomenon for general readers.

We have added a new graphical summary of our work (**Figure 6**) which is designed to display the main conclusions and implications of the acrocentric PHRs. We hope that this figure will improve the communication of our findings to general readers. This figure displays the PHRs and their distribution over acrocentrics (**Figure 6A**), provides a visual hint of what physical proximity means for heterologous recombination (**Figure 6B**), shows the most common form of recombination (non-crossover or gene conversion) in the context of the PHRs (**Figure 6C**), and illustrates the genesis of a Robertsonian translocation through crossover type recombination in the SST1-associated region of the PHRs on 13p (or 21p) and the relatively-inverted region on 14p (**Figure 6D**).

Referee #2 (Remarks to the Author):

The rules governing the behaviour of the short arms of human acrocentric chromosomes are one of the major remaining unanswered question in human chromosome biology. These chromosome arms bear nucleolar organiser regions (NORs) that are composed of ribosomal gene (rDNA) arrays and seed formation of the nucleolus. How sequence uniformity of rDNA arrays across these chromosome arms is maintained is a key question. It has been hypothesised for quite some time that exchanges/recombination between the short p-arms of heterologous acrocentric chromosomes is involved. This is further supported by the shared satellite composition of these chromosome arms, as demonstrated by fluorescent in situ hybridisation (FISH) experiments. One noted example being sequence similarities of higher order (HOR) alpha satellite arrays on chromosomes 13/21 and 14/22.

Through the efforts of large sequencing consortia, the Telomere to Telomere (T2T) and latterly the Human Pangenome Reference consortium (HPRC), evidence for exchanges between heterologous chromosomes can now be described at the highest level of resolution, DNA sequence. Primarily using PacBio HiFi reads supplemented with Nanopore ultralong reads, the T2T determined the close to complete genome sequence, T2T-CHM13, for a highly homozygous genome of a complete hydatidiform mole cell line (CHM13). The only significant gaps being the longer rDNA arrays on some of the acrocentrics, which could none the less be modelled using other approaches. Ongoing work in the HPRC has yielded 94 haplotype-resolved assemblies for a series of normal diploid human cell lines (HPRCy1), based on HiFi sequencing. For some cell lines, including HG002, more complete genome drafts have been obtained by combing HiFi with Nanopore ultralong reads, HiC and BioNano (optical mapping) data.

In this manuscript, HPRCy1 is used to explore patterns of variation among these chromosome arms and to reveal evidence for “ongoing recombination exchange” between heterologous chromosomes. With reference to the pseudo-autosomal regions (PAR1 and PAR2) on the X and Y sex chromosomes, the elements on acrocentric p-arms supporting these exchanges are described as “pseudo-homologous regions”. This is a significant advance on our understanding of the chromosome biology of these critical regions of our genome and will undoubtedly seed further investigation into when, where and how these exchanges occur. It will likely also lead to a molecular explanation of how Robertsonian translocations, the most common form of chromosomal translocation in the human population, arise.

We thank the reviewer for their perspective on the topic of study and the context of our work within it.

The possibility that this work will seed further analysis has prompted us to review several technical aspects of our analysis—including the definition of the pseudo-homologous regions (PHRs), which we have modified to be consistent with standard definitions of the sex chromosome PARs.

We furthermore have brought greater focus on the issue of Robertsonian translocations and their relationship to the patterns of recombination exchange that we observe in the PHRs. This led to several improvements to the manuscript. A deeper review of existing literature led to our recovery of a prior work that uses GRC reference clones to establish a likely breakpoint for the most common Robertsonian translocation. A re-assessment of these FISH experimental results in the context of T2T-CHM13 demonstrates that they are consistent with the 13p/14p/21p SST1-linked PHRs hosting the crossover point for 14/21 Robertsonian translocation. We have updated the manuscript to reflect this finding, and added a figure panel to link our pangenome based results with prior cytogenetics (**Figure 4D**).

We additionally considered the potential role of PRDM9 in meiotic recombination (itself a known driver of Robertsonian translocations). We find that PRDM9 DNA motifs are highly enriched in rDNA repeats, completely depleted in acrocentric centromeres, and present in higher density within the 1.4kb SST1 array repeat unit relative to the non-rDNA regions of 13p, 14p, and 21p (**Figure 3F and Supplementary Figure 37**). These findings are suggestive that PRDM9 motifs are tolerated and perhaps enriched in these repetitive sequences. Their absence from acrocentric centromeres matches previous evidence from population sequencing that demonstrates there is little or no recombination within human centromeres. Their enrichment in rDNA is to our knowledge a novel finding that may help to explain the very high rate of recombination in the rDNA arrays, also known as concerted evolution. We believe this interesting, if incidental, bioinformatic finding that lends an additional biological dimension to the work, and hope we have presented it cautiously enough to not mislead readers.

Comments

The initial set of results describes visualisation of similarities between acrocentric short-arms by grouping sequences according their homologies, thus identifying “chromosome communities”. Fig 1A and B shows a homology map with >16k contigs of 1Mb from HPRCv1 covering 98.72% of the genome. Chromosome identity was determined by mapping to T2T-CHM13 and GRCh38 and the number of hits was limited to 3 for ease of visualisation. This is further refined by use of community detection algorithm that identified a community involving the p-arms of all five acrocentrics as well as the 21 and 22 q-arms. Although not stated, I assume that inclusion of these q-arms is related to their short length compared to 13-15, rather than direct sequence homologies. While really just confirmatory in nature, these results set the scene for a more refined comparison of acrocentric p-arms at the sequence level.

The reviewer is right, the inclusion of the q-arms of chromosomes 21 and 22 in the acrocentric communities is probably related to them being shorter than chromosomes 13,

14, and 15. We indicate this likelihood in the revised manuscript. We note that the process is not only visualizing the similarities, but also applying a quantitative framework (the community detection) to a much richer all-vs-all mapping graph, which is itself too complex to visualize.

As HPRCy1 assemblies are derived solely from HiFi reads derived from normal diploid cell lines, they are less complete than T2T-CHM13. Nevertheless, by building assemblies from contigs anchored on the long q-arm of each acrocentric and spanning the centromere, sequences of rDNA proximal regions for each p-arm and haplotype could be determined. As before rDNA spanning contigs were rare. To bolster this list two versions of a more complete genome from cell line HG002, T2T-CHM13 and relevant contigs from GRCh38 were included, resulting in a collection, referred to as HPRCy1-acro. Pan genome graph builder (PGGB) was used to build pangenome variation graphs (PVGs) representing all pairwise alignments between contigs. The visualisation of these alignments is shown in a 2d graph in Fig 3. Not unsurprisingly, it reveals that all the across are anchored in rDNA, and that centred on alpha satellite HORs, 13 and 21 show close similarity, as do 14 and 22 and 15 is more isolated. More

intriguing is identification of “pseudo-homologous regions” around tandem arrays of SST1 elements close to centromeres on T2T-CHM13’s 13p, 14p and 21p (identity >99%). However, I am confused by the statement in the legend to Fig 3 that they are in the “direct orientation” on these chromosome arms. As I understood from Fig 4 in the original Nurk et al T2T Science paper, this region was inverted on 14 relative to 13 and 21. Is this an error? If so, could it be that this is due to many of the contigs in HPRCy1-acro being based only on HiFi and not supported by Nanopore reads? This should be clarified as it may have relevance to the origin of the most common Robertsonian translocations, involving these region on the same chromosomes.

That statement was an error that we corrected in the revised version of the manuscript. We apologize for the mistake. The reviewer is right, in the T2T-CHM13 genome, the pseudo-homologous region around the SST1 element *is* inverted on chr14 with respect to 13 and 21, and this pattern is also reflected in our HPRCy1 acrocentric pangenome variation graph (acro-PVG). Therefore, there is no inconsistency between HiFi and Nanopore data. To further investigate that, we performed additional analyses on the orientation of the contigs, focusing on the PHRs centered in the SST1 (± 1 Mbp), adding new supplementary figures (**Supplementary Figures 23-25**). To simplify review, we report here **Supplementary Figure 24** that shows multiple untangling of T2T-CHM13, GRCh38, HG002-Verkko haplotypes, and a few HPRCy1-acro contigs versus T2T-CHM13:

Target
 ■ chm13#chr13
 ■ chm13#chr14
 ■ chm13#chr15
 ■ chm13#chr21
 ■ chm13#chr22

Strand
 -
 +

Chromosome 14 results are represented, in the chr14:5,960,008-7,988,409 region (censat_14_39 coordinates \pm 1Mbp). Transparency shows the different orientations of the mappings: low transparency indicates direct orientation, while high transparency (corresponding to washed-out colors) is in the reverse complement orientation. We display all mappings above 90% estimated pairwise identity. To analyze simultaneous hits to all acrocentrics, each grouping shows the first 3 best alternative mappings. The figure shows that all but one of chromosome 14's contigs map in forward orientation on T2T-CHM13 chromosome 14 (gold rectangles), while their mappings are inverted on chromosomes 13

and 21 (transparent orange and cyan rectangles). In the sole exception, HG01071#2#JAHBCE01000082.1, the assembly suggests that the locus is on chromosome 14, but oriented the same way as the homologous loci on CHM13's 13 and 21. The same haplotype appears to host a structural variant downstream of the locus, which could be consistent with an inversion polymorphism, which can be associated with deletions, but caution is warranted as we only observe a single instance of this pattern and lack cross-validation beyond the FLAGGER confident regions. We leave further exploration of polymorphism of the inversion on 14p to future studies based on new near-T2T assemblies that will become available in subsequent releases from the HPRC.

The core findings of this manuscript are displayed in Fig 4 where the acrocentric PVG is “untangled” by deriving pairwise alignments, mapping each part of each HPRCy1-acro contig against T2T-CHM13, retaining only those that have >90% estimated identity. The amount of data presented in this figure is very large and the annotation is incomplete, particularly in relation to the colour coded repeat masking data. Nevertheless, evidence for significant levels of recombination/exchanges within “pseudo-homologous regions” (PHRs), particularly involving 13, 14 and 21, is very striking. Diversity entropy is another metric describing the degree of disorder in the untangling of HPRCy1-acro contigs. This begins at zero in the acrocentric q-arms where there is no evidence of exchange to approaching 1 in PHRs.

We have provided a legend for the Centromeric Satellite Annotation (CenSat Annotation) track in a supplementary figure. In general, the reviewers' comments encouraged us to reduce the complexity of the “untangling” homology mosaic figures in the main text. We now focus on a single chromosome (13) in **Figure 3**, and add further tracks of annotation on top of these to illustrate the development of the entropy metric that we use to establish the PHRs' boundaries. We have also updated our terminology surrounding the two kinds of entropy metrics that we use. For reader convenience, we indicate that these are “regional” and “positional” homology entropy, describing them by their utility for our study. The positional homology entropy is helpful for establishing more sharp boundaries to the PHRs, which this terminology also helps to explain.

The data presented in Fig 5 focuses on a subset of the HPRCy1-acro contigs for 13, 14, and 21. It shows the best 5 mappings to T2T-CHM13 for each contig. This data further refines PHRs as linked to rDNA arrays and surrounding the STS arrays on 13, 14 and 21. Moreover, better estimates of their lengths can be determined. Chromosomes 13, 14 and 21 have PHRs of comparable sizes to the PARs on sex chromosomes. In line with previous data in this manuscript, chromosome 15 has the shortest PHR at ,0.5Mb. Does this higher resolution data provide any information on the lengths of typical exchanges. Are their lengths compatible with gene conversion or do they arise from double crossover events?

We note that it can be difficult to distinguish these two processes using only sequence data from a population setting. In our case, the data is compatible with both gene conversion and crossover events, and we also cannot confidently distinguish if it arises from meiotic or mitotic recombination. Single crossover events between PHRs can in principle lead to swapping of the entire distal region of the p-arms, and double crossovers would resemble gene conversion. The handful of assemblies which cross rDNA arrays do suggest that the

distal regions are nearly identical and duplicated on all acrocentrics, which is compatible with crossover between heterologs. However, because our data is not conclusive in terms of its resolution of the rDNA and rDNA-distal regions of the p-arms, we emphasize gene conversion (which is significantly more common than crossover) as a likely primary source for the patterns we observe (**Figure 6C**). That noted, the existence of Robertsonian translocations (**Figure 6D**) with recurrent breakpoints within the 13p/14p/21p SST1-linked PHR (**Figure 4C**) indicates that *some* crossover type recombination is occurring within these regions. T2T-CHM13 presents diverse satellite sequences in the distal regions of the short arms. If crossover occurs frequently, our intuition suggests that these satellites may swap freely between chromosomes and exhibit a very high level of diversity in the population. We look forward to new data that allows us to evaluate this possibility with precision.

To lend more insight to this issue, we have added a small study of PRDM9 motif occurrences relative to the SAACs and PHRs. PRDM9 motifs are enriched in the rDNA arrays in CHM13. They are extremely depleted in the centromeres (consistent with reports of almost nonexistent crossover in these regions). However, we find an interesting association with PRDM9 motifs and the SST1 arrays. The SST1 array presents the highest local density of PRDM9 motifs outside of the rDNA anywhere on the acrocentric short arms (**Figure 3F and Supplementary Figure 37**), suggesting the potential for SST1 to be a meiotic recombination hotspot.

The next data concerns the decay of linkage disequilibrium by plotting the r^2 allele correlation versus distance for variants on the q-arm, the p-arm, and within the PHRs. They observe that for chromosomes 13, 14, and 22, the LD decay is faster in the PHRs compared to p- and q-arms and point out that “This effect is strongest within the chromosomes that other analyses suggest to share the most homologous sequence: 13, 14, and 22”. I find this somewhat strange as much of the previous data concentrates on homologies between 13,14 and 21 (not 22) involving PHRs that include SST1 elements. Can the authors comment.

Indeed, the reviewer is correct that this statement was incompatible with our results. We have revised both this section and the analysis it describes significantly to resolve these concerns.

The LD analysis in our initial submission was hampered by short PHR tract lengths, which in turn resulted from overly-aggressive filtering of the positional homology entropy metric that we used to establish the PHRs. We realized this when deepening our evaluation of the sex-chromosome PARs. Ultimately, we decided to change the PHRs' definition so that our method would precisely highlight previously established PARs when applied to a sex chromosome PVG.

By applying the change to our results, we derive longer PHRs, which allows an increase in the number of SNP marker pairs that are available on most chromosomes. This had a particularly strong impact on the results for 21 and 15 because it increased the number of data points substantially. Our overall conclusion does not change, but we are now able to confidently observe the same trends for all acrocentrics except 15, which still has too few SNP marker pairs to provide clear results in its PHRs. We have adjusted the text to reflect this update. We believe that this update increases the clarity, but not impact, of our findings.

Additionally, the definition of PHRs is likely to evolve as we obtain many complete assemblies of entire SAACs, a point that we now touch on in our discussion. For instance, the rDNA arrays probably fall inside PHRs, but our current assemblies do not span the arrays and we prefer to not overstate the scope of these regions without complete evidence.

Finally, as most HPRCy1-acro contigs are assembled using only HiFi contigs there is an attempt to validate this data by comparing the two HG002 genome builds HG002-HPRCy1 and HG002-Verkko. The results suggest that HiFi-only assemblies are “likely” accurate representation of haplotypes.

Validation of our results is difficult, but the Verkko-based comparison lends confidence that recurrent homology mosaic patterns are not driven by artifacts of sequence assembly. However, to reduce the technical burden on readers of the manuscript, we note that we have moved the section on validation to the methods, and refer to it in summary at the critical point in the text where the homology mosaic plots are introduced.

The authors rightly declare that their population-based study that does not speak to mechanism or timing of exchanges/recombination between heterologous acrocentrics. However, it would be interesting to hear their views on whether examination of family pedigrees would be expected to shed light on mechanism and frequency of exchanges. For example like Robertsonian translocations do they primarily arise during female meiosis.

Indeed, it would be fascinating to be able to observe heterologous acrocentric recombination in a family pedigree. Some caution is warranted however, in that the scale of such a study might need to be rather large before it could accumulate numerous positive examples for study. We refer the reviewer to estimates that we have developed in our response to Reviewer 3 that suggest that we would require hundreds of trios to develop power to study this issue by looking at de novo events in pedigrees.

Additional comment:

In a number of instances the text describes association of acrocentric chromosomes with the “nucleolar membrane” (eg last paragraph of discussion). It is now firmly established that nucleoli are membraneless organelles.

We thank the reviewer for this catch. We have fixed the error in the revised version of the manuscript.

Referee #3 (Remarks to the Author):

That acrocentric chromosomes share sequence identities, as discovered in the T2T human genome sequencing project, suggestive of ongoing recombination is exciting. Indeed, rDNA clusters from different clusters localizing to the nucleolus within cells makes for an opportunity for non-homologous chromosomes to potentially share sequence. This study seeks to detect evidence of recombination between non-homologous chromosomes rDNA regions, resulting in shared sequences. The authors analyze high-quality, haplotype-resolved, sequences of a diversity of humans to address this question. They provide convincing evidence that recombination has shaped the evolution of the human

acrocentric chromosomal regions around the rDNA clusters. This is an important finding and step to understanding these complex genomic regions. However, outside of simply finding there are shared sequences across the non-homologous chromosomes, the manuscript does not address some fundamental questions on these shared sequences. Moreover, the manuscript text is heavy on jargon, making it difficult to follow for the general genetics/genomics/bioinformatics reader.

On fundamental questions about shared sequences: By reading the reviewer comments in the 'Major concerns', we believe that the fundamental questions to which he/she refers are: significance of the shared sequence homologies; distinction between gene conversion and crossing over; frequency of sequence exchange between the acrocentrics; parallelism between PHRs and X-Y PARs. While we agree that these are all valid and interesting questions stemming from the major findings of our work, we believe they are beyond the scope of the work presented here, and each of them is in fact an entirely stand-alone project that may depend on substantial investment in new data and methods. Nevertheless, we took time to do preliminary investigations for each of them, and we give point by point responses to each concern. Where possible, we have motivated improvements to the manuscript in response to these questions.

Considering jargon in the manuscript text: We acknowledge that, in the original manuscript, we have not presented our results in a broadly accessible manner. We took this aspect seriously and rewrote the whole manuscript by focusing more on chromosome biology and eliminating avoidable jargon. This has led to greater simplicity and focus in the sections describing the technical objects that we develop (the chromosome community analysis and acrocentric pangenome graph), and more exposition of patterns of recombination and relevant biological features revealed in our study of the pangenome graph.

Major concerns:

1. What is the significance of the shared sequence homologies? For example, do they provide an anchor for pairing rDNA clusters the nucleus? Outside of stating just that these shared homologies exist, it would be important to provide an understanding of what is driving the sharing of these sequences.

We believe that the sharing may occur because the NORs are fulfilling the same biological roles. We now touch on this in the revised discussion.

Expounding on this topic: Since all acrocentric p-arms participate in the formation of the same nuclear compartment—the nucleolus—they must have sequences that can do this, and must not have sequences that can interfere with this. This is probably analogous to why telomeres and centromeres are homologous, with one important remark: there are generally fewer nucleoli in the nucleus (1-3, rarely more) than acrocentric chromosomes (10). Therefore, multiple acrocentric p-arms would participate in the formation of the same nucleolus, so they have to be functionally equivalent or at least cognate. It's also important to note that the biosynthetic program of ribosome biogenesis that takes place in the nucleolus is one of the key housekeeping processes in the cell. It works constitutively and must be effective if the cell wants to be viable, so there is probably a pretty strong evolutionary pressure against divergent sequences that do not support the effective

nucleolar formation and/or function. We expect this same pressure to drive the homogenization of the sequences.

2. Can the authors distinguish between gene conversion or crossing over to explain the homologous sequences in the acrocentric chromosomes? It is only inferred that there are homologies, but how these homologies are generated, outside of speculating that recombination, is important. Examining homogenization tract lengths be assessed across the diversity panels to assess if there are any patterns that illuminate the recombination dynamics?

In principle, it should be possible to distinguish rates of gene conversion and crossing over in the pseudo-homologous regions (PHRs). However, we have restrained ourselves from attempting to do so due to limitations of our current data and the lack of available methods to measure these processes in the context of a PVG. Our data consists of a very small number of haplotypes, and the assemblies across the SAACs remain relatively incomplete. Future releases of the HPRC will expand the scale of this data set by an order of magnitude, while greatly improving assembly quality so that all assemblies are similar to HG002-Verkko in contiguity and quality. We believe it will be more clear to focus on the issue of recombination type with such a data set available. A major issue is the possibility of local misassembly in the PHRs, which might suggest inaccurate homogenization tract lengths between diverged haplotypes.

To extend our work to a diversity panel based on short reads would require remapping the entire data set to T2T-CHM13 and/or the HPRC pangenome. Ideally, we would be able to use these alignments to infer phased haplotypes across the acrocentric chromosomes—but methods to do so in the strange, polyploid world of the PHRs (or directly on the PVG) do not yet exist.

We hope that the reviewer understands the limitations of our data and our desire not to overstep the bounds in which we can make confident claims about the processes driving the existence and evolution of the PHRs. We now note in the discussion that we anticipate future extension of our work to study recombination type using improved assemblies of the SAACs and new methods to integrate short read based cohort data into our analysis. In the aggregate, our data strongly suggest recombination between heterologous chromosomes. However, for technical reasons, elucidating the fine details of this process remains a topic for future, appropriately powered, studies.

An aside that the reviewer's comment prompted: A recent study of a recombination hotspot in yeast suggests that branch migration is an essential step in crossover formation (<https://doi.org/10.1016/j.molcel.2021.08.003>). An intriguing possibility is that non-crossover recombination might be relatively more likely than crossover between lower-identity regions because branch migration will be inhibited by variation. Thus branch migration is itself a kind of homology check that prevents crossover in the presence of insufficient homology between chromatids. If this were true, then we might expect recombination between diverged sequences to be predominantly non-crossover with short homogenization tract lengths (as perhaps occurs in strata 4 and 5 on the X and Y as discussed in a following response). We worry that this is overly speculative and have not included it in our discussion. We do cite this study to highlight the complex and variegated patterns of homogenization that arise from

both crossover and non-crossover recombination, which present difficulty for sequence-based studies of recombination type.

3. The authors argue that there is ongoing sequence exchange between the acrocentric chromosome autosomes. What is the frequency of this exchange? Does it happen every generation? For example, to assess the frequency of recombination, could the authors use trio data or comparing diverged haplotypes? Are some regions more “hotspots” for exchange versus other regions?

We believe that using trios or pedigree data to estimate rates might be extremely difficult. Assuming that the PHRs behave similarly to the rest of the genome, where we have established rates of recombination, we can work back to an estimate of the rate at which we expect to observe recombination in the PHRs. Starting from an estimate of non-crossover recombination of $5.9e-6$ /bp/generation, we would expect only a few non-crossover events per generation in the PHRs (<https://doi.org/10.7554/eLife.04637>). Given that crossover is perhaps 1:10 as frequent as non-crossover, this would lead to a baseline expectation of less than single crossover in the acrocentric PHRs per generation. Many of these may occur between homologs, and only a fraction may be between heterologs. Given these estimates, we would expect to need hundreds of trios to establish a well-powered picture of meiotic “ectopic” recombination rates in the PHRs. Alternatively, we could take the rate of de novo Robertsonian translocations, which could be around 1/1000 per generation, and work back to a rate of heterologous PHR crossovers once every one to two hundred meioses. Observing these events directly is a daunting task—a “fishing expedition” even—leading to a paper titled “FISHing for Acrocentric Associations between Chromosomes 14 and 21 in Human Oogenesis,” which we cite (<https://doi.org/10.1016/j.ajog.2004.02.062>). This paper concludes that in “a total of 272 nuclei at zygotene, pachytene, and diplotene substages with informative hybridization signals, 1 pachytene nucleus was observed to have fluorescent signals for chromosomes 14 and 21 that could represent a heterologous association between these 2 acrocentric chromosomes.” To highlight this, in our discussion we note that 1/220 nuclei at pachytene shows evidence for a 14/21 association. Again, this rate lies in the expected order based on our estimates, and it suggests that we would need to look at large numbers of meioses to confidently observe patterns in the recombination landscape of the PHRs.

This said, we do expect there to be recombination hotspots, as elsewhere in the human genome, and these may be enriched or distributed unevenly in the SAACs. To explore this with available data, we have scanned T2T-CHM13 for PRDM9 motif hits. To our knowledge, this may be the first such implementation of this analysis to the SAACs, and it leads to some suggestive patterns. First, we find that rDNA repeats are enriched for PRDM9 motifs, which may be consistent with reports of recombination rates of 10% per cluster per generation (Stults 2010, DOI 10.1101/gr.6858507). Also, we find that the SST1 array, which lies at the center of a region of the PHRs on 13, 14, and 21 that we focus on, is enriched for PRDM9 hits. These would be obvious candidate recombination hotspots, and that would generally match the results observed in e.g. the genome homology mosaic plots.

4. The authors argue “In the homology mapping graph of the HPRCy1, only the acrocentric and sex chromosomes form combined communities containing multiple chromosomes. The sex chromosome community reflects the pseudoautosomal regions (PARs) on X and Y (23), which are telomeric regions where these otherwise non-homologous chromosomes recombine as if they were homologs.” While the four acrocentric chromosomes are near each other in the community plot (Fig. 1), relating it to the X and Y doesn’t seem justified. The X and Y are separated on the plot, similar to other chromosomes. Other autosomes, for example 11 and 1 or 18 and 11 look closer, than the sex chromosomes. Do the other autosomes have examples of non-rDNA repeats, satellite DNA, or other repetitive sequence that promote recombination between the autosomes, like the rDNA regions?

Figure 1 shows a strongly simplified view of the real mapping graph because we considered only 3 mappings for each node (contig) for computing the shown graph layout, instead of up to 93 (number of the haplotypes minus 1) mappings. Furthermore, the 2D layout of the mapping graph is rendered using Gephi, a tool for exploring and manipulating graphs, which features an interactive and non-reproducible layout mechanism. For this reason, the distances between the communities can not be used as a metric for the sequence similarity of the contigs that compose them. At the same time, despite the reduced number of mappings, the acrocentric community (and less strongly the sex community) tends to be visible also in this simplified graph. In the full mapping graph, which has a too high number of connections to be visualizable, the underlying community structure is the one reported in Figure 1C, with only the acrocentric chromosome and sex chromosomes falling in chromosome non-specific communities. Therefore, we don’t have any other strong signals of non-rDNA repeats that could promote recombination between the autosomes. We apologize for our lack of clarity in the homology mapping graph visualization. In the revised manuscript we have worked to improve our discussion of the meaning of the visualizations in Figure 1 and clarify why we apply a quantitative technique for community detection.

5. To build on using the sex chromosomes as a control, the authors could map or highlight strata 4 + 5 of the X and Y, which have similar levels of nucleotide identity and see if they connect to each other on the community plots? These last two strata are the primary regions known to still undergo homologous recombination between the X and Y, albeit not that frequently.

We first took the reviewer’s suggestion to encourage a stronger use of the PARs as a positive control. We revised our PHR definition so that our method, when applied to an X+Y graph, yields the previously-established PARs as “PHRs” (**Supplementary Figure 35**). In this context, we did not find any signal on strata 4 and 5 even under more permissive alignments. This analysis was nevertheless useful, because, in short, we found that we were too-aggressively filtering candidate regions of the PHRs, leading to more fragmented regions even in the case of X and Y. This in turn has improved our analysis of linkage disequilibrium decay by increasing the number of available marker pairs.

We then investigated the reviewer’s question about strata 4 + 5 through two mechanisms, resulting in (**Supplementary Figure 1**) and an update to the discussion of the sex chromosomes in the context of the community analysis.

First, we used a mapping analysis of T2T-CHM13's X and Y to establish the patterns of inversion, rearrangement, and sequence identity that define these strata. Visualization with Saffire (<https://mrvollger.github.io/SafFire/>) reveals that strata 5 and 4 feature numerous inversions and some rearrangements between X and Y (**Supplementary Figure 1A-B**). Furthermore, we observe that the nucleotide identity in these regions is around or below 90%, which tends to be much lower than most of the homologies that underlie the acrocentric PHRs. This high divergence plus the inversions and partial rearrangement likely inhibit recombination between these regions.

Then, we used mapping to identify contigs that fit into these strata and highlighted them on the community plots (**Supplementary Figure 1C-D**). This allowed us to understand that the two communities in the sex chromosomes are (1) all of Y, plus most contigs in the PARs, plus Xq, which is apparently tied in due to the X-transposed region and possibly PAR2, and (2) proximal regions of Xp, including some PAR1-matching contigs and all contigs that match strata 4 and 5. This region of X thus appears to not recombine with Y sufficiently to pull its contigs into the same community. As anticipated by the reviewer, the distinction between the PARs and strata 4 and 5 results in important features visible in the community analysis.

Reviewer Reports on the First Revision:

Referees' comments:

Referee #1 (Remarks to the Author):

Guarracino et.al did a thorough revision of the manuscript based on comments from the first round. The additional evidence from recurrent Robertsonian translocations and the pseudo-homologous regions (PHRs), rewritten details, and graphical summary give access to broader readers. The demonstration of the ROB was noteworthy and the illustration in Figure 6 was easy to understand and succinct. This work not only gives population-based sequence confirmation for heterologous recombination between acrocentric chromosomes, but also inspires efforts toward understanding the mutation process with graph pangenome on large population-level complete assemblies. All my previous concerns have been well addressed and I have no further comments.

Referee #2 (Remarks to the Author):

In this revised manuscript the authors provide robust data supporting the identification of so called pseudo homology regions (PHRs) present on the short arms of acrocentric chromosomes (SAACs) in humans, located in the interval between nucleolar organiser regions (NORs) and centromeres. They further provide evidence for frequent exchanges between heterologous SAACs. The major improvements in this revised manuscript are the improved accessibility of the bioinformatics to non-specialist readers and the new data concerning Robertsonian translocations (RTs). The latter adds true biological significance to their results.

With respect to my review comments on the original manuscript, I can report that they have all be adequately addressed. The inclusion of data and discussion of RTs, summarised in the new figure 6, is particularly gratifying. The demonstration of concerted evolution of SST1 arrays on SAACs and the finding that PRDM9 binding motifs are enriched on both rDNA and SST1 arrays provide strong clues as to how exchanges between heterologous SAACs and RTs arise.

With respect to the other reviewers comments, it would appear to this reviewer (a non-specialist) that their comments on the bioinformatic approaches employed have been satisfactorily be dealt with. As outlined above, I would argue that the issue of 'biological significance' has been also addressed.

In summary, I believe that this revised manuscript is greatly improved and represents a significant advanced in our understanding of the complex genetics of these critical regions of our genome.

Referee #3 (Remarks to the Author):

The authors have made significant improvements to the manuscript, addressed all of my major concerns. The main text is written with greater clarity, removing much of the jargon, so will be accessible to a broader readership. The addition of the Robertsonian Translocation potential is nice as it highlights and improves the biological significance of the PHRs. The added final figure, Figure 6, is also a welcome addition.

Author Rebuttals to First Revision:

Referees' comments:

Referee #1 (Remarks to the Author):

Guarracino et.al did a thorough revision of the manuscript based on comments from the first round. The additional evidence from recurrent Robertsonian translocations and the pseudo-homologous regions (PHRs), rewritten details, and graphical summary give access to broader readers. The demonstration of the ROB was noteworthy and the illustration in Figure 6 was easy to understand and succinct. This work not only gives population-based sequence confirmation for heterologous recombination between acrocentric chromosomes, but also inspires efforts toward understanding the mutation process with graph pangenome on large population-level complete assemblies. All my previous concerns have been well addressed and I have no further comments.

Referee #2 (Remarks to the Author):

In this revised manuscript the authors provide robust data supporting the identification of so called pseudo homology regions (PHRs) present on the short arms of acrocentric chromosomes (SAACs) in humans, located in the interval between nucleolar organiser regions (NORs) and centromeres. They further provide evidence for frequent exchanges between heterologous SAACs. The major improvements in this revised manuscript are the improved accessibility of the bioinformatics to non-specialist readers and the new data concerning Robertsonian translocations (RTs). The latter adds true biological significance to their results.

With respect to my review comments on the original manuscript, I can report that they have all be adequately addressed. The inclusion of data and discussion of RTs, summarised in the new figure 6, is particularly gratifying. The demonstration of concerted evolution of SST1 arrays on SAACs and the finding that PRDM9 binding motifs are enriched on both rDNA and SST1 arrays provide strong clues as to how exchanges between heterologous SAACs and RTs arise.

With respect to the other reviewers comments, it would appear to this reviewer (a non-specialist) that their comments on the bioinformatic approaches employed have been satisfactorily be dealt with. As outlined above, I would argue that the issue of 'biological significance' has been also addressed.

In summary, I believe that this revised manuscript is greatly improved and represents a significant advanced in our understanding of the complex genetics of these critical regions of our genome.

Referee #3 (Remarks to the Author):

The authors have made significant improvements to the manuscript, addressed all of my major concerns. The main text is written with greater clarity, removing much of the jargon, so will be accessible to a broader readership. The addition of the Robertsonian Translocation

potential is nice as it highlights and improves the biological significance of the PHRs. The added final figure, Figure 6, is also a welcome addition.

We thank all the reviewers for their attentive scrutiny and expertise. Their inputs have been instrumental in significantly enhancing the manuscript, particularly by helping us formulate a concise hypothesis on the sequence basis for Robertsonian translocations in humans and substantially refining our presentation to be more accessible to a general audience. Furthermore, we believe that the reviewers' efforts have enabled us to establish a connection between bioinformatic techniques designed for analyzing pangenomes across multiple frames of reference and resolving challenging biological queries that cannot be tackled by concentrating solely on a single reference genome.